# A skin-permeable polymer for non-invasive transdermal insulin delivery

Qiuyu Wei[1,4], Zhi He[2,4], Zifan Li[1,4], Zhuxian Zhou[1,4], Ying Piao[1], Jianxiang Huang[2], Yu Geng[1], Runnan Zhang[1], Yaqi Fu[2], Jiayi Ye[1], Yue Yuan[1], Haoru Zhu[1], Jiaheng Zeng[1], Yan Zhang[1], Quan Zhou[1], Mingyu Xu[2], Shiqun Shao[1], Jianbin Tang[1], Jiajia Xiang[1✉], Rongjun Chen[3✉], Ruhong Zhou[2✉] & Youqing Shen[1✉]

Non-invasive skin permeation is widely used for convenient transdermal delivery of small-molecule therapeutics (less than 500 Da) with appropriate hydrophobicities[1]. However, it has long been deemed infeasible for large molecules—particularly polymers, proteins and peptides[2,3]—due to the formidable barrier posed by the skin structure. Here we show that the fast skin-permeable polyzwitterion poly[2-(*N*-oxide-*N,N*-dimethylamino)ethyl methacrylate] (OP) can efficiently penetrate the stratum corneum, viable epidermis and dermis into circulation. OP is protonated to be cationic and is therefore enriched in the acidic sebum and paracellular stratum corneum lipids containing fatty acids, and subsequently diffuses through the intercorneocyte lipid lamella. Beneath the stratum corneum, at the normal physiological pH, OP becomes a neutral polyzwitterion, 'hopping' on cell membranes, enabling its efficient migration through the epidermis and dermis and ultimately entering dermal lymphatic vessels and systemic circulation. As a result, OP-conjugated insulin efficiently permeates through the skin into the blood circulation; transdermal administration of OP-conjugated insulin at a dose of 116 U kg$^{-1}$ into mice with type 1 diabetes quickly lowers their blood glucose levels to the normal range, and a transdermal dose of 29 U kg$^{-1}$ normalizes the blood glucose levels of diabetic minipigs. Thus, the skin-permeable polymer may enable non-invasive transdermal delivery of insulin, relieving patients with diabetes from subcutaneous injections and potentially facilitating patient-friendly use of other protein- and peptide-based therapeutics through transdermal delivery.

Transdermal delivery of biomacromolecules, such as proteins and peptides[4], through topical application is advantageous in terms of convenience, patient compliance, avoiding denaturation and minimal first-pass effects[5,6]. However, it has been considered to not be feasible owing to the impermeable barriers presented by the unique skin structure[2,3], consisting of hydrophobic stratum corneum (SC) layers, a 10–15-μm-thick matrix with dehydrated and dead corneocytes embedded in highly ordered lipid layers, as well as the tight junctions in the viable epidermis and dermis[7]. Subdermal insulin injection is still the standard treatment for type 1 and advanced type 2 diabetes and is associated with pain, needle phobia, skin complications and poor patient compliance[8]. Non-invasive insulin administration has been extensively explored but has not yet been successful[9].

Various strategies have been explored to enhance the skin permeability of biomacromolecules[10], including chemical penetration enhancers that fluidize the SC lipid bilayers, electrical devices that force penetration, ultrasound and jet injection that create transient channels on the skin surface[11–13], and microneedles that pierce the SC into dermal tissues[14,15]. These invasive techniques compromise skin integrity, raising inconvenience, infection and safety concerns.

One may imagine that a skin-permeable material is required to concentrate on the skin surface and then efficiently diffuse through the hydrophobic intercorneocyte lipid matrix of the SC into the hydrophilic viable epidermis[16]. Cationic peptides, which can electrostatically bind to the negatively charged alkyl carboxylic acids in the sebum and SC, have been tested for transdermal delivery, and some have been reported to be skin permeable[17,18]. However, such skin permeation does not act by diffusion through the intercorneocyte lipid matrix because the strong binding immobilizes them in the SC without diffusion, and instead acts through the appendageal paths[19], including hair follicles and sweat glands, and is therefore inefficient in humans because the appendageal areas occupy less than 0.1% of the human skin area[20,21].

Thus, we propose that a polymer capable of transitioning from a polycation that binds to the skin SC surface to a polyzwitterion in the deeper SC layers for free diffusion would be skin permeable. Inspired by the skin's characteristic acidic (pH ≈ 5)-to-neutral pH gradient from

[1]Zhejiang Key Laboratory of Smart Biomaterials and Center for Bionanoengineering, Key Laboratory of Biomass Chemical Engineering of Ministry of Education, State Key Laboratory of Chemical Engineering, College of Chemical and Biological Engineering, Zhejiang University, Hangzhou, China. [2]Institute of Quantitative Biology, Zhejiang Key Laboratory of Cell and Molecular Intelligent Design and Development, College of Life Science, Zhejiang University, Hangzhou, China. [3]Department of Chemical Engineering, Imperial College London, London, UK. [4]These authors contributed equally: Qiuyu Wei, Zhi He, Zifan Li, Zhuxian Zhou. ✉e-mail: xiang_jj@zju.edu.cn; rongjun.chen@imperial.ac.uk; rhzhou@zju.edu.cn; shenyq@zju.edu.cn

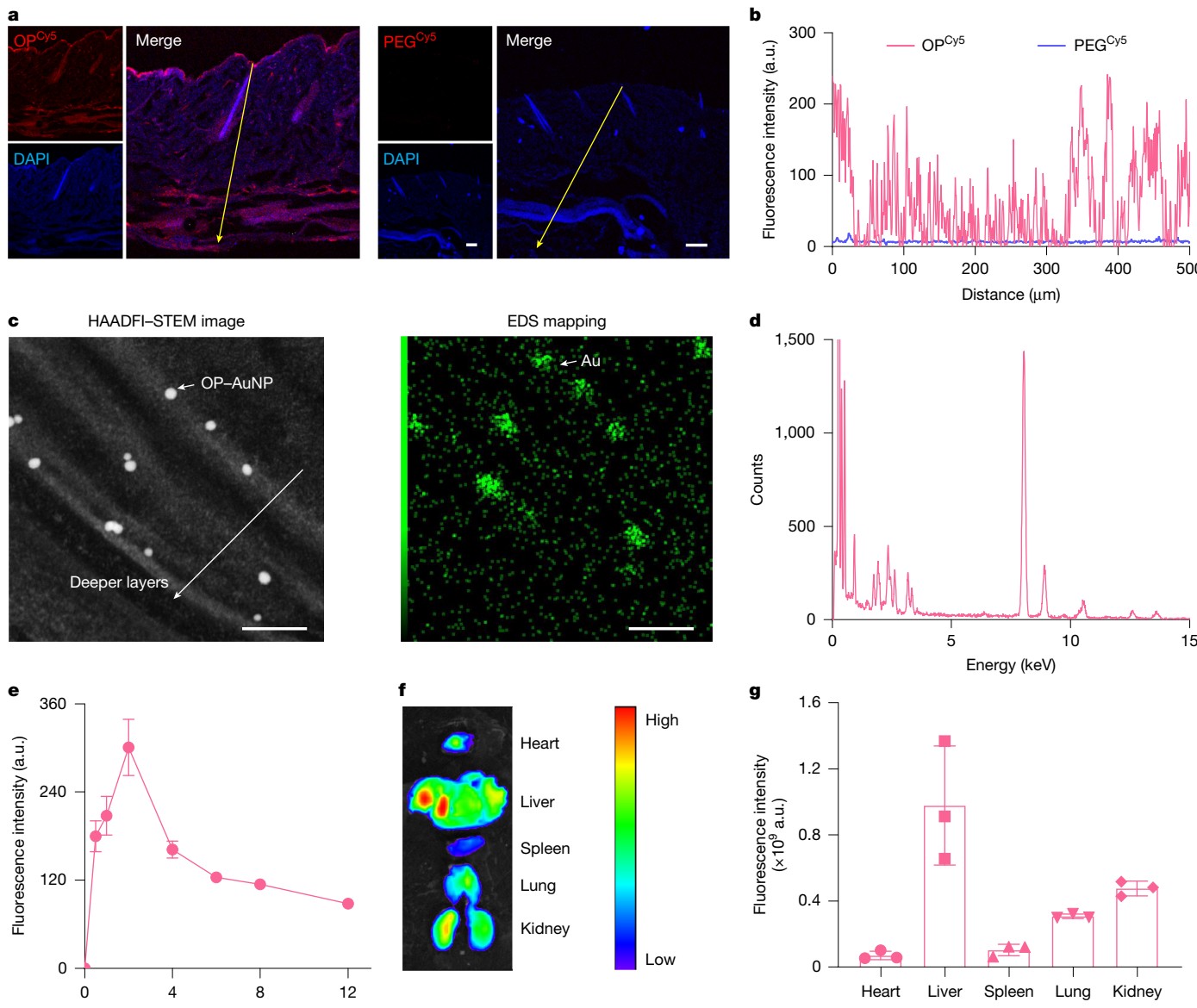

**Fig. 1 | Skin permeability of OP. a**, CLSM images of the section slices of the C57BL/6J mouse dorsal skin after 4 h of post-topical application with OP$^{Cy5}$ or PEG$^{Cy5}$ (Cy5-equivalent dose, 0.2 ml of 10 μg ml$^{-1}$) through a diffusion cell (1.13 cm$^2$); the nuclei were stained with DAPI (blue). The images are representative of $n = 3$ independent experiments. **b**, Cy5 fluorescence intensities plotted from the skin surface to the subcutis along randomly selected lines (yellow arrows in **a**). **c,d**, Transmission electron microscopy characterization of cryosections of the SC layer of the dorsal skin after 4 h of topical application with OP–AuNPs (OP-equivalent dose, 0.2 ml of 0.5 mg ml$^{-1}$; application area, 1.13 cm$^2$). HAADFI–STEM and EDS elemental mapping (**c**) and EDS analysis (**d**) are shown. A large view of the SC region is shown in Supplementary Fig. 4b. The images are representative of $n = 3$ independent experiments. **e**–**g**, Cy5 fluorescence intensity in the blood as a function of the time of topical application with OP$^{Cy5}$ on the mouse dorsal skin (**e**), and ex vivo fluorescence imaging (**f**) and fluorescence intensity quantification (**g**) of the major organs of the mice after 2 h of topical application of OP$^{Cy5}$ (Cy5-equivalent dose, 0.2 ml of 10 μg ml$^{-1}$; application area, 1.13 cm$^2$). Data are mean ± s.d. $n = 3$ mice. Scale bars, 50 μm (**a**) and 50 nm (**c**).

the sebum layer and SC surface to the deeper layers of the SC and viable epidermis[22], we further propose that the highly water-soluble polyzwitterion OP[23] is skin permeable. OP was protonated and positively charged at pH 5 or lower and deprotonated to a polyzwitterion at neutral pH (Supplementary Fig. 1). This alignment of OP's pH-dependent charge transition with the skin pH gradient can render it able to bind to the skin surface and efficiently diffuse through the intercorneocyte lipid matrix; it therefore has high skin permeability (Extended Data Fig. 1).

The molecular mass of OP was controlled by living radical polymerization to approximately 4.5 kDa (Supplementary Fig. 2), and a terminal amine was introduced to label the polymer chain with a fluorescent dye (Supplementary Fig. 3), giving OP$^{Cy5}$. The skin permeability of OP was first assessed through a topical application on the dorsal skin of

male C57BL/6J mice. Four hours after application of OP$^{Cy5}$, Cy5 fluorescence was observed throughout all skin layers in histological sections imaged using confocal laser-scanning microscopy (CLSM) (Fig. 1a,b). By contrast, control PEG$^{Cy5}$ remained confined to the skin surface. The SC penetration of OP was further validated by high-resolution imaging of 5 nm gold nanoparticles tethered to OP chains (OP–AuNPs) (Supplementary Fig. 4a) applied onto the mouse dorsal skin. High-angle annular dark-field imaging–scanning transmission electron microscopy (HAADFI–STEM), energy-dispersive X-ray spectroscopy (EDS) elemental mapping (Fig. 1c and Supplementary Fig. 4b) and EDS analysis (Fig. 1d) revealed the distribution of OP–AuNPs within the intercorneocyte lipid lamella of the SC. These findings confirm that OP does penetrate the SC layers and exhibited exceptional skin permeability.

As a result, the topical OP$^{Cy5}$ percutaneously entered the bloodstream within 0.5 h and reached a maximum concentration at about 2 h after topical application (Fig. 1e). In circulation, OP predominantly accumulated in the liver, followed by the kidneys and lungs (Fig. 1f,g).

The transdermal ability of OP$^{Cy5}$ was further corroborated using porcine skin, which closely resembles human skin in structure, thickness, hair sparseness, and collagen and lipid composition[24]. OP$^{Cy5}$ topically applied onto a minipig abdominal skin distributed across all of the skin layers (Supplementary Fig. 5).

The skin permeability of OP was gauged by the hypoglycaemic effect of its percutaneously delivered insulin. Insulin was conjugated to OP (OP–I) using the strain-promoted alkyne-azide cycloaddition reaction[25] of recombinant human insulin with an azide group at the lysine residual amine and OP with a terminal DBCO group (molecular mass of 4.5 kDa; Supplementary Fig. 6). A PEGylated insulin (PEG–I, PEG molecular mass of 5 kDa) was synthesized similarly and was used as a control[26]. Both conjugates were characterized by reversed-phase high-performance liquid chromatography (RP-HPLC), matrix-assisted laser desorption ionization time-of-flight mass spectrometry (MALDI-TOF MS) and gel-permeation chromatography (Extended Data Fig. 2a–c). The conjugation did not alter the secondary structure of insulin as analysed by circular dichroism (Extended Data Fig. 2d) or its hypoglycaemic efficacy through blood glucose level (BGL) analysis (Extended Data Fig. 2e). For BGL analysis, 5 U kg$^{-1}$ subcutaneous insulin was used, as it produced a faster and more-sustained glucose-lowering effect compared with lower doses (Supplementary Fig. 7). Fluorescently labelled conjugates were also prepared using a similar method and were stable (Supplementary Fig. 8).

Surface plasmon resonance (SPR) analysis on a sensor chip immobilized with the insulin receptor extracellular domains (ECD-IR) or the extracellular domain of IGF1R (ECD-IGF1R) was used to measure the receptor binding kinetics of insulin and OP–I (Extended Data Fig. 3a,b). No substantial difference was observed in their dissociation constants ($K_d$) (26.06 nM for OP–I versus 14.03 nM for insulin). OP–I and insulin also had comparable association rate constants ($k_{on}$), dissociation rate constants ($k_{off}$) and half-lives ($t_{1/2}$) to the immobilized ECD-IR, demonstrating similar binding kinetics (Extended Data Fig. 3c). Similar to insulin, OP–I also showed minimal binding to the extracellular domain of IGF1R—a membrane receptor that is closely related to IR (Extended Data Fig. 3d). The results together indicate that the OP conjugation does not change the affinity and specificity of insulin to its receptor.

All-atom molecular dynamics (MD) simulations illustrated that OP–I stably adsorbed at two major binding sites of its receptor (Extended Data Fig. 3e and Supplementary Video 1). Potential of mean force (PMF) analyses estimated the binding affinities of OP–I with the receptor site 1 and site 2 to be −14.0 and −22.9 kcal mol$^{-1}$, respectively, comparable to those of insulin (−16.7 and −22.0 kcal mol$^{-1}$) (Extended Data Fig. 3f). The simulations confirm that the OP conjugation preserves the binding affinity of insulin to the ECD-IR. Collectively, these findings suggest that OP–I preserves the binding affinity of the insulin receptor and activation of downstream signalling pathways, and therefore its hypoglycaemic effects.

The blood-clearance kinetics of insulin and OP–I were compared after intravenous administration. The blood-clearance curves revealed that the half-life of OP–I (around 15–20 min) was marginally longer than that of native insulin (around 5–10 min) (Supplementary Fig. 9). This extended circulation time is attributed to the zwitterionic nature of the conjugated OP, which reduces interactions with plasma proteins and limits uptake by the reticuloendothelial system[23]. The protein resistance of OP made it unable to hitchhike on plasma proteins[27].

The time-dependent skin permeation of the conjugates was first visualized using CLSM in the three-dimensional (3D) skin-equivalent EpiKutis model, which mimics the reconstructed human epidermis[28]. The fluorescence of OP–I$^{FITC}$ spread throughout the entire epidermal layer within 30 min, whereas PEG–I$^{FITC}$ and insulin$^{FITC}$ gave only weak fluorescence confined to the EpiKutis surface (Fig. 2a). The steady-state flux ($J_{ss}$) of OP–I in EpiKutis was measured to be 0.50 ± 0.12 µg cm$^{-2}$ h$^{-1}$, significantly higher than that of PEG–I (0.10 ± 0.01 µg cm$^{-2}$ h$^{-1}$) and insulin (0.05 ± 0.01 µg cm$^{-2}$ h$^{-1}$). The permeability coefficient ($K_p$) of OP–I was 4.50× and 9.17× higher than that of PEG–I and insulin, respectively (Fig. 2b and Supplementary Table 1).

Sequential $z$-stack imaging of the inner layers of mouse skin using intravital two-photon microscopy demonstrated that OP–I$^{FITC}$ penetrated deeply into the skin tissues and even reached the subcutis after topical application on C57BL/6J mice (Supplementary Fig. 10). By contrast, PEG–I$^{FITC}$ and free insulin$^{FITC}$ gave only sparse and scattered fluorescence within the skin. The skin permeation of OP–I was fast and time dependent. Cy5 fluorescence was already substantial in the subcutaneous adipose tissue labelled with BODIPY after 10 min of topical application of OP–I$^{Cy5}$ on the dorsal skin, and accumulated more over time (Extended Data Fig. 4a,b and Supplementary Fig. 11). By contrast, little PEG–I$^{FITC}$ and insulin$^{FITC}$ were found in the skin (Extended Data Fig. 4c,d).

OP–I$^{FITC}$ was found to localize in all mouse skin compartments, including the epidermis (SC and viable epidermis), dermis, hair follicles and subcutis (Fig. 2c). Extensive OP–I$^{Cy5}$ was around or inside rat lymphatic capillaries (Fig. 2d), suggesting that OP–I enters leaky lymphatic capillaries and subsequently the bloodstream, consistent with the reports that subcutaneously administered large molecules enter systemic circulation through lymphatic uptake[29,30]. Moreover, OP–I$^{Cy5}$ effectively permeated minipig skin after 4 h of topical application, further demonstrating its transdermal delivery potency (Fig. 2e,f).

After topical application of OP–I, the mouse plasma insulin concentration increased rapidly, peaking at around 230 µU ml$^{-1}$ at 1 h after treatment, and then gradually decreased to around 30 µU ml$^{-1}$ after 12 h (Fig. 3a), similar to the blood concentration profile of topical OP (Fig. 1e). Notably, while topical OP–I exhibited a comparable plasma profile to subcutaneously injected insulin within the initial 2 h, its plasma levels remained consistently higher thereafter by 1.6–6 fold. Topically applied insulin or PEG–I did not affect the blood insulin levels (Fig. 3a). Moreover, repeated topical applications of OP–I gave very similar plasma insulin profiles, demonstrating high reproducibility and reliability (Fig. 3b).

Topically applied OP–I$^{Cy5}$ accumulated in key glucose-regulating tissues, including the liver, adipose tissues and skeletal muscles (Supplementary Figs. 12–14). It was taken up by their cells (Supplementary Figs. 15–17) and effectively activated the insulin receptor signalling pathway, as shown by the phosphorylation levels of both the receptor and its downstream protein kinase B (also known as AKT) protein, comparable to those induced by subcutaneous-injected native insulin[31,32] (Supplementary Figs. 18 and 19). Thus, OP–I exerted the functions of insulin in these tissues, including enhancing glucose uptake, promoting glycogenesis and inhibiting gluconeogenesis[33]. As a result, the interplay of skin permeation, tissue distribution and metabolism of OP–I gave its plasma concentration profile (Fig. 3a). By contrast, subcutaneous-injected insulin was rapidly cleared from the bloodstream, with minimal accumulation in adipose and muscle tissues, limiting its ability to sustain hypoglycaemic effects over time.

The ability of topical OP–I to regulate glycaemic levels was evaluated using streptozotocin (STZ)-induced type 1 diabetic mice (Fig. 3c and Supplementary Fig. 20). OP–I, PEG–I or native insulin solution (insulin-equivalent dose, 116 U kg$^{-1}$; 0.2 ml of 0.5 mg ml$^{-1}$) was applied to the dorsal skin (1.13 cm$^2$) of randomly grouped diabetic mice using diffusion cells, with subcutaneous insulin (5 U kg$^{-1}$) injection as a positive control. Subcutaneous-injected insulin rapidly reduced BGLs from around 400 to 100 mg dl$^{-1}$ within 1 h, followed by a rebound to hyperglycaemic levels (around 400 mg dl$^{-1}$) within 4 h. Topically applied insulin and PEG–I had a negligible effect on BGLs. By contrast, topical OP–I demonstrated efficient, dose-dependent hypoglycaemic effects. At a dose of 116 U kg$^{-1}$, OP–I rapidly lowered BGLs to below 200 mg dl$^{-1}$

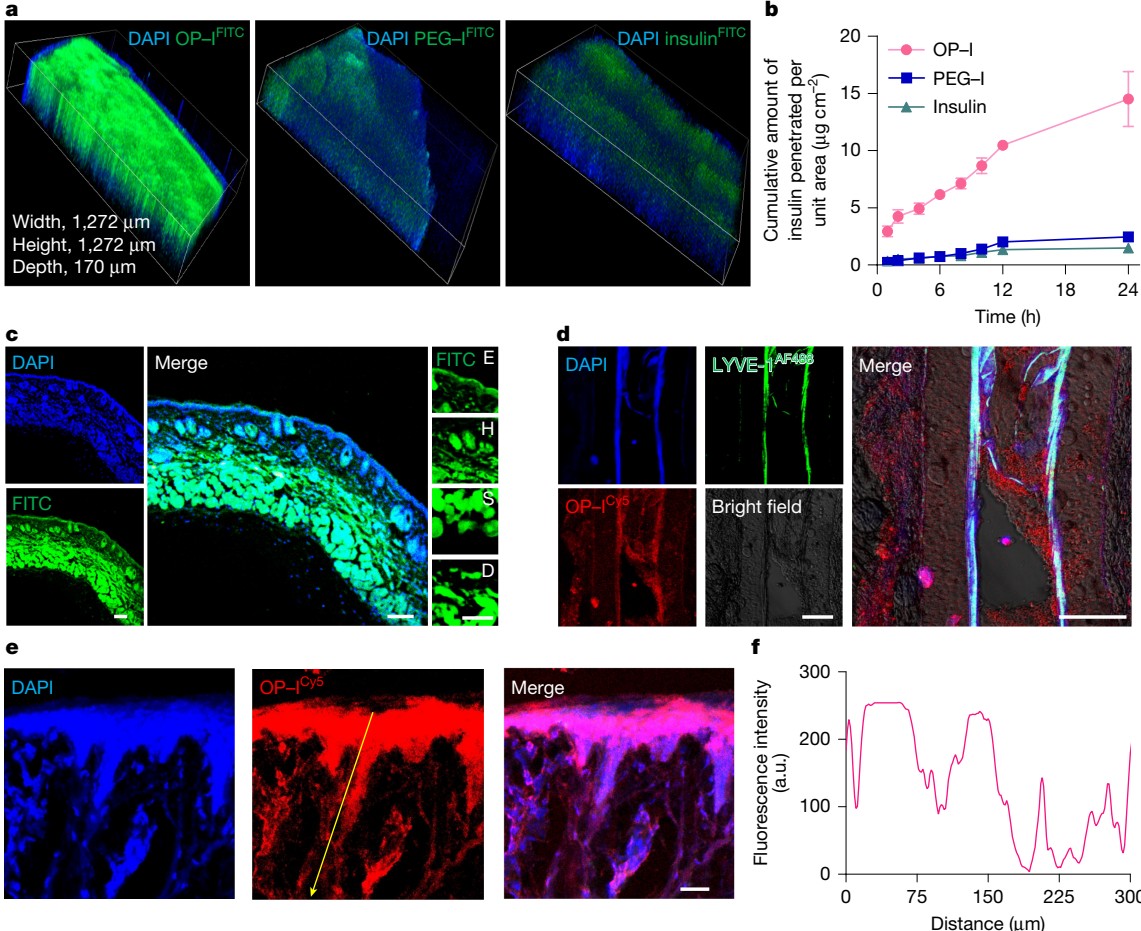

**Fig. 2 | Skin permeability of OP–I. a**, CLSM images of the distribution of OP–I$^{FITC}$, PEG–I$^{FITC}$ or insulin$^{FITC}$ across the in vitro 3D skin equivalent EpiKutis model at 4 h after treatment (FITC-equivalent dose, 0.2 ml of 10 μg ml$^{-1}$; application area, 0.081 cm$^2$; width, 1,272 μm; height, 1,272 μm; depth, 170 μm). **b**, Time-dependent permeation curves of OP–I, PEG–I and insulin across the EpiKutis model (insulin-equivalent dose, 0.2 ml of 0.5 mg ml$^{-1}$; application area, 0.081 cm$^2$). Data are mean ± s.d. $n$ = 3 independent experiments. **c**, CLSM images of the mouse dorsal skin after 4 h of topical application of OP–I$^{FITC}$ (FITC-equivalent dose, 0.2 ml of 10 μg ml$^{-1}$; application area, 1.13 cm$^2$). E, epidermis; H, hair follicle; S, subcutis; D, dermis. The images are representative of $n$ = 3 independent experiments. **d**, CLSM images of immunofluorescence staining of LYVE-1$^{AF488}$ (lymphatic vessel endothelial hyaluronan receptor-1, green) in the subcutaneous tissue of Sprague–Dawley rats after 4 h topical application of OP–I$^{Cy5}$ (red) (Cy5-equivalent dose, 0.2 ml of 10 μg ml$^{-1}$; application area, 1.13 cm$^2$). The nuclei were labelled with DAPI (blue). The images are representative of $n$ = 3 independent experiments. **e**, CLSM images of the slices of the minipig abdominal skin after 4 h of topical application of OP–I$^{Cy5}$-containing cream (Cy5-equivalent dose, 10 ml of 10 μg ml$^{-1}$; application area, 100 cm$^2$). The images are representative of $n$ = 3 independent experiments. **f**, Cy5-fluorescence intensity profile from the skin surface to the subcutis plotted along a randomly selected line (yellow arrow in **e**). Scale bars, 100 μm (**c**,**d**) and 50 μm (**e**).

within 1 h, comparable to the subcutaneous insulin, but maintained normoglycaemic levels (50–200 mg dl$^{-1}$) for 12 h. Lower doses of topical OP–I (58 or 29 U kg$^{-1}$) reduced the BGLs more slowly but still effectively restored BGLs to the normal range (Fig. 3d), highlighting the potential for dose optimization for individual patients. The hypoglycaemic efficacy of OP–I was further validated in healthy mice (Supplementary Fig. 21). Notably, after removal of the diffusion cell, OP–I$^{Cy5}$ was cleared from the skin within 8 h (Supplementary Fig. 22). Importantly, topical application sites did not affect the transdermal delivery efficiency or hypoglycaemic activity of OP–I (Fig. 3e,f).

Intraperitoneal glucose tolerance tests (IPGTTs) further confirmed the glycaemic regulation ability of topical OP–I. Diabetic mice were first treated with topical OP–I for 1 h and then injected intraperitoneally with glucose solution (1.5 g per kg). The mouse BGLs peaked at 330 mg dl$^{-1}$ within 30 min after glucose administration and gradually declined to around 100 mg dl$^{-1}$ within 1 h, maintaining normoglycaemia thereafter. As a reference, healthy mice subjected to the same glucose administration exhibited a similar BGL profile, but their BGLs were above 200 mg dl$^{-1}$ even after 2 h. The BGLs of the

diabetic mice topically applied with insulin or PEG–I increased to around 600 mg dl$^{-1}$ and maintained a hyperglycaemic state after the intraperitoneal glucose administration (Fig. 3g). The area under the curve (AUC) from 0 to 2 h demonstrated that transdermal OP–I provided superior BGL regulation compared with other treatments (Fig. 3h).

OP–I, PEG–I or native insulin was incorporated into a water-in-oil cream and smeared onto the abdominal skin of STZ-induced diabetic minipigs to evaluate their in vivo hypoglycaemic performance (Fig. 3i and Supplementary Fig. 23). The OP–I cream reduced BGLs to normoglycaemic levels within 2 h, reaching a minimum of 100 mg dl$^{-1}$ at 6 h and maintaining normoglycaemia for 12 h. The hypoglycaemic effect of OP–I in minipigs was also dose dependent; lower doses produced a similar hypoglycaemic trend but a reduced BGL-regulating efficiency. Again, topical PEG–I or insulin treatment caused almost no obvious changes in minipig BGLs. These results demonstrate that topical OP–I effectively exerts hypoglycaemic effects in both diabetic mice and minipigs, highlighting its potential as a non-invasive alternative to subcutaneous insulin injections.

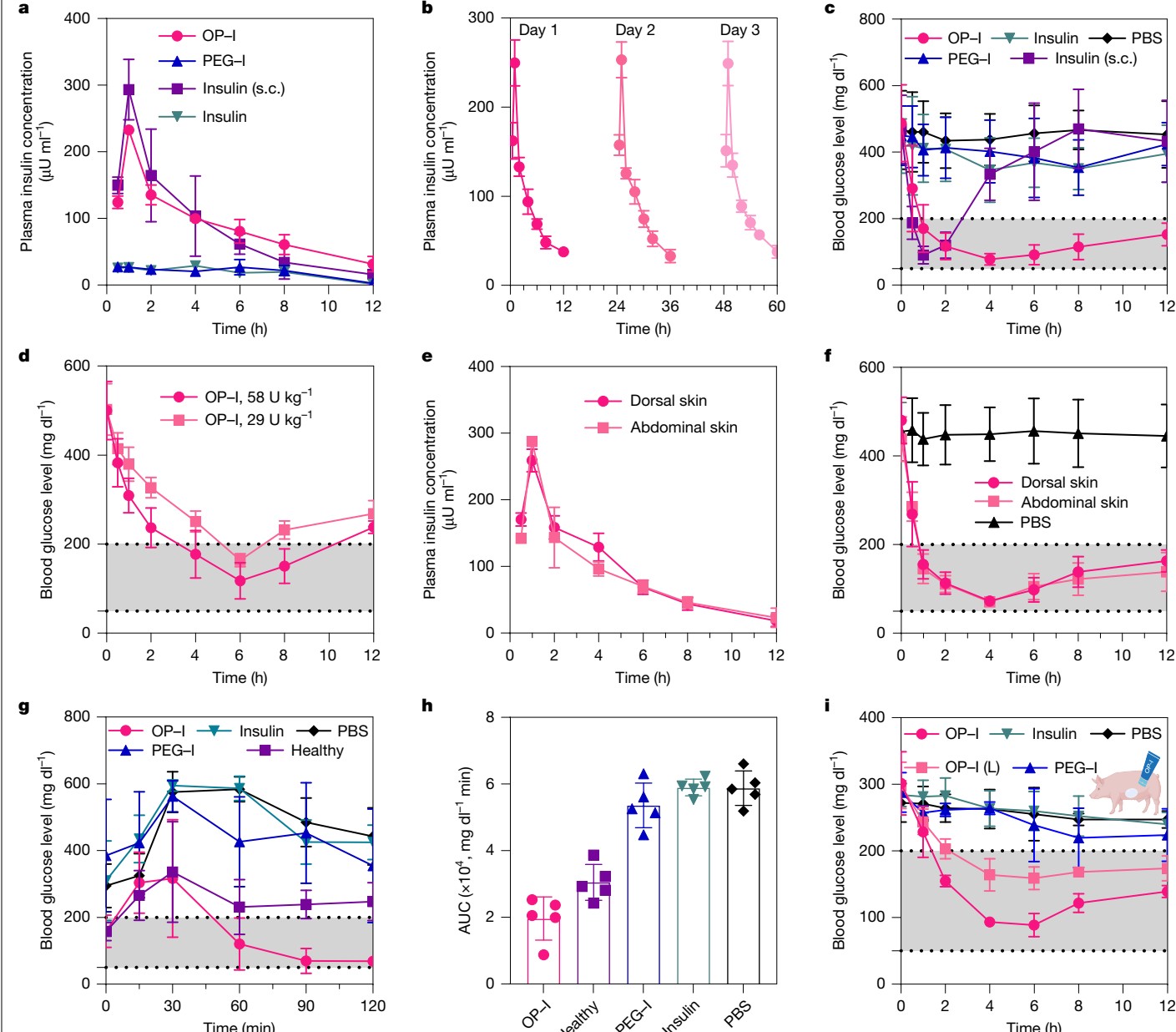

**Fig. 3 | Hypoglycaemic effect of topical OP–I in STZ-induced diabetic mice and minipigs. a**, Plasma insulin concentrations in diabetic mice after topical application of PBS, insulin, PEG–I or OP–I (insulin-equivalent dose, 116 U kg⁻¹; 0.2 ml of 0.5 mg ml⁻¹, 1.13 cm² dorsal skin). $n = 5$ mice. Mice injected subcutaneously with insulin (5 U kg⁻¹) were used as a control. **b**, The plasma insulin concentrations in diabetic mice after three consecutive days of topical application of OP–I as in **a**. $n = 5$ mice. **c**, BGLs of diabetic mice after treatments as in **a**. $n = 8$ mice. **d**, BGLs of the diabetic mice after topical application with lower doses of OP–I (insulin-equivalent dose, 58 or 29 U kg⁻¹; 0.1 ml or 0.05 ml of 0.5 mg ml⁻¹, 1.13 cm² dorsal skin). $n = 8$ mice. **e,f**, The plasma insulin concentrations (**e**; $n = 5$ mice) and BGLs (**f**; $n = 8$ mice) of diabetic mice after topical application of OP–I on the dorsal or abdominal skin as in **a**. The two experiments were performed using separate groups. **g**, IPGTTs in diabetic mice. $n = 5$ mice. Mice received treatments as in **a** and, 1 h later, were injected intraperitoneally with glucose (1.5 g per kg); their BGLs were then measured. Healthy mice were used as controls. **h**, The AUC (0–120 min) in the IPGTT experiment in **g**. $n = 5$ mice. **i**, BGLs in the diabetic minipigs after topical application on the abdominal skin with insulin, PEG–I or OP–I dispersed in water-in-oil cream (insulin-equivalent dose, 29 U kg⁻¹, 1 mg ml⁻¹, 40 ml, 400 cm²; low-dose group (L), 7.25 U kg⁻¹, 1 mg ml⁻¹, 10 ml, 100 cm²); $n = 3$ minipigs. Data are mean ± s.d. For **c**, **d**, **f**, **g** and **i**, the shaded areas outline the normal blood glucose range (50–200 mg dl⁻¹). The diagram in **i** was created using BioRender.

Topical application of OP–I caused no irritation or damage to the mouse skin (Extended Data Fig. 5a). The skin treated with OP–I showed no micromorphological differences, negligible neutrophil infiltration, no changes in thickness (Extended Data Fig. 5b) and no increased cell apoptosis (Extended Data Fig. 5c) compared with the PBS-treated group. Similar observations were found in porcine skin (Extended Data Fig. 5d,e). Furthermore, no adverse effects were observed on blood cell counts, biochemical parameters, or liver and kidney function (Supplementary Table 2).

Next, the mechanism of OP–I penetration through the skin SC layer was investigated. The outer SC layers on tape were peeled from the mouse skin after 4 h of topical application of OP–I^Cy5 using a previously reported method[34]; NBD-C6-HPC was used to stain SC lipids. CLSM revealed a pronounced overlap of Cy5 fluorescence (red) with NBD-C6-HPC staining (green), indicating that OP–I localizes within the SC lipids surrounding corneocytes (Fig. 4a and Extended Data Fig. 6a). Intravital two-photon microscopy further confirmed the presence of OP–I^FITC around corneocytes (Fig. 4b and Supplementary Fig. 24).

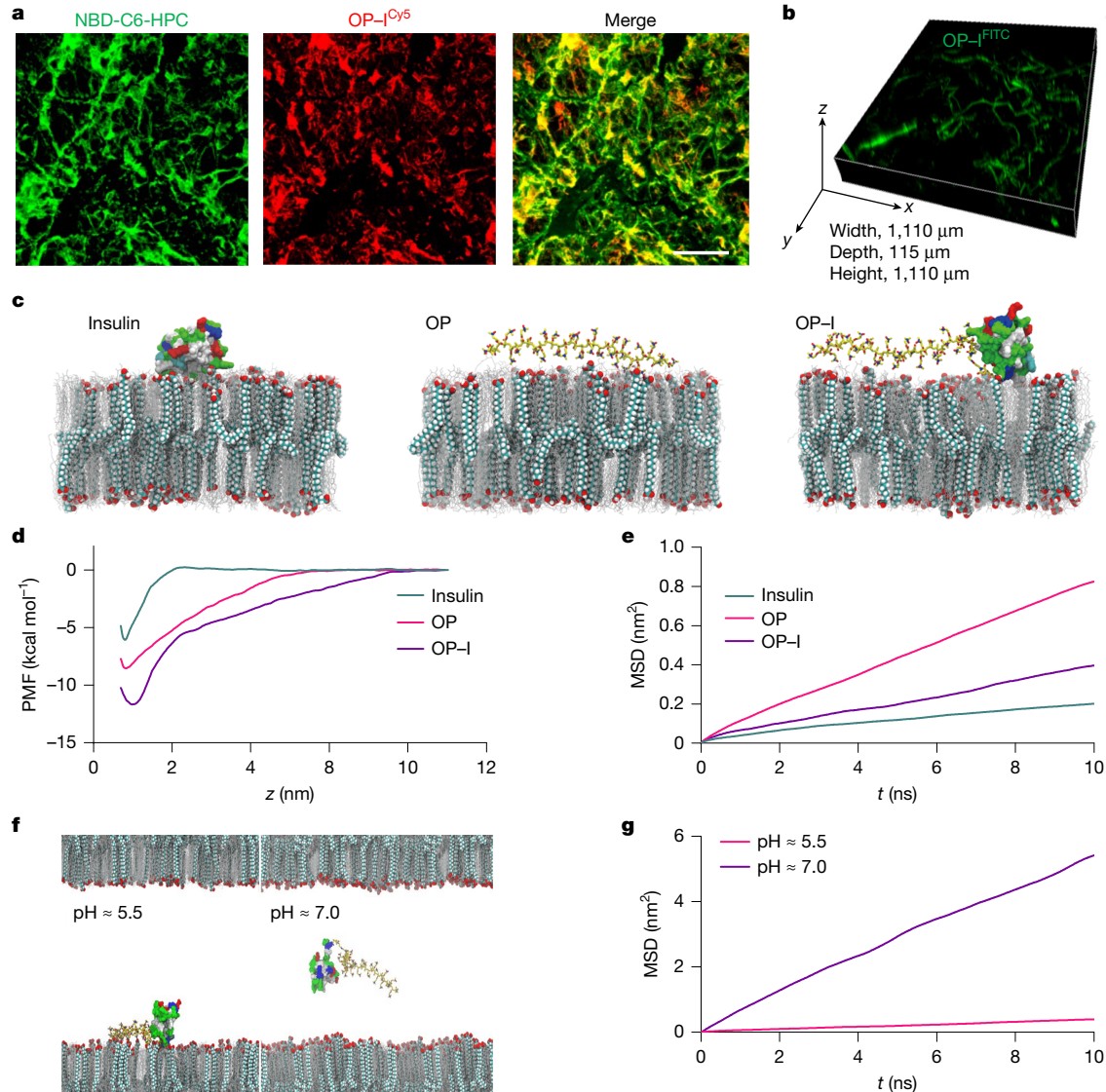

**Fig. 4 | Mechanism study of the SC penetration of OP and OP–I. a**, CLSM images of SC samples from the mouse dorsal skin after 4 h of topical application with OP–I$^{Cy5}$ (red; Cy5-equivalent dose, 0.2 ml of 10 µg ml$^{-1}$, 1.13 cm$^2$). The SC samples were obtained by peeling the skin using adhesive tape, and the SC intercellular lipids were stained with NBD-C6-HPC (green). Additional images are provided in Extended Data Fig. 6a. The images are representative of $n = 3$ independent experiments. **b**, 3D reconstructed view from sequential $z$-stack intravital two-photon microscopy imaging of the mouse skin after 4 h of topical application with OP–I$^{FITC}$ (FITC-equivalent dose, 0.2 ml of 10 µg ml$^{-1}$, 1.13 cm$^2$). Separate images are provided in Supplementary Fig. 24. The images are representative of $n = 3$ independent experiments. **c**, Representative binding modes from MD simulations of insulin, OP and OP–I on SC lipids at pH 5.5. The SC lipid membrane, composed of equimolar ceramide, cholesterol and free fatty acids, was generated using the membrane builder of CHARMM-GUI. See also Supplementary Video 2. **d**, PMF results perpendicular to the SC surface, showing the binding free energies of insulin, OP and OP–I to SC lipids at pH 5.5. $z$ represents the distance between the centre of mass of insulin, OP or OP–I, and the SC lipid surface. Umbrella sampling distance, 9 nm; window resolution, 0.1 nm; sampling time, 35 ns per window; restraint force constant, 1,000 kJ mol$^{-1}$ nm$^{-2}$. **e**, MSD results parallel to the SC surface, comparing the diffusivities of insulin, OP or OP–I on SC lipids at pH 5.5. **f**, Representative interaction modes of OP–I with SC lipids at pH 5.5 and pH 7.0. See also Supplementary Video 4. **g**, MSD results comparing the diffusivities of OP–I on SC lipids under weak acidic (pH 5.5) and neutral (pH 7.0) conditions. Data are from $n = 3$ independent experiments. Scale bar, 50 µm (**a**).

Similar results were observed in the minipig skin, of which the thicker SC layers allowed for easier observation, showing that OP–I$^{Cy5}$ coincided with SC lipids around corneocytes (Extended Data Fig. 6b–d). These findings demonstrate that OP–I accumulates and diffuses through the intercorneocyte lipid matrix, effectively penetrating the SC layers. Moreover, Fourier transform infrared (FTIR) spectroscopy studies revealed that neither OP nor OP–I enhanced the lipid fluidity within the SC (Supplementary Fig. 25). Consistently, mixing OP with Cy5 or insulin$^{Cy5}$ did not enhance their transdermal penetration, and topical application of OP and insulin mixture did not exhibit hypoglycaemic efficacy. By contrast, conjugation with OP (OP$^{Cy5}$, OP–I$^{Cy5}$ or OP–I) granted them efficient skin permeability (Supplementary Figs. 26–28).

These results indicate that OP or OP–I permeates the SC without altering its lipid order or fluidization.

MD simulations were conducted to examine the interaction between OP–I and SC lipids; the model SC lipids were composed of an equimolar mixture of ceramide, cholesterol and fatty acids. The binding process of OP and OP–I to SC lipids under mildly acidic conditions (pH 5.5) is illustrated in Supplementary Video 2, demonstrating that both OP and OP–I adsorbed onto the SC lipids more rapidly than insulin. Figure 4c illustrates the corresponding binding configurations. PMF analyses (Fig. 4d) estimated a binding free energy to SC lipids of −8.5 kcal mol$^{-1}$ for OP, −11.7 kcal mol$^{-1}$ for OP–I and −6.0 kcal mol$^{-1}$ for insulin, indicating stronger binding to the lipid membrane of OP and

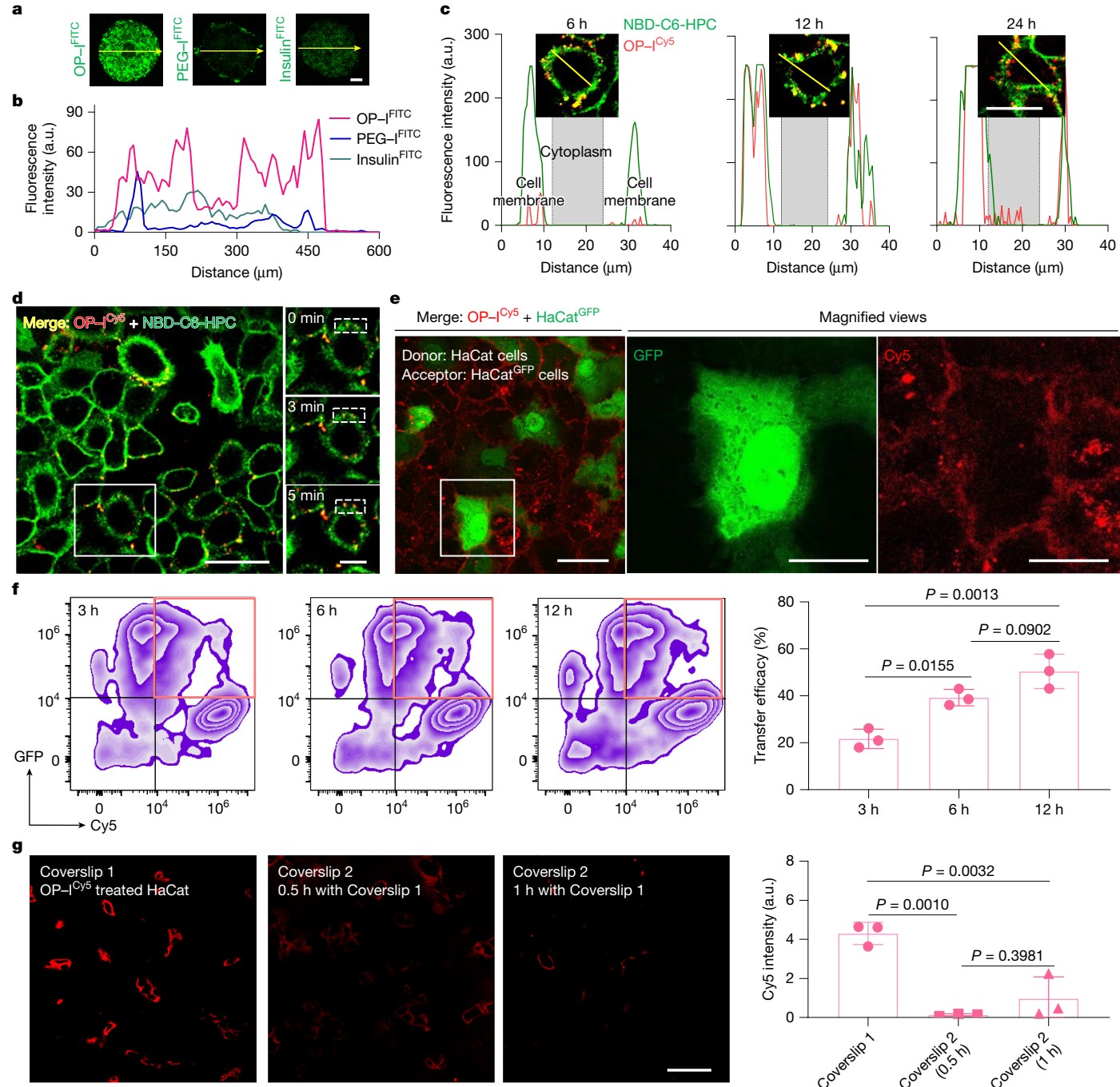

**Fig. 5 | The mechanism of OP–I penetration in the viable epidermis.**
**a,b**, Penetration of OP–I^FITC in HaCat spheroids observed by CLSM (**a**) and line-scan analysis of fluorescence intensity along the yellow arrows in **a** (**b**). Spheroids were incubated with each formulation (FITC-equivalent dose, 1 μg ml⁻¹) for 12 h, and the middle layers of the spheroids were imaged. The images are representative of *n* = 3 independent experiments. **c**, Fluorescence intensities of NBD-C6-HPC (green) and OP–I^Cy5 (red) along the yellow arrows in CLSM images of HaCat cells after 6, 12 or 24 h of incubation with OP–I^Cy5 (Cy5-equivalent dose, 1 μg ml⁻¹). See also the CLSM images in Extended Data Fig. 8a. The images are representative of *n* = 3 independent experiments. **d**, CLSM images of OP–I^Cy5 hopping on HaCat cell membranes. The images show the fixed views from a time-lapse acquisition mode in Supplementary Video 7 (Cy5-equivalent dose, 1 μg ml⁻¹). The images are representative of *n* = 3 independent experiments. **e,f**, Cell-contact-dependent transfer of OP–I^Cy5 from OP–I^Cy5-pretreated HaCat

cells to untreated HaCat^GFP cells. **e**, Representative CLSM images of *n* = 3 independent experiments. **f**, Flow cytometry analysis of transfer efficiency at different timepoints. *n* = 3 independent experiments. Cy5-equivalent dose, 1 μg ml⁻¹. **g**, Cell-contact-dependent transfer of OP–I^Cy5 between HaCat cells on two apposed coverslips. HaCat cells on coverslip 1 were treated with OP–I^Cy5 (Cy5-equivalent dose, 1 μg ml⁻¹) for 12 h, washed and imaged using CLSM. Coverslip 1 was then pressed face-to-face with coverslip 2 seeded with untreated HaCat cells. The paired coverslips were incubated in fresh medium for 0.5 or 1 h at 37 °C, washed and analysed using CLSM imaging (left) and ImageJ fluorescence quantification (right). Data are mean ± s.d. *n* = 3 independent experiments. Significance was determined using one-way analysis of variance for multiple comparisons. Scale bars, 500 μm (**g**), 100 μm (**a**), 50 μm (**d** (left), **e** (left)), 25 μm (**c**) and 20 μm (**d** (right) and **e** (right)).

OP–I at the acidic pH. PMF analyses along the surface of SC lipids (in the *x* direction) estimated diffusion energy barriers to be $0.9 \pm 0.1$ kcal mol$^{-1}$ for OP, $1.7 \pm 0.2$ kcal mol$^{-1}$ for OP–I and $2.2 \pm 0.3$ kcal mol$^{-1}$ for insulin (Extended Data Fig. 7a). Accordingly, the friction coefficient ($\gamma$) derived from the Stokes–Einstein relation was $2.07 \times 10^{-10}$ Ns m$^{-1}$ for OP, $4.31 \times 10^{-10}$ Ns m$^{-1}$ for OP–I and $8.49 \times 10^{-10}$ Ns m$^{-1}$ for insulin. The lower friction coefficients of OP and OP–I with SC lipids suggest that they have reduced 'local trapping' by SC lipids and faster diffusion compared with insulin (Extended Data Fig. 7b). This observation was further supported by Supplementary Video 3 and Extended Data Fig. 7c, as well as the mean squared displacement (MSD) calculations shown in Fig. 4e. MD simulations also compared the OP–I interactions with SC lipids at weakly acidic (pH 5.5) and neutral (pH 7.0) conditions (Fig. 4f and Supplementary Video 4). At pH 7.0, OP transitioned to a zwitterionic state (Supplementary Fig. 1), substantially reducing its electrostatic interactions with SC lipids and allowing OP and OP–I to diffuse freely (Fig. 4g). Notably, insulin was negatively charged while OP was slightly positively charged at pH 5.3–7.4. MD simulations demonstrated that, at pH 6.0, insulin and OP did not bind to form aggregates owing to the low charge density of insulin and the strong hydrophilicity of OP (Supplementary Fig. 29 and Supplementary Video 5).

As a result, once OP or OP–I was applied to the skin, the acidic surface protonated OP, enabling both OP and OP–I to bind to SC lipids and become enriched within the SC layers. The reduced acidity beneath the upper SC layers freed this binding, enabling OP and OP–I to diffuse through the intercorneocyte lipids and penetrate deeper into the viable epidermis (Extended Data Fig. 1).

After entering the viable epidermis of the skin from the SC, OP–I was transported either through transcytosis pathways[23] or through intercellular spaces. This process was first investigated using 3D-cultured spheroids of the human immortal keratinocyte line (HaCat)[35]. The spheroids treated with OP–I$^{FITC}$ for 12 h exhibited intense fluorescence, whereas those incubated with PEG–I$^{FITC}$ or insulin$^{FITC}$ had only weak fluorescence (Fig. 5a,b). Pretreatment with endocytosis or exocytosis inhibitors did not suppress OP–I$^{FITC}$ infiltration in the HaCat spheroids (Supplementary Fig. 30). Notably, even after 24 h of incubation, OP–I$^{Cy5}$ still exclusively localized on cell membranes, with no detectable fluorescence in the cytoplasm (Fig. 5c and Extended Data Fig. 8a). Similarly, CLSM imaging of porcine skin slices topically treated with OP–I$^{Cy5}$ revealed that OP–I$^{Cy5}$ was predominantly distributed on cell membranes rather than inside cells (Extended Data Fig. 8b,c).

Total internal reflection fluorescence microscopy (TIRFM) was used to study the movement patterns of OP–I on cell membranes using membrane-labelled (NBD-C6-HPC) cells (green) cultured with OP–I$^{Cy3}$ (Supplementary Video 6). The active movement of red fluorescent dots on cell membranes indicated that OP–I$^{Cy5}$ hopped between adjacent cells (Fig. 5d and Supplementary Video 7). An experiment observing the direct cell-to-cell transfer of OP–I$^{Cy5}$ was performed by mixing the OP–I$^{Cy5}$-pretreated HaCat cells (donor cells) with fresh HaCat cells stably expressing green fluorescent protein (HaCat$^{GFP}$, acceptor cells) (Supplementary Fig. 31). CLSM imaging and flow cytometry analysis showed that HaCat$^{GFP}$ cells gradually acquired Cy5 fluorescence from the donor cells (Fig. 5e,f). The transfer efficiency was enhanced by prolonging the contact time between donor and acceptor cells (Fig. 5f) or by increasing the densities of both cell populations (Extended Data Fig. 8d). Placing coverslip 1 (OP–I$^{Cy5}$-treated HaCat cells) and coverslip 2 (untreated HaCat cells) in face-to-face contact also facilitated the transfer of OP–I$^{Cy5}$ (Fig. 5g). On the other hand, the transfer of OP–I$^{Cy5}$ was completely inhibited when the two cell populations were separated even in the same culture medium (Supplementary Fig. 32). Together, these observations confirm the cell-contact-dependent transfer of OP–I.

Thus, OP–I transport in the epidermis or dermis did not involve intracellular processes[36], and instead occurred through an intercellular process using membrane-mediated diffusion.

The synergy between the characteristic acidic-to-neutral pH gradient of the skin and the pH-dependent binding of OP with SC lipids underpins the fast skin penetration of OP and OP–I. OP was zwitterionic at pH > 7 but gradually protonated to be positively charged as the pH decreased (Supplementary Fig. 1). At the acidic surface of the skin (pH of around 5), cationic OP bound strongly to and thus was enriched in the SC and intercorneocyte lipids, which contain about 15% fatty acids[37]. As the pH gradually increased to around 7 in deeper SC layers, OP transitioned to a neutral state, losing its affinity for SC lipids and diffusing freely through the para-corneocyte spaces into the viable epidermis (Extended Data Fig. 1). In the viable epidermis and dermis, OP and OP–I did not enter cells but hopped on the cell membranes, which avoided intracellular degradation and facilitated efficient skin permeation. OP and OP–I in the dermis were then drained into leaky lymphatic capillaries and subsequently entered systemic circulation, similar to other large molecules[29,30].

The efficient skin permeation of OP–I enabled sufficient plasma insulin delivery (Fig. 3) and accumulation in key glucose-regulating tissues, particularly the liver, adipose and muscles, producing immediate, robust and prolonged hypoglycaemic effects, as well as IPGTT outcomes comparable to, or even superior to, subcutaneous-injected insulin in diabetic mice and minipigs.

The skin permeation of OP and OP–I was entirely non-invasive and non-irritative. Repeated topical application caused no structural changes to SC microstructures, no corneocyte shedding, no widening of intercellular gaps and no signs of inflammation or cell death (Extended Data Fig. 5). Thus, OP represents a promising non-invasive transdermal insulin delivery system, offering an ideal alternative to hypodermic injections for diabetes management[38].

In summary, we present a non-invasive transdermal insulin delivery system that achieves in vivo hypoglycaemic efficacy comparable to subcutaneous injections for diabetes treatment, resulting from the efficient skin permeation of OP. The OP conjugation is versatile for transdermal delivery of biomacromolecules such as peptides, proteins and nucleic acids, with broad therapeutic applications, warranting further investigation in future studies.

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

## Methods

### Cell lines and animals

The HaCat, mouse hepatoma (AML-12), mouse embryonic fibroblast (3T3-L1) and mouse skeletal muscle (MSMC) cell lines were obtained from the Cell Bank of the Chinese Academy of Sciences. The cells expressing green fluorescent protein (HaCat$^{GFP}$ cells) were established by lentivirus transfection of GFP plasmids into HaCat cells according to the manufacturer's protocol (Shanghai Genechem). All of the cell lines were incubated in a nutritious DMEM medium containing 10% FBS and 1% (v/v) penicillin–streptomycin at 37 °C with 5% $CO_2$.

Male C57BL/6J mice (aged 6–8 weeks, 25 g) and female Sprague–Dawley (SD) rats (aged 8–12 weeks, 200 g) were purchased from Shanghai SLAC Laboratory Animal. The mice and rats were housed in the Laboratory Animal Center of Zhejiang University under specific-pathogen-free conditions. Guangxi Bama-minipigs (male, aged 6 months, 35–40 kg) were purchased from Shanghai Jiagan Laboratory Animal and housed in the Laboratory Animal Center of Zhejiang University. The animals were fed with a standard diet and maintained under a 12 h–12 h light–dark cycle, with free access to water throughout the experiment unless otherwise specified. The ambient environment was controlled at 20–26 °C and 50–70% relative humidity. All of the animal experiments were carried out according to the protocols approved by the Institutional Animal Care and Use Committee of Zhejiang University (ZJU20250230).

### Experimental materials

Unless otherwise indicated, all materials were purchased from Sinopharm Chemical Reagent. Dichloromethane ($CH_2Cl_2$) and tetrahydrofuran (THF) were distilled over calcium hydride ($CaH_2$) or treated with a 4 Å molecular sieve. Trifluoroacetic acid (TFA), 2,2′-azobis(2-methylpropionitrile) (AIBN), 2-(dimethylamino) ethyl methacrylate (DMA), *N*-Boc-ethylenediamine, *N,N*-diisopropylethylamine (DIPEA), 1-(3-dimethylaminopropyl)-3-ethylcarbodiimide hydrochloride (EDC·HCl) and 1-hydroxybenzotriazole (HOBT) were purchased from Energy Chemical. Fluorescein isothiocyanate (FITC), sulfo-cyanine3 succinimidyl ester (Sulfo-Cy3-NHS) and sulfo-cyanine5 succinimidyl ester (Sulfo-Cy5-NHS) were purchased from Lumiprobe. Human recombinant insulin (29 U mg$^{-1}$, I8830) was purchased from Solarbio Science & Technology. NBD-C6-HPC was purchased from J&K Scientific. 4-Cyano-4-[[(dodecylthio)carbonothioyl]thio]pentanoic acid (CTA) was purchased from Aladdin. 2-Iminothiolane hydrochloride and STZ were purchased from Macklin. Gold nanoparticles (AuNPs, 5 nm) were purchased from Feynman Biotechnology. *N*-Succinimidyl 15-azido-4,7,10,13-tetraoxapentadecanoate (NHS-PEG$_4$-N$_3$) was purchased from Aikon. *N*-succinimidyl 4-[(5-aza-3,4:7,8-dibenzocyclooct-1-yne)-5-yl]-4-oxobutyrate (DBCO-NHS) was purchased from New Research Biosciences.

### Synthesis of *N*-[2-(*N*-tert-butoxycarbonylamino)]ethyl-4-(dodecyltrithio-carbonate)-4-cyanopenteramide (*N*-Boc-CTA)

CTA (2 mmol), EDC·HCl (2.5 mmol) and HOBT (2.5 mmol) were dissolved in 20 ml dry $CH_2Cl_2$ and stirred at room temperature for 4 h. *N*-Boc-ethylenediamine (2.5 mmol) and DIPEA (2.5 mmol) were dissolved in $CH_2Cl_2$ (10 ml) and added dropwise to the above solution to continue the reaction overnight at room temperature. The mixture was successively washed twice with a saturated solution of $Na_2CO_3$, a 0.1 M HCl solution and a saturated brine solution, and then dried over $MgSO_4$. The crude product was passed through a column packed with silica gel using a mobile phase of *n*-hexane and ethyl acetate mixture (1:1). A yellow solid was obtained.

### Synthesis of Boc-amino-terminated poly[2-(*N,N*-dimethylamino) ethyl methacrylate] (*N*-Boc-PDMA)

DMA (30 mmol), *N*-Boc-CTA (0.35 mmol) and AIBN (0.07 mmol) were dissolved in THF (30 ml) in a Schlenk flask and bubbled with dry $N_2$ for 20 min. The reaction was carried out at 65 °C for 12 h. After terminating the polymerization by opening the flask, the solution was concentrated and poured into cold *n*-hexane. The precipitated *N*-Boc-PDMA was isolated and then dried under a vacuum.

### Synthesis of Boc-amino-terminated poly[2-(*N*-oxide-*N,N*-dimethylamino)ethyl methacrylate) (*N*-Boc-OP)

*N*-Boc-PDMA (0.5 g) was dissolved in 5 ml of 30% hydrogen peroxide ($H_2O_2$) solution. The mixture was stirred at room temperature for 4 h and then dialysed against deionized water to remove the unreacted $H_2O_2$ completely. *N*-Boc-OP was obtained after lyophilization.

### Synthesis of OP-NH$_2$

*N*-Boc-OP (500 mg) dissolved in 5 ml $CH_2Cl_2$ and 5 ml TFA was added dropwise under ice cooling. This solution was stirred for 2 h at room temperature. The reaction solution was evaporated to remove TFA, vacuum-dried and then dissolved in deionized water. The pH of the solution was adjusted to 7.4 with sodium hydroxide solution (1 M) and then dialysed against deionized water. OP-NH$_2$ was obtained after lyophilization.

### Synthesis of OP-DBCO

OP-NH$_2$ (30 mg) was dissolved in 3 ml of PBS (pH 7.4), and DBCO-NHS (3 mg) in 2 ml of DMF was added. The mixture was stirred at room temperature for 4 h and then dialysed sequentially against a water-DMF mixture (3:2 (v/v)) and water. The product, OP-DBCO, was obtained by lyophilization. PEG-DBCO was prepared according to the same procedure.

### Synthesis of OP–I

The lysine residue on insulin was introduced with an azide group using N$_3$-PEG$_4$-NHS. Insulin (21.6 mg) was dissolved in 5 ml of 0.1 M Na$_2$CO$_3$ (adjusted to pH 10), and N$_3$-PEG$_4$-NHS (1.44 mg, 10 mg ml$^{-1}$ in DMSO) was added at an azido-to-amine ratio of 1:3. The mixture was stirred at room temperature for 4 h, and the azide-modified insulin (insulin-N$_3$) was purified using preparative HPLC (prep-HPLC). For OP–I synthesis, insulin-N$_3$ (2 mg, 0.29 µmol) and OP-DBCO (3.9 mg, 0.88 µmol) were dissolved in PBS (pH 7.4) and stirred at room temperature for 4 h. The product, OP–I, was purified using prep-HPLC. PEG–I was synthesized similarly.

### Labelling OP, OP–I, PEG–I and insulin with FITC, Cy3 or Cy5

FITC, Sulfo-Cy3-NHS or Sulfo-Cy5-NHS solutions in DMSO (5 mg ml$^{-1}$) were added dropwise with gentle stirring to OP-NH$_2$, PEG-NH$_2$, insulin or their conjugates (OP–I or PEG–I) dissolved in PBS (10 mg ml$^{-1}$, pH 7.4) at a dye-to-insulin molar ratio of 1:1. The reaction was carried out overnight at room temperature in the dark. The labelled polymers, insulin and conjugates were purified using Sephadex G-25 resin to remove the unreacted dye and then lyophilized. The products were stored in the dark at 4 °C for further use.

### RP-HPLC analysis

The RP-HPLC analysis was performed using a 1260 binary HPLC pump equipped with a ZORBAX SB-C18 column (5 µm, 4.6 × 250 mm) and a 1260 infinity II variable wavelength detector set at 280 nm. The mobile phase consisted of water with 0.1% TFA (phase A) and acetonitrile with 0.1% TFA (phase B). The flow rate was set at 1 ml min$^{-1}$. The elution gradient was set as follows: 30% to 40% B, 0–5 min; constant 40% B, 5–10 min; 40% to 100% B, 10–11 min; constant 100% B, 11–16 min; 100% to 30% B, 16–17 min; constant 30% B, 17–22 min.

### Stability of OP–I$^{Cy5}$

OP–I$^{Cy5}$ (0.4 mg) was incubated in 200 µl of DMEM culture medium with 10% FBS at 37 °C. The solution was sampled at 0 h, 6 h and 12 h for RP-HPLC detection (*n* = 3 per timepoint), as described above, except that

an Agilent 1260 Infinity II fluorescence detector (640 nm excitation, 660 nm emission) was used. The mobile phase remained the same, but the elution gradient was as follows: 0%–100% B, 0–15 min; isocratic at 100% B, 15–20 min; 100%–0% B, 20–21 min; and isocratic at 0% B, 21–26 min. OP–I$^{Cy5}$ had a retention time of 8.8 min. The target peak integral area was quantified, confirming its structural stability even after 12 h.

## Prep-HPLC for purification

Prep-HPLC was performed using the Waters Prep 150 LC system equipped with a Pursuit 5 C18 column (250 × 21.2 mm) and a Waters 2489 UV/VIS detector set at 214 nm and 280 nm. The mobile phases included water with 0.1% TFA (phase A) and acetonitrile with 0.1% TFA (phase B). The elution gradient and flow rates were set as follows: 20% to 40% B with flow rates increasing from 10 to 15 ml min$^{-1}$, 0–10 min; constant 40% B with a constant flow rate of 15 ml min$^{-1}$, 10–20 min. The retention times were 11.8 min for insulin-N$_3$, 10.1 min for OP–I and 12.8 min for PEG–I.

## Gel-permeation chromatography

The Shimadzu Prominence Plus LC-20AD LC system was equipped with two columns connected in series (PL aquagel-OH MIXED-H and PL aquagel-OH 30), a refractive index detector and a UV/VIS detector. The mobile phase was prepared by mixing acetic acid (100 ml), acetonitrile (150 ml) and deionized water (200 ml). The pH was then adjusted to 2.3 using a concentrated ammonia solution, and the final volume was brought to 500 ml with deionized water. The flow rate was 0.5 ml min$^{-1}$, and the column temperature was 40 °C.

## Circular dichroism spectroscopy

Far UV circular dichroism spectra were recorded at 37 °C on the JASCO J-815 spectropolarimeter. Quartz cuvettes with a path length of 1 mm were used. Each spectrum was an average of four scans recorded from 260 to 200 nm at 1 nm steps.

## MALDI-TOF MS analysis

All MALDI-TOF MS analyses were performed on a Bruker Autoflex maX TOF/TOF mass spectrometer (Bruker), equipped with a modified Nd:YAG laser in positive-ion mode; data acquisition was conducted using the Bruker flexControl 3.4 software. α-Cyano-4-hydroxycinnamic acid was used as the matrix. As OP did not generate detectable signals in mass spectra owing to its zwitterionic $N$-oxide structure, the OP–I was then reduced to PDMA-I by bis(pinacolato)diboron[39] for analysis. Ions were extracted into the mass spectrometer in reflection mode using an extraction potential of 20 kV using a high-mass detection method.

## Zeta potential measurements

The solutions were prepared by dissolving OP (0.1 mg ml$^{-1}$) or OP–I (0.04 mg ml$^{-1}$) in HEPES buffers at varied pH values (10 mM). A Nano ZS Zetasizer (Malvern Instruments) was used to measure the zeta potentials using a 4 mW 633 nm He-Ne laser at 25 °C.

## Synthesis of OP–AuNPs

OP-NH$_2$ (60 mg) was dissolved in 6 ml of PBS (pH 8.0), and 2-iminothiolane hydrochloride (Traut's reagent, 10 mg ml$^{-1}$) was added. The mixture was stirred at room temperature for 4 h and dialysed against water (MWCO 1 kDa) to remove the unreacted components. The product was lyophilized to obtain OP-SH. OP-SH (1 mg) was added to a solution of AuNPs (50 µg ml$^{-1}$) and stirred at 4 °C for 12 h. The resulting OP–AuNPs were purified by centrifugation at 19,000$g$ for 10 min at 4 °C and then resuspended in water.

## In vivo skin permeation of OP$^{Cy5}$, OP–I$^{FITC}$, OP–I$^{Cy5}$ and their mixtures

The dorsal skin of male C57BL/6J mice was exposed using depilatory cream and cleaned using PBS. A diffusing cell of 1.13 cm$^2$ was attached onto the dorsal skin. OP$^{Cy5}$, OP–I$^{FITC}$, PEG–I$^{FITC}$, insulin$^{FITC}$, OP–I$^{Cy5}$ (FITC or Cy5-equivalent dose, 10 µg ml$^{-1}$; 0.2 ml per mouse) or their physical mixture was injected into the cell. After timed topical administration, the mice were euthanized. The treated skin sites were washed and dissected. The entire skin samples were carefully washed three times with PBS, fixed with 4% paraformaldehyde (PFA) solution and sectioned into 10-µm-thick slices using a cryostat (UV800, Leica Microsystems). 4′,6-Diamidino-2′-phenylindole (DAPI) was used to stain and label the nuclei of skin tissues, and BODIPY was used to stain and localize subcutaneous fat deposits. Fluorescence images were taken using CLSM with excitation at 405 nm for DAPI, 488 nm for BODIPY or FITC, and 640 nm for OP–I$^{Cy5}$. The CLSM images were analysed using ImageJ.

OP$^{Cy5}$ solution (0.2 ml, Cy5-equivalent dose, 10 µg ml$^{-1}$) was applied to the mice as described above. Blood samples (200 µl) were collected from the orbit venous plexus at timed intervals, and their Cy5 fluorescence intensities were measured using a microplate spectrophotometer with excitation at 640 nm. The main organs of the mice, including the heart, liver, spleen, lung and kidneys, were also collected at timed topical administration of OP$^{Cy5}$, and their fluorescence intensities were measured using the IVIS Spectrum imaging system (IVIS Lumina XRMS Series III, PerkinElmer).

OP$^{Cy5}$ was dispersed in a cream (water-in-oil, Aquaphor; Cy5-equivalent dose, 10 µg ml$^{-1}$). The OP$^{Cy5}$ cream was topically applied on the minipig abdominal skin (cream volume, 10 ml; application area, 100 cm$^2$) for 4 h. The minipigs were euthanized, and the treated skin sites were dissected and sectioned into slices, as mentioned above. Fluorescence images were taken using CLSM with excitation at 405 nm for DAPI and 640 nm for OP–I$^{Cy5}$. The CLSM images were analysed using ImageJ.

## Biological half-life of OP–I

Male C57BL/6J mice were randomly assigned to groups ($n$ = 3 for each group). The solution (0.1 ml; Cy5-equivalent dose, 0.1 mg per kg) of insulin$^{Cy5}$ or OP–I$^{Cy5}$ was injected through the tail vein. Blood samples were collected from the orbital venous plexus at timed intervals. The fluorescence intensity was quantified using the IVIS Spectrum system. The insulin concentration was also measured using human insulin ELISA kits (Elabscience). The half-lives of insulin and OP–I were calculated using DAS2 software based on the plasma concentration data.

## HAADFI–STEM-EDS analysis of OP–AuNP-treated skin cryosections

C57BL/6J male mice were topically administered with 0.2 ml OP–AuNPs (OP-equivalent dose, 0.5 mg ml$^{-1}$). After 4 h, the mice were anaesthetized. The skin at the treated site was dissected and fixed overnight at 4 °C. The samples were trimmed into small sections and then treated with 1% osmium tetroxide (diluted in 100 mM cacodylate buffer) for 2 h, dehydrated with ethanol and acetone, and embedded in Spurr resin. Then, 10-µm-thick slices were sectioned (LKB 11800 Pyramitome) and stained with toluidine blue to select the desired regions. The ultrathin sections (50–100 nm) were prepared using an ultramicrotome (UC7, Leica).

HAADFI STEM-EDS analysis of the skin cryosections was performed on a field emission transmission electron microscope (JEOL JEM F200) at an accelerating voltage of 200 keV equipped with an EDS detector.

## Cellular uptake

AML-12 cells, 3T3-L1 differentiated adipocytes and MSMC cells were seeded into confocal dishes at a density of 1 × 10$^5$ cells per ml and incubated for 24 h. The cells were treated with OP–I$^{Cy5}$ (Cy5-equivalent dose, 1 µg ml$^{-1}$) for 4 h. After incubation, the nuclei were stained with Hoechst 33342 (2 µM) for 20 min. The cells were washed twice with PBS and imaged using CLSM at an excitation wavelength of 405 nm for Hoechst 33342 and 640 nm for Cy5.

## SPR analysis of binding affinity

SPR experiments were performed on a Biacore X100 instrument (Cytiva) with data acquisition using Biacore X100 system control software (v.2.0.1.201). The CM5 chip sensor (GE Healthcare) was used in this study. Anti-His antibodies were immobilized on the surface of the CM5 chip sensor according to the manufacturer's instructions. His-tag-ECD-IR/IGF1R protein (Sino Biological) was then injected over the anti-His antibody-coated surface of the CM5 chip sensor at a specific concentration and flow rate. The binding affinities of insulin and OP–I were evaluated by injecting various concentrations of insulin (9.75, 19.5, 39.0, 78.0 and 156 nM) or OP–I (4.06, 8.13, 16.3, 32.5 and 65.0 nM) in HEPES running buffer (pH 7.4) over the His-tag-ECD-IR/IGF1R-coated surface under a single-cycle kinetics mode. Each cycle included a 180 s binding phase and a 300 s dissociation phase. The dissociation equilibrium constant ($K_d$), association rate constant ($k_{on}$) and dissociation rate constant ($k_{off}$) were determined using the BIA evaluation software (v.2.0.1.201). The $t_{1/2}$ values, which define the residence time, were determined using the formula $\ln 2/k_{off}$.

## Western blotting for insulin signalling analysis

Mice were injected subcutaneously with PBS or insulin (insulin-equivalent dose, 5 U kg$^{-1}$), or topical administration with OP–I (insulin-equivalent dose, 116 U kg$^{-1}$; concentration, 0.5 mg ml$^{-1}$; volume, 0.2 ml; diffusing area, 1.13 cm$^2$). The mice were euthanized, and their skeletal muscle tissues were collected at 1 h after subcutaneous injection or 4 h after topical administration. Muscle samples were weighed and lysed in radioimmunoprecipitation buffer (Sigma-Aldrich) containing protease and phosphatase inhibitors (Thermo Fisher Scientific) and incubated on ice for 30 min. The lysates were centrifuged at 20,000$g$ for 20 min at 4 °C and quantified using the BCA protein assay. The proteins were resuspended in Laemmli sample buffer with 2.5% 2-mercaptoethanol, denatured at 95 °C for 5 min, separated by SDS–PAGE and transferred to polyvinylidene fluoride membranes. The membranes were blocked in Tris-buffered saline with 0.5% Tween-20 (TBST) and 5% BSA for 1 h, followed by overnight incubation at 4 °C with primary antibodies against phosphorylated IRβ/IGF1Rβ (1:1,000), phosphorylated AKT-Thr308 (1:1,000) or GAPDH (1:5,000). After washing three times with TBST, the membranes were incubated with horseradish-peroxidase-conjugated secondary antibodies (HRP-labelled goat anti-rabbit IgG (H+L) or HRP-labelled Goat Anti-Mouse IgG (H+L)) at a dilution of 1:1,000 at room temperature for 1 h. Signal detection was performed using Immobilon Forte chemiluminescence substrate on the ChemiScope 3600 Mini Imaging System (Clinx Science Instruments). The membranes were next incubated in stripping buffer (BL526, Biosharp) at room temperature with gentle agitation for 15 min, followed by extensive washing with TBST, and then overnight incubation at 4 °C with primary antibodies against IRβ and AKT. The membranes were processed as previously described, including incubation with HRP-conjugated secondary antibodies and signal detection through chemiluminescence.

## OP–I permeation in 3D skin equivalent EpiKutis

The 3D skin model (EpiKutis) was fabricated according to a previously reported method[40]. In brief, keratinocytes ($5 \times 10^5$) were seeded onto the permeable membrane of a Transwell chamber, cultured at 37 °C in a 5% CO$_2$ atmosphere for 2 days and then cultured at the air–liquid interface for 8 days with daily medium replacement. The complete EpiKutis 3D model was obtained and used as a skin model for investigating OP–I skin permeability. OP–I, PEG–I or insulin solution (insulin-equivalent dose, 0.5 mg ml$^{-1}$; 0.2 ml) was added to the donor compartment, and 0.4 ml fresh medium was added to the receiving compartment. The temperature was maintained at 37 °C. At timed intervals, 50 μl solution was withdrawn from the receiving compartment and an equal volume of fresh medium was added. The insulin concentration was quantified

by ELISA kits. The unit conversion was calculated according to the following formula, and the cumulative amount of insulin permeating per unit area of the model skin ($Q_n$) was calculated:

$$Q_n = \frac{C_n V_r + \sum_{i=1}^{n-1} C_i V_s}{A}$$

Where, $Q_n$ is the cumulative amount of insulin permeating per unit area (μg cm$^{-2}$); $C_n$ is the insulin concentration in the receiver cell at sampling timepoint t (μg ml$^{-1}$); $C_i$ is the insulin concentration of the receiving liquid at the intermediate point (μg ml$^{-1}$); $V_r$ is the volume of the receiving pool (0.4 ml); $V_s$ is the volume of the sampled receiving solution (50 μl); and $A$ is the effective transmission area of the EpiKutis (0.081 cm$^2$).

The steady-state flow rate ($J_{ss}$) was obtained as the slope of the curve of $Q_n$ as a function of time. The apparent permeability coefficient ($K_p$) was calculated by the following formula:

$$K_p = \frac{J_{ss}}{C_0}$$

Where $C_0$ is the initial concentration of insulin or its conjugate in the donor cell.

EpiKutis (0.081 cm$^2$) was incubated with 0.2 ml OP–I$^{FITC}$, PEG–I$^{FITC}$ or insulin$^{FITC}$ (FITC-equivalent concentration, 1 μg ml$^{-1}$) for 4 h, then washed three times with PBS, stained with DAPI and finally imaged using CLSM with the z-stack tomoscan model at a 10-μm interval from the bottom to top of the 3D skin model with excitation at 405 nm for DAPI and 488 nm for FITC.

## Skin penetration analysis by intravital two-photon microscopy

The mice were topically treated with OP–I$^{Cy5}$, as described above. The skin was mounted between a coverslip and a sliding glass for two-photon imaging analysis. The laser wavelength for two-photon excitation was 480 nm, and the laser power delivered to the skin sample was 90 mW. Sequential z-stack images were captured at 3-μm intervals from the skin surface until the fluorescence signal became undetectable. The xz-axis orthogonal view of the SC layers was reconstructed using volume viewer plugins in ImageJ.

## Subcutaneous lymphatic vessel co-localization

OP–I$^{Cy5}$ (Cy5-equivalent dose, 0.2 ml of 10 μg ml$^{-1}$) was topically applied on the dorsal skin of SD rats. After 4 h, the tissue at the application site was excised and frozen-sectioned. The lymphatic vessels were immunostained using LYVE-1$^{AF488}$ (1:1,000), and the nuclei were counterstained with DAPI. Subsequently, the distribution of OP–I$^{Cy5}$ within the lymphatic vessels was examined using CLSM.

## Skin retention of OP–I$^{Cy5}$

After 4 h of topical application with OP–I$^{Cy5}$ on the dorsal skin of C57BL/6J mice (Cy5-equivalent dose, 10 μg ml$^{-1}$; 0.2 ml per mouse), the OP–I$^{Cy5}$ solution was removed and the treated skin site was gently washed three times with PBS. Subsequently, the mice were euthanized at 4 or 8 h after removal of the OP–I$^{Cy5}$ solution. The fluorescence intensity in the skin was imaged using the IVIS Spectrum imaging system and CLSM.

## In vivo skin permeation of OP–I$^{Cy5}$ in minipigs

The OP–I$^{Cy5}$ cream (Cy5-equivalent dose, 10 μg ml$^{-1}$) was topically applied onto the minipigs' abdominal skin (cream volume, 10 ml; application area, 100 cm$^2$). After 4 h, the minipigs were euthanized, and the treated site skins were washed three times with PBS, dissected, fixed with 4% PFA and sectioned into slices, as described above. The slices were stained with DAPI to label the cell nuclei and with NBD-C6-HPC to label the SC lipids and the cell membranes in the viable epidermis.

Fluorescence images were taken using CLSM with excitation at 405 nm for DAPI, 488 nm for NBD-C6-HPC and 640 nm for OP–I$^{Cy5}$. The CLSM images were analysed using ImageJ.

## OP–I$^{Cy5}$ biodistribution in mice
Male C57BL/6J mice were randomly grouped ($n$ = 3 mice). Each mouse was topically applied with 0.2 ml OP–I$^{Cy5}$, PEG–I$^{Cy5}$ or insulin$^{Cy5}$ solution (Cy5-equivalent dose, 10 μg ml$^{-1}$) on the mouse dorsal skin for 4 h, and then the solution was removed. Insulin$^{Cy5}$ was administered subcutaneous to the mice in the control group (Cy5-equivalent dose, 25 μg per kg). The mice were euthanized at timed intervals. The main organs and tissues, including the heart, liver, spleen, lung, kidneys, adipose tissues (including brown adipose tissue, subcutaneous white adipose tissue and visceral white adipose tissue), and skeletal muscles were collected, and their fluorescence intensities were measured using the IVIS Spectrum imaging system.

## In vivo studies using STZ-induced diabetic mice
Type 1 diabetic mice were established through intraperitoneal delivery of STZ (150 mg per kg in 10 mg ml$^{-1}$ disodium citrate buffer, pH 4.5) into healthy male C57BL/6J mice (aged 6–8 weeks, ~25 g). BGLs were monitored, and the mice with BGLs exceeding 300 mg dl$^{-1}$ were considered diabetic. Diabetic mice with fasting BGLs within the 300 to 600 mg dl$^{-1}$ range were selected for the experiments. BGLs were measured from tail-vein blood (around 3 μl) using a calibrated Sinocare glucose meter.

A diffusion cell (diffusing area: 1.13 cm$^2$) containing insulin, OP–I or PEG–I (insulin-equivalent dose, 116, 58 or 29 U kg$^{-1}$; concentration, 0.5 mg ml$^{-1}$; volume, 0.2, 0.1 or 0.05 ml per mouse) or the mixture of OP and insulin (OP + insulin; insulin-equivalent dose, 116 U kg$^{-1}$, 0.5 mg ml$^{-1}$; 0.2 ml per mouse) was applied on the dorsal or abdominal skin of STZ-induced diabetic mice. For the positive control group, mice were administered insulin subcutaneously at a dose of 5 U kg$^{-1}$. BGLs were measured at timed intervals. The skin insulin concentrations and plasma insulin concentrations in the blood samples (25 μl) collected from the tail vein were quantified using ELISA kits according to the manufacturer's protocol. The diabetic mice were subjected to a fasting period of 12 h during the experimental procedures.

The repeatability of OP–I's hypoglycaemic effect was assessed by applying OP–I (insulin-equivalent dose, 116 U kg$^{-1}$, 0.5 mg ml$^{-1}$; 0.2 ml per mouse) topically onto the same diabetic mice over three consecutive days. Plasma insulin concentrations were measured at timed intervals using ELISA.

For the IPGTT experiment, diabetic mice were topically applied with OP–I (insulin-equivalent dose, 116 U kg$^{-1}$, 0.5 mg ml$^{-1}$; 0.2 ml per mouse); then, 1 h later, they were intraperitoneally injected with 0.2 ml of glucose solution (1.5 g per kg). BGLs were monitored over time. The diabetic mice were subjected to a fasting period of 3 h during the experimental procedures.

## In vivo studies using STZ-induced diabetic minipigs
Guangxi Bama minipigs (aged 6 months; weight, 35–40 kg) were infused with STZ in freshly prepared disodium citrate buffer (75 mg ml$^{-1}$, pH 4.5) at a 150 mg per kg dose within 10 min and then maintained for recovery. The glucose levels were monitored using CGMS[15] (Dexcom G4 Platinum Continuous Glucose Monitor System, Dexcom). A BGL that is constantly higher than 250 mg dl$^{-1}$ indicates the successful establishment of the insulin-deficient diabetic minipig model.

OP–I, PEG–I or native insulin was dispersed in the cream (water-in-oil) at an insulin-equivalent concentration of 1 mg ml$^{-1}$, as described above. The cream was topically applied onto the minipig abdominal skin at an insulin-equivalent dose of 40 mg or 10 mg ($n$ = 3 for each group). The area with the cream was wrapped with plastic film. The BGLs of minipigs were continuously monitored using the CGMS. The diabetic minipigs were subjected to a fasting period of 12 h during the experimental procedures.

The skin samples were excised from the administration skin site at the end of the experiment. The samples were processed for sectioning and stained with haematoxylin and eosin and terminal deoxynucleotidyl transferase dUTP nick end labelling (TUNEL). Moreover, control skin samples were collected from healthy minipigs in the same anatomical region without treatment.

## SC sample collection and observation
The procedure was performed according to a previously reported method[34]. At 4 h after topical application of OP–I$^{Cy5}$, PEG–I$^{Cy5}$ or insulin$^{Cy5}$ (Cy5-equivalent dose, 10 μg ml$^{-1}$; 0.2 ml per mouse), the treated sites were carefully washed three times with PBS and dried. Double-sided adhesive tape (Scotch 3M) was pressed onto the skin surface for 2 s and then peeled off in the longitudinal direction. The middle part of the tape was fixed on a microscopy slide, stained with NBD-C6-HPC and then observed under CLSM at 488 nm excitation for NBD-C6-HPC and 640 nm excitation for Cy5.

## FTIR spectrometry
PBS (1 ml), OP (0.5 mg ml$^{-1}$ in 1 ml PBS), OP–I (0.5 mg ml$^{-1}$ in 1 ml PBS) or oleic acid (50 mg ml$^{-1}$ in 1 ml propylene glycol) was evaluated using RYJ-12B Franz diffusion cells with male C57BL/6J mouse skin. Mouse skin samples (3 cm × 3 cm) were fixed in glass holders with a 2.2 cm$^2$ circular permeation area, mounted in Franz cells with the epidermal side up, and the acceptor compartment (8 ± 0.5 ml) was filled with pH 7.4 PBS. The setup was maintained on a magnetic stirrer in a water bath at 37 ± 0.5 °C. After incubating for 24 h with the test compounds, skin samples were gently washed with PBS and analysed using a Thermo Fisher Scientific Nicolet iS50 FTIR spectrometer. Spectra were acquired by co-adding 128 scans at 4 cm$^{-1}$ resolution over the frequency range of 4,000–650 cm$^{-1}$.

## MD simulations
The insulin (PDB: 1AI0 (ref. 41), chain I) and its receptor (IR, PDB: 6SOF (ref. 42)) structures were obtained from the Protein Data Bank (PDB). The structures of OP and OP–I were constructed using Avogadro software[43]. OP was set to have 32 repeating units. The force-field parameters of OP and OP–I were obtained from the Paramchem webserver[44] and CGenFF[45]. The SC lipid membrane, composed of an equimolar mixture of ceramide, cholesterol and free fatty acids, was generated by the membrane builder CHARMM-GUI[46,47]. The CHARMM36[48] force field was used to model the insulin and SC lipids. At pH 5.5, 20% of the $N$-oxide groups of OP were assumed to be protonated, whereas 50% of the fatty acids were considered to be deprotonated. At pH 7.0, OP was zwitterionic, while all fatty acids were deprotonated. Each system was solvated in an 11.4 × 11.4 × 16.1 nm$^3$ water box with around 219,000 atoms. To study the binding interactions of insulin or OP–I with IR, the respective system was solvated in a 23.0 × 23.0 × 23.0 nm$^3$ water box containing around 1,225,000 atoms. To study the interactions of multiple insulin, OP or OP–I molecules, the respective system was solvated in a 14.7 × 14.7 × 14.7 nm$^3$ water box containing around 325,000 atoms. Water molecules were modelled by the TIP3P water model[49]. Na$^+$ and Cl$^-$ ions were added to neutralize each system and bring its total ionic strength to 0 at the physiological concentration of 150 mM.

All MD simulations were carried out using the program GROMACS 2020.6[50,51]. VMD[52] was used for trajectory visualization. The covalent bonds with hydrogen atoms were constrained by the LINCS algorithm[53], which allowed a time step of 2 fs. The long-range electrostatic interactions were calculated using the particle-mesh Ewald method[54], whereas the van der Waals interactions were calculated with a smooth cut-off of 1.2 nm. Periodic boundary conditions were applied in all directions. The NPT ensemble with semi-isotropic pressure coupling was applied with the pressure (1 bar) controlled by the Parrinello–Rahman barostat[55] and the temperature (310 K) by the v-rescale thermostat[56]. Before production runs, the lipid system was equilibrated for 100 ns; then, 200-ns runs

were conducted to monitor insulin, OP or OP–I adsorption on the SC lipids. After adsorptions, three independent 100-ns runs were further performed for insulin, OP or OP–I to monitor their diffusions on SC lipids. To analyse the binding interactions of insulin or OP–I with IR, 200-ns MD simulations were conducted for each system. To investigate the interactions of multiple insulin, OP or OP–I molecules at pH 6.0, 400-ns MD simulations were conducted for each system. In production simulations, all atoms were free to move.

The friction coefficient $\gamma$ of insulin/OP/OP–I on SC lipids was derived from the Stokes–Einstein relation:

$$\gamma = \frac{k_B T}{D}$$

where $k_B$ is the Boltzmann constant, $T$ is the temperature and the diffusion coefficient $D$ follows

$$D = \frac{\langle r^2(t) \rangle}{k t}$$

Where $k = 4$ on a 2D surface, the time $t$ and the displacement $r(t)$ were obtained from three independent 100-ns MD runs after adsorption.

## PMF analyses

The PMF results were calculated using the umbrella sampling protocol[57]. The PMF setups were similar to the aforementioned MD setups to estimate the energy barriers for the diffusion of insulin, OP or OP–I on the SC lipids. The total transverse distance along each representative path was 2 nm, which was divided into 20 windows with a resolution of 0.1 nm. Position restraints were applied to the SC lipids when the energy barriers were scanned. For monitoring the adsorption of insulin, OP, or OP–I on SC lipids, the simulation boxes were extended to $11.4 \times 11.4 \times 22.1$ nm$^3$ by adding 0.15 mM NaCl solution in the boxes. After a further 30-ns equilibration, the sampling path of each system was obtained by pulling insulin, OP or OP–I in the perpendicular ($z$) direction to the SC lipid membrane with a constant velocity of 0.2 nm ns$^{-1}$. The total sampling distance in the $z$ direction was 9 nm for each system, which was divided into 90 windows with a resolution of 0.1 nm. At each window, the system was first equilibrated for 5 ns, followed by a 35-ns productive umbrella sampling with a restraint force constant of 1,000 kJ mol$^{-1}$ nm$^{-2}$. To evaluate the binding affinity of insulin or OP–I with IR, the sampling path of each system was determined by pulling insulin or OP–I perpendicular to the IR surface at a constant velocity of 0.2 nm ns$^{-1}$. The total sampling distance of 5 nm was divided into 50 windows with a resolution of 0.1 nm. At each window, the system underwent a 2-ns equilibration phase followed by a 20-ns productive umbrella sampling simulation, applying a restraint force constant of 1,000 kJ mol$^{-1}$ nm$^{-2}$.

## VE penetration study using 3D-cultured multilayer HaCat spheroids

Multilayer cell spheroids were prepared using the hanging-drop method. HaCat cells were suspended in fresh DMEM medium (containing 0.12% (w/v) methylcellulose) at a density of $4 \times 10^5$ cells per ml. The cell suspensions (25 µl) were dropped onto the lids of the cell culture plate to form uniform droplets, and 20 ml PBS was added to the plate to keep the droplets moist. The cells were incubated for 72 h and formed dense spheroids, which were transferred to an agarose-coated (1% (w/v) in PBS) 96-well plate with one spheroid per well and incubated for another 72 h to mature. The spheroids were incubated with OP–I$^{FITC}$, PEG–I$^{FITC}$ or insulin$^{FITC}$ at an FITC-equivalent dose of 1 µg ml$^{-1}$ for timed intervals. The spheroids were washed with PBS and imaged using CLSM by $z$-stack tomoscan at 20 µm intervals from the bottom to the middle of the spheroids. The integration of FITC fluorescence density and linescan analysis were performed using Image J.

For the effects of the endocytosis and exocytosis inhibitors on the penetration of OP–I$^{FITC}$, the HaCat spheroids were separately treated with PBS, wortmannin (2 µM), cytochalasin D (20 µM), monensin (20 µM), nocodazole (10 µM) or brefeldin A (10 µM) for 2 h, and then incubated with OP–I$^{FITC}$ (FITC-equivalent dose, 1 µg ml$^{-1}$) for 4 h. The HaCat spheroids were imaged and analysed as described above.

## Localization of OP–I$^{Cy5}$ at HaCat cell membranes

HaCat cells were plated onto glass-bottom dishes at a density of $1 \times 10^5$ cells per dish and incubated for 24 h. The HaCat cells were incubated with OP–I$^{Cy5}$ (Cy5-equivalent dose, 1 µg ml$^{-1}$) for 6 h, 12 h or 24 h. The cell membrane was stained with NBD-C6-HPC (1 µM) for 5 min, and then the cells were washed three times with PBS and observed under CLSM. Fluorescence images were taken using CLSM with excitation at 488 nm for NBD-C6-HPC and 640 nm for Cy5.

The cells were imaged immediately at maximum projection after adding OP–I$^{Cy5}$ (Cy5-equivalent dose, 1 µg ml$^{-1}$) for time-lapse videos.

## Observation of contact-dependent direct transfer of OP–I among HaCat cells

HaCat cells were seeded into six-well plates at $1.25 \times 10^5$ or $2.5 \times 10^5$ cells per well and allowed to adhere overnight. The cells were then incubated with OP–I$^{Cy5}$ (Cy5-equivalent dose, 1 µg ml$^{-1}$) for 12 h and then extensively rinsed with sterilized PBS and isolated. The OP–I$^{Cy5}$-treated HaCat cells were mixed with untreated HaCat$^{GFP}$ cells at the same cell density. The mixed cells were co-cultured in DMEM medium for 12 h and imaged with CLSM with excitation at 488 nm for GFP and 640 nm for Cy5. The mixed cells co-cultured in DMEM medium for 3 h, 6 h or 12 h were further isolated and analysed by flow cytometry. First, an FSC-A versus SSC-A gate was applied to exclude debris and select live cells based on forward scatter (FSC) and side scatter (SSC) characteristics. Next, a GFP gate (FL1-H) was set to include cells with fluorescence in the GFP channel, followed by establishing a Cy5 gate (FL4-H) to isolate Cy5-positive cells. Finally, an intersection gate between the GFP and Cy5 gates was used to identify dual-positive cells. Every 10,000 cells were counted to determine GFP-positive cells at the FL1 channel and Cy5-positive cells at the FL4 channel. The experiment was repeated three times independently; FlowJo (v.10.0) software was used for analysis. The transfer efficiency of OP–I$^{Cy5}$ to HaCat$^{GFP}$ cells was calculated according to the formula.

$$\text{Transfer efficiency(\%)} = \frac{\text{Number of HaCat}^{GFP+Cy5} \text{ cells}}{\text{Total counted number of HaCat}^{GFP} \text{ cells}} \times 100$$

## Non-contact inhibition of intercellular transfer of OP–I$^{Cy5}$

HaCat cells were seeded onto two coverslips (1 and 2) and incubated overnight. The cells on a coverslip 1 were first cultured with OP–I$^{Cy5}$ (Cy5-equivalent dose, 1 µg ml$^{-1}$) for 4 h, rinsed with PBS three times and then co-incubated with the coverslip 2 with untreated cells in a fresh medium for 12 h. The cells on the coverslips were washed with PBS, and the cell membrane was stained with NBD-C6-HPC (1 µM) for 5 min. The cells were imaged with CLSM at 488 nm excitation for NBD-C6-HPC and 640 nm for Cy5.

## Intercellular transfer between cells on separate coverslips

Two coverslips were seeded with HaCat cells ($10^5$ each) and cultured overnight to ensure full adherence. The cells on one coverslip were incubated with OP–I$^{Cy5}$ (Cy5-equivalent dose, 1 µg ml$^{-1}$) for 12 h and then washed three times with PBS, noted as coverslip 1. Coverslip 1 was placed on top of the other coverslip with untreated HaCat cells (coverslip 2) and pressed slightly. The coverslips were cultured together in DMEM medium for 0.5 or 1 h and then observed using CLSM for the transfer of OP–I$^{Cy5}$.

## TIRFM imaging

HaCat cells were seeded onto a confocal dish at a density of $1 \times 10^5$ cells per well and incubated for 24 h. Subsequently, OP–I$^{Cy3}$ (Cy3-equivalent dose, 1 µg ml$^{-1}$) was added. After 4 h, NBD-C6-HPC was added to stain the cell membrane for 5 min, and the cells were then washed three times with PBS. Time-lapse imaging of the cell membrane was conducted using the TIRF function of an Olympus IX83 microscope. ImageJ was used for image analysis.

## Statistical analyses

Statistical tests were performed using Prism (GraphPad software, v.10.4.0). One-way analysis of variance was used for multiple comparisons. Unpaired Student's *t*-tests were used to analyse the difference between two groups.

## Reporting summary

Further information on research design is available in the Nature Portfolio Reporting Summary linked to this article.

## Data availability

Data supporting the findings of this study are provided in the Article and its Supplementary Information. The input files (.tpr files) for the key simulations in this study are available at Zenodo[58] (https://doi.org/10.5281/zenodo.17078485). Source data are provided with this paper.

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

**Acknowledgements** This work was supported by the National Key Research and Development Program (2021YFA1201200), the National Natural Science Foundation (52495010, 51833008, 52203193, U1967217) of China, the National Center of Technology Innovation for Biopharmaceuticals (NCTIB2022HS02010), Shanghai Artificial Intelligence Lab (P22KN00272) and the Starry Night Science Fund (SN-ZJU-SIAS-003). We thank Q. Lin for her technical assistance on TEM (JEOL JEM F200) characterization; J. Li for her technical assistance on FTIR and MALDI-TOF MS characterization; S. Fang and S. Liu for their technical support on CLSM and TIRFM; D. Xu for help with minipig experiments. Cartoons in Fig. 3i and Extended Data Fig. 3a were created using BioRender with granted publication licences.

**Author contributions** Y.S. conceived and supervised the project. R. Zhou, J.X., Z.Z. and R.C. co-supervised the project. Q.W., Y.P., R. Zhang, Y.F., J.Y. and Y.Y. performed the biological experiments and analysed the data. Z.L., Y.G. and H.Z. carried out the synthesis. Z.H., J.H. and M.X. carried out the MD simulations supervised by R. Zhou. J.Z. and Y.Z. established type 1 diabetic models. Q.Z., S.S. and J.T. helped with the discussion of the synthesis and mechanism. All of the authors discussed the results and contributed to the writing of the manuscript.

**Competing interests** The authors declare no competing interests.

**Additional information**
**Correspondence and requests for materials** should be addressed to Jiajia Xiang, Rongjun Chen, Ruhong Zhou or Youqing Shen.

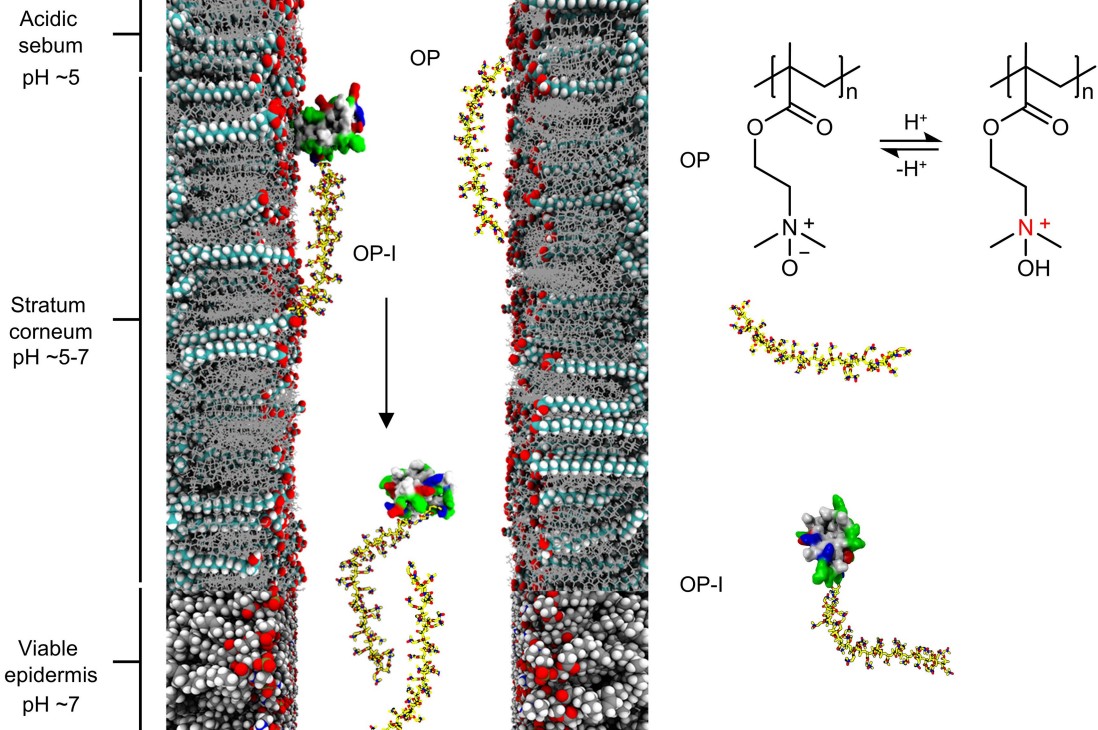

**Extended Data Fig. 1 | Schematic of the skin penetration mechanism of OP and its conjugate with insulin (OP-I).**

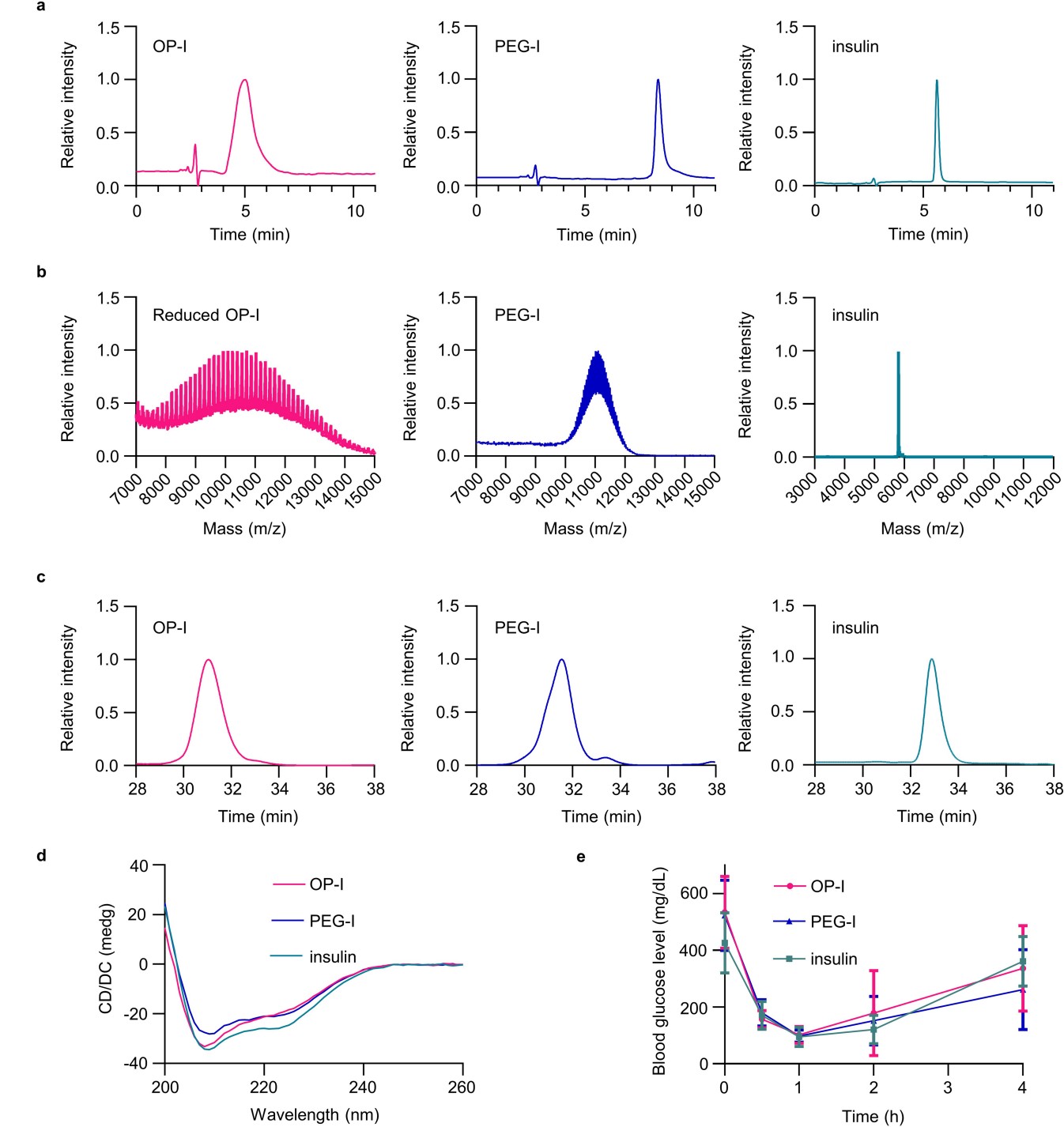

**Extended Data Fig. 2 | Characterization of insulin conjugates OP-I and PEG-I. a**, RP-HPLC traces at an absorbance of 280 nm. **b**, MALDI-TOF MS spectra (OP-I was reduced to PDMA-I). **c**, GPC traces in $H_2O$. **d**, CD spectra in $H_2O$. **e**, BGLs of diabetic mice after subcutaneous (s.c.) injection of OP-I, PEG-I, or native insulin; insulin-equivalent dose, 5 U $kg^{-1}$. Data are mean ± s.d.; $n = 5$ mice.

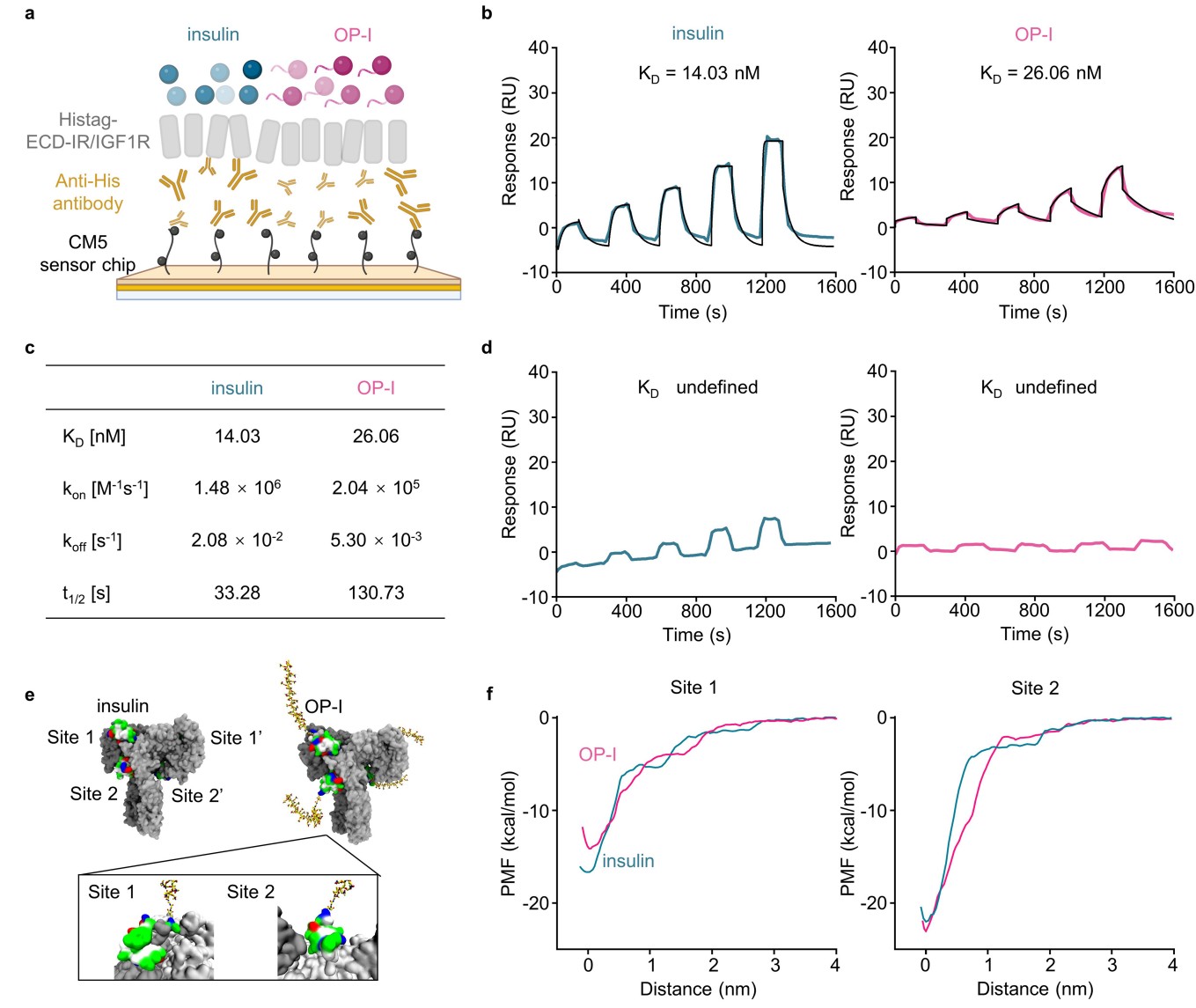

**Extended Data Fig. 3 | SPR characterization of insulin receptor (IR) binding and activation of insulin and OP-I. a**, Schematic illustration of the experimental setup, where anti-His antibody was immobilized on the CM5 chip sensor surface, followed by the addition of Histag-ECD-IR/IGF1R, and then insulin/OP-I was introduced to study their interactions. Created in BioRender.com. **b**, SPR traces showing the binding of insulin and OP-I to ECD-IR; RU, resonance unit. **c**, Dissociation constant ($K_D$), association rate constant ($k_{on}$), dissociation rate constant ($k_{off}$), and half-life ($t_{1/2}$) of insulin or OP-I binding to ECD-IR, calculated from SPR binding curves in Extended Data Fig. 3b. **d**, SPR traces showing minimal binding of insulin and OP-I to ECD-IGF1R; RU, resonance unit. **e**, Representative binding modes from all-atom MD simulations of insulin and OP-I on the ECD-IR. **f**, PMF analysis of the binding affinities with two major ECD-IR binding sites. Umbrella sampling distance: 5 nm; window resolution: 0.1 nm; sampling time: 20 ns per window; restraint force constant: 1000 kJ mol⁻¹ nm⁻².

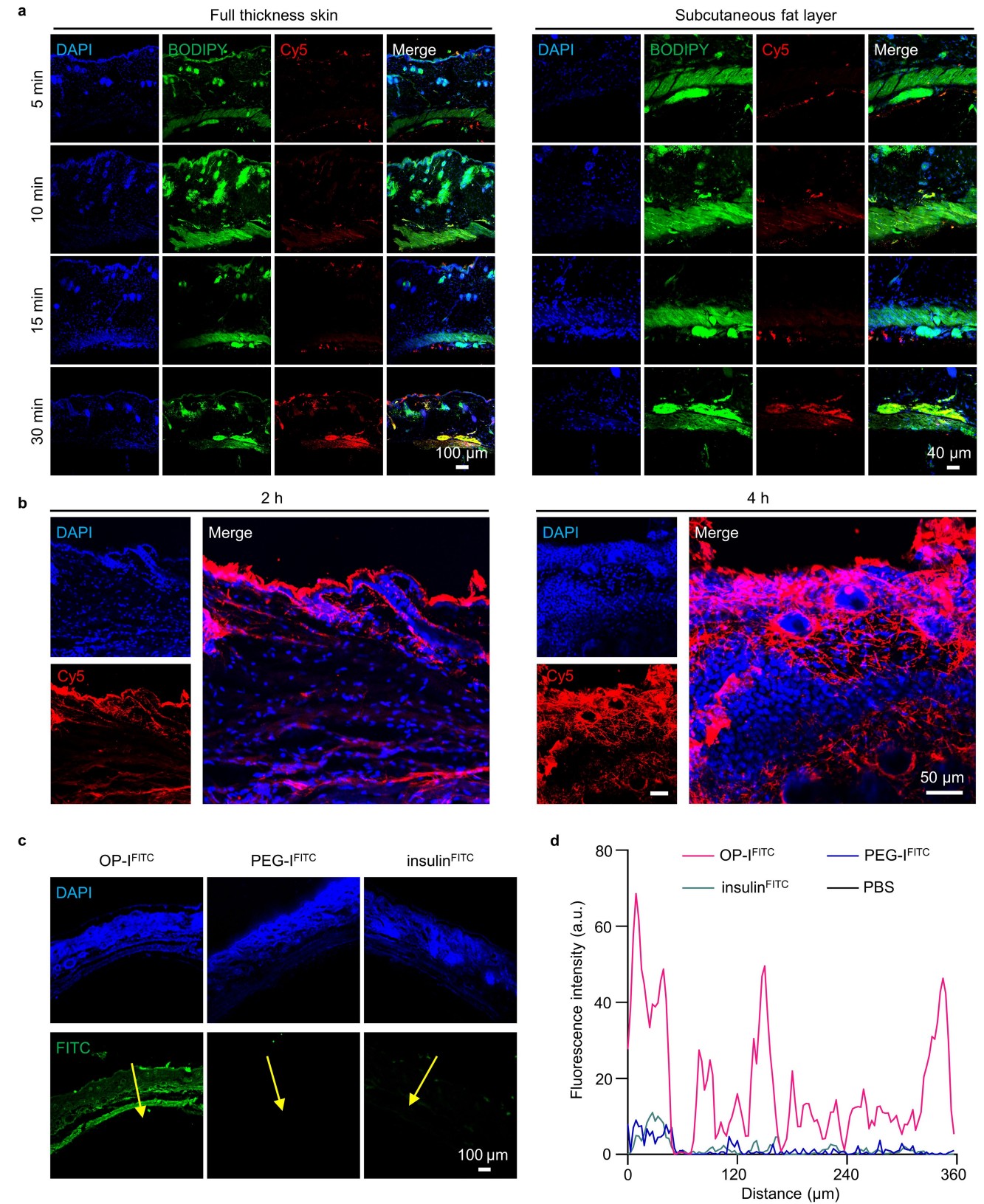

**Extended Data Fig. 4 | Characterization of OP-I permeability in mouse skin.**
**a**, Fluorescence distribution in the skin (left) and its s.c. fat (right, enlarged view) of the C57BL/6J mouse dorsal skin after topical application with OP-I$^{Cy5}$ for timed intervals (Cy5-equivalent dose, 0.2 ml of 10 µg ml$^{-1}$ applied on 1.13 cm$^2$). BODIPY (green) was used to label the s.c. adipose tissue, and DAPI (blue) was used to label the nuclei. **b**, CLSM images of the slices of the C57BL/6J mouse

dorsal skin after topical application with OP-I$^{Cy5}$ (Cy5-equivalent dose, 0.2 ml of 10 µg ml$^{-1}$ applied on 1.13 cm$^2$) for 2 or 4 h. **c,d**, CLSM images (**c**) and line-scan analysis of FITC fluorescence intensity from the skin surface to the subcutis along randomly drawn lines (yellow arrows in **c**) (**d**) of vertically sectioned slices of treated mouse dorsal skin after 4 h topical administration of OP-I$^{FITC}$. The images are representative of $n = 3$ independent experiments.

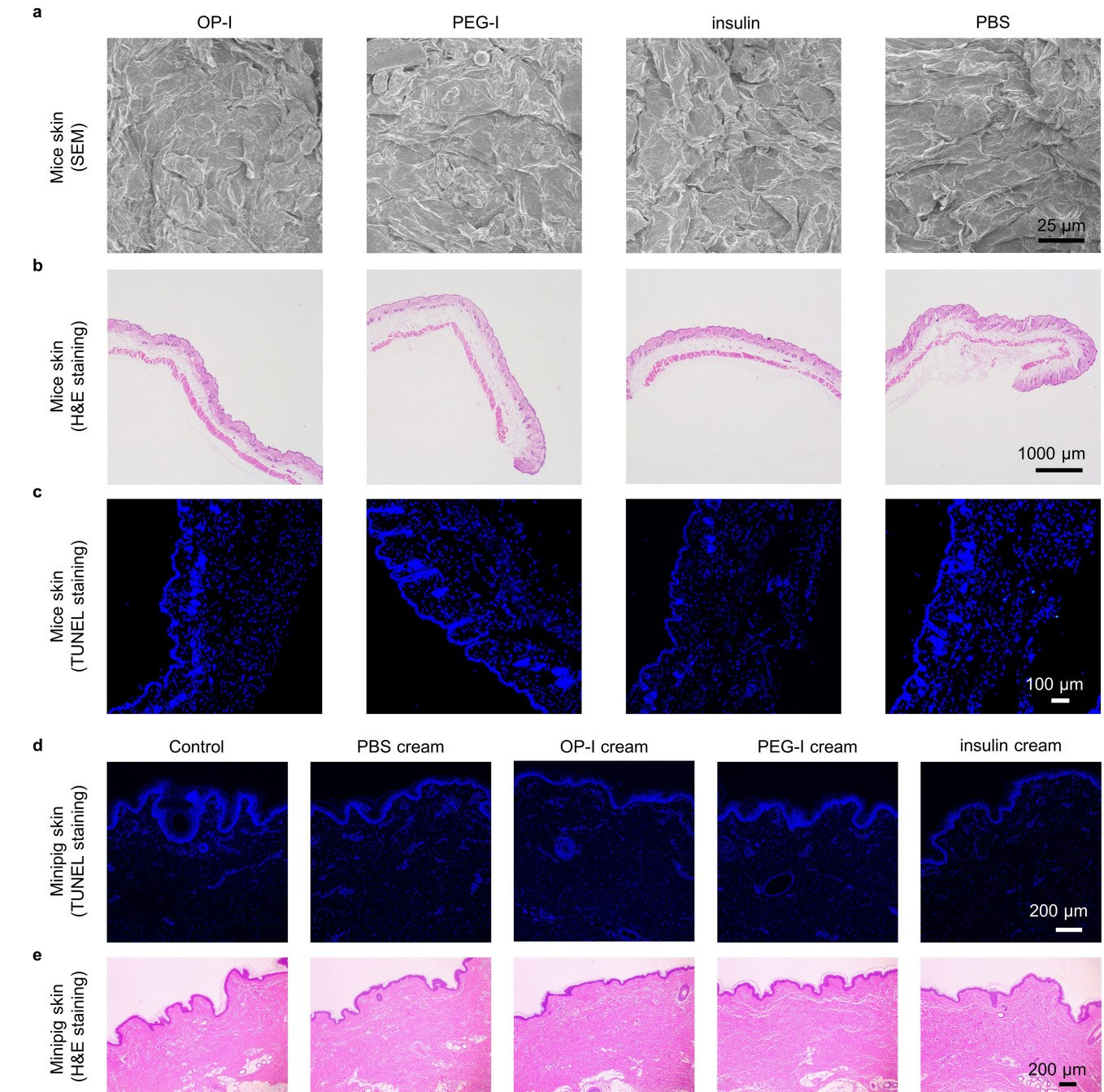

**Extended Data Fig. 5 | Biological safety evaluation of OP-I for transdermal application. a-c**, Representative images of SEM (**a**), H&E staining (**b**), and immunohistological staining with TUNEL assay (green) and Hoechst (blue) (**c**) of the mouse dorsal skin after 4 h of topical treatment with OP-I, PEG-I, native insulin, or PBS (insulin-equivalent dose, 116 U kg$^{-1}$; 0.2 ml of 0.5 mg ml$^{-1}$; application area, 1.13 cm$^2$). **d,e**, Representative images of H&E staining (**d**) and immunohistological staining with TUNEL assay (green) and Hoechst (blue) (**e**) of the treated sites of the minipig abdominal skin after 4 h of topical treatment with OP-I, PEG-I, native insulin, or PBS (insulin-equivalent dose, 40 ml of 1 mg ml$^{-1}$, 29 U kg$^{-1}$; application area, 400 cm$^2$). The untreated skin was designated as the Control group. The images are representative of $n = 3$ independent experiments.

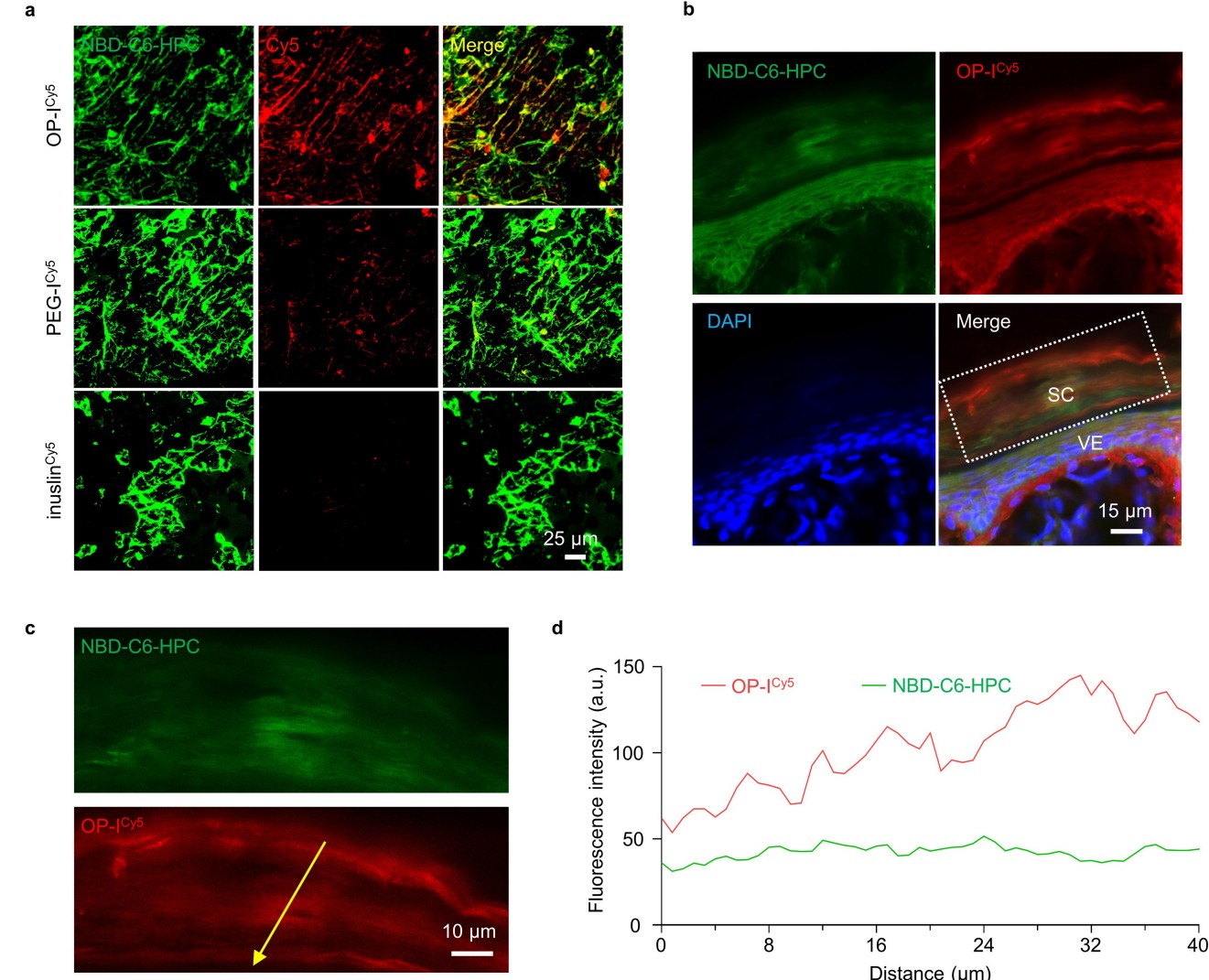

**Extended Data Fig. 6 | Co-localization of OP-I with SC lipids. a**, CLSM images of the adhesive tape peeled from the mouse dorsal skin after 4 h of topical application with OP-I$^{Cy5}$, PEG-I$^{Cy5}$, or insulin$^{Cy5}$ (Cy5-equivalent dose, 0.2 ml of 10 µg ml$^{-1}$; application area, 1.13 cm$^2$). The images are representative of $n$ = 3 independent experiments. **b**, CLSM images of the vertically sectioned slices of minipig abdominal skin after 4 h of topical application with OP-I$^{Cy5}$ (Cy5-equivalent dose, 10 ml of 10 µg ml$^{-1}$; application area, 100 cm$^2$). The images are representative of $n$ = 3 independent experiments. **c**, Enlarged view of the selected area in (**b**) outlining a corneocyte. **d**, Line-scan analysis of NBD-C6-HPC and Cy5 fluorescence along the yellow arrow in (**c**). Cy5 fluorescence is shown in red, and SC intercellular lipids stained with NBD-C6-HPC are in green.

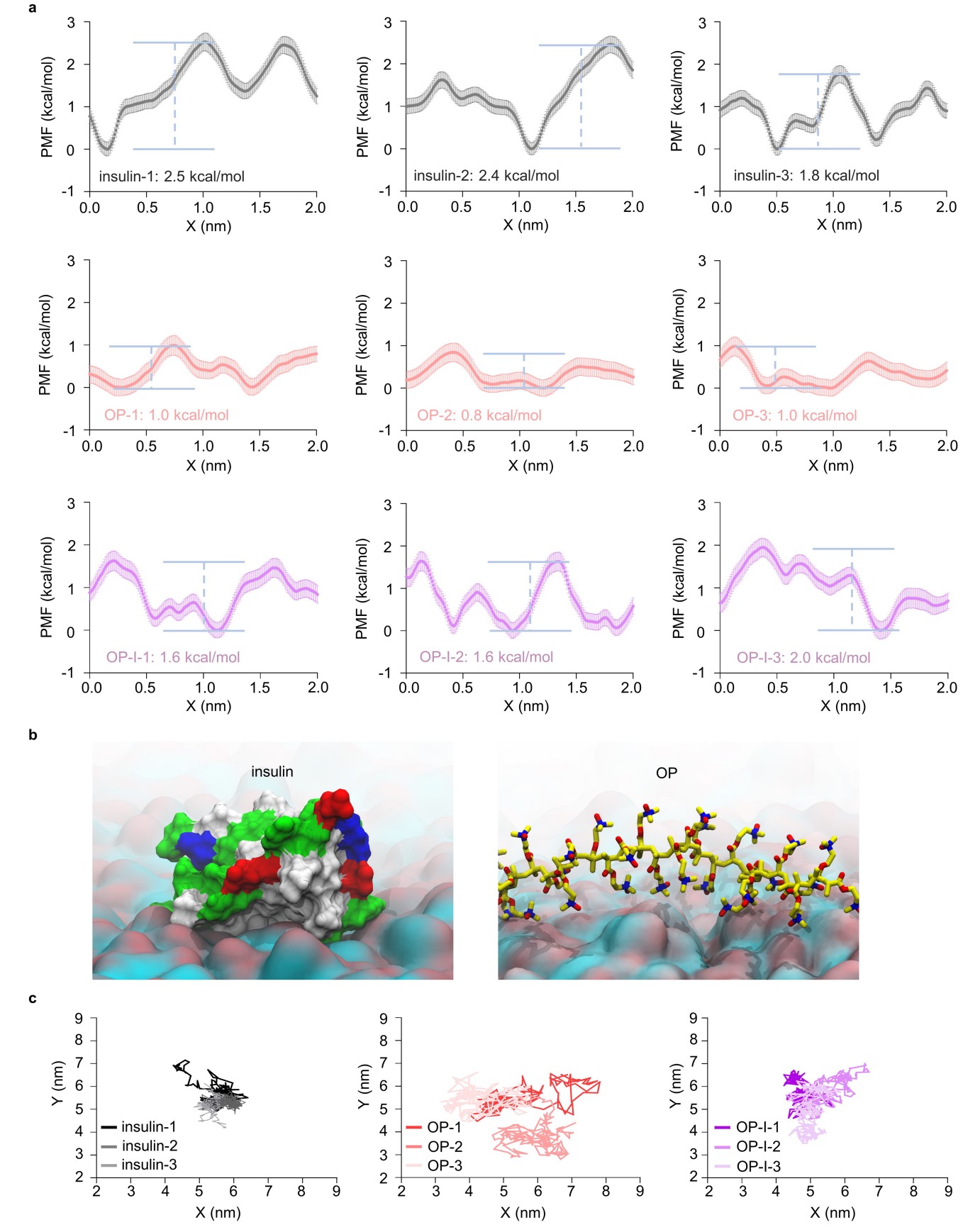

**Extended Data Fig. 7** | See next page for caption.

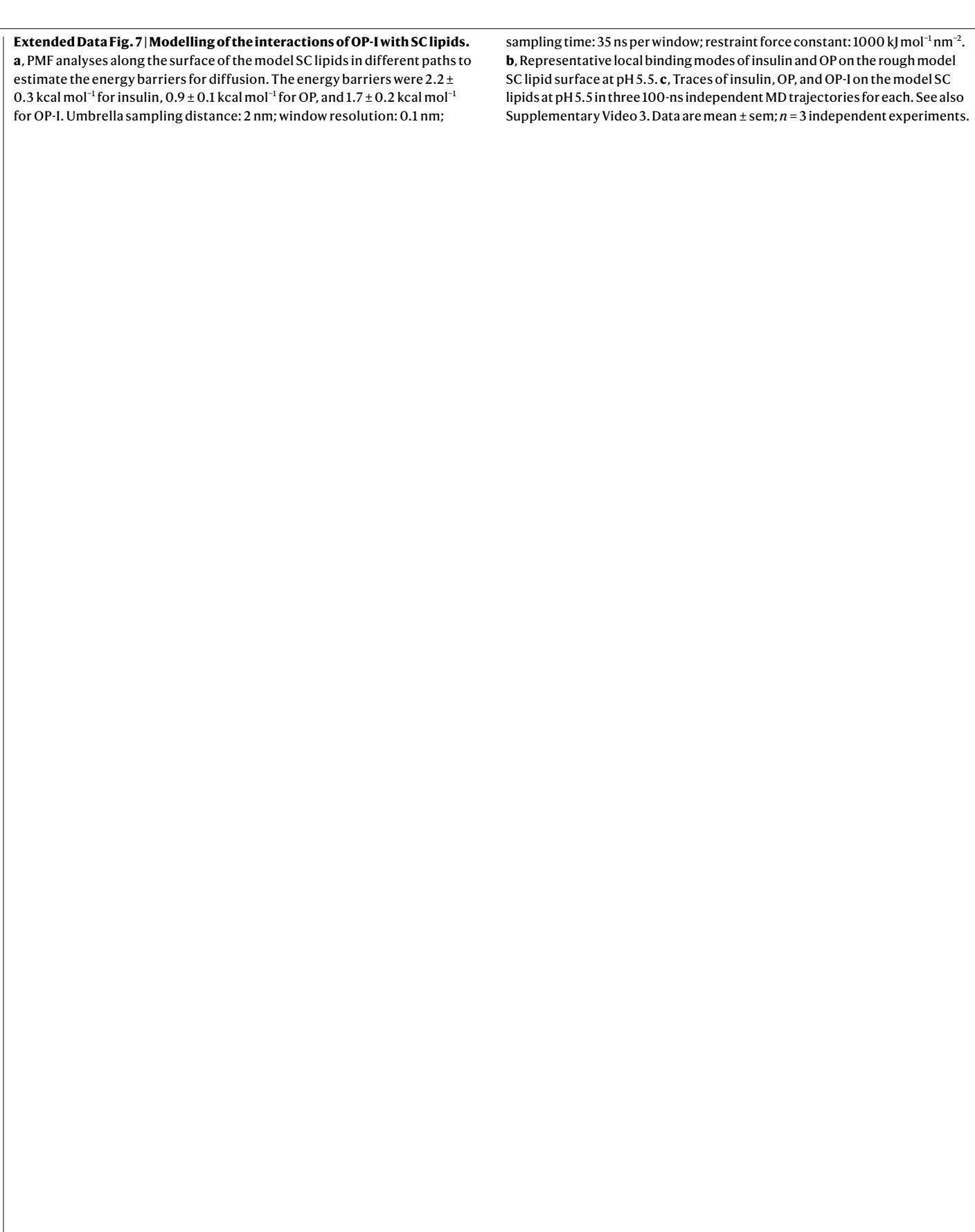

**Extended Data Fig. 7 | Modelling of the interactions of OP-I with SC lipids.** **a**, PMF analyses along the surface of the model SC lipids in different paths to estimate the energy barriers for diffusion. The energy barriers were $2.2 \pm 0.3$ kcal mol$^{-1}$ for insulin, $0.9 \pm 0.1$ kcal mol$^{-1}$ for OP, and $1.7 \pm 0.2$ kcal mol$^{-1}$ for OP-I. Umbrella sampling distance: 2 nm; window resolution: 0.1 nm; sampling time: 35 ns per window; restraint force constant: 1000 kJ mol$^{-1}$ nm$^{-2}$. **b**, Representative local binding modes of insulin and OP on the rough model SC lipid surface at pH 5.5. **c**, Traces of insulin, OP, and OP-I on the model SC lipids at pH 5.5 in three 100-ns independent MD trajectories for each. See also Supplementary Video 3. Data are mean ± sem; $n = 3$ independent experiments.

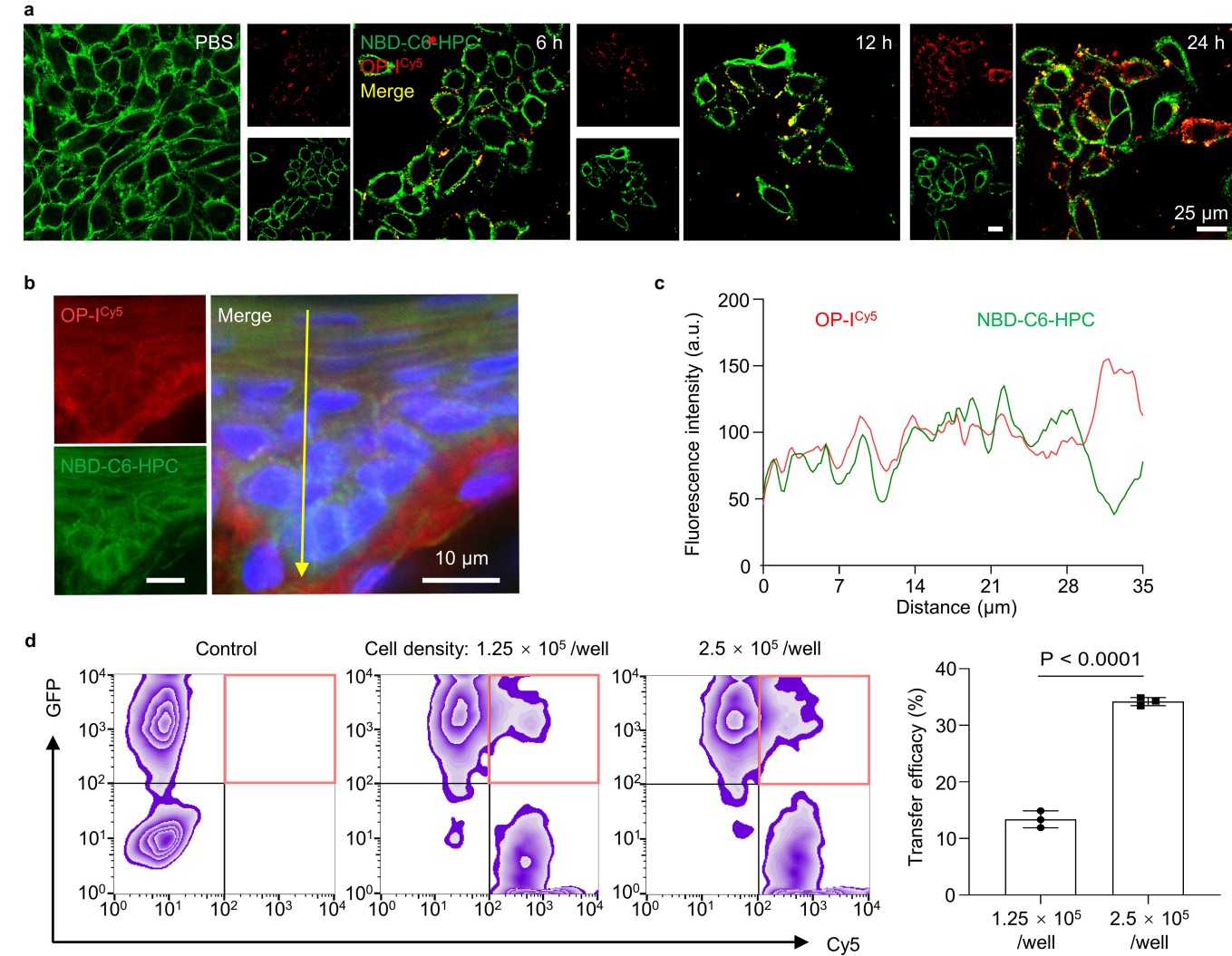

**Extended Data Fig. 8 | Cell-contact-mediated transfer of OP-I between HaCat cells. a**, CLSM imaging of the OP-I$^{Cy5}$ retention in cell membranes of HaCat cells cultured with OP-I$^{Cy5}$ for 6, 12, or 24 h (Cy5-equivalent dose, 1 μg ml$^{-1}$); the cell membrane was stained with NBD-C6-HPC in green. The images are representative of $n = 3$ independent experiments. **b**, Enlarged view of the viable epidermis layer in Extended Data Fig. 6b. Minipig abdominal skin was treated with OP-I$^{Cy5}$ (Cy5-equivalent dose, 10 ml of 10 μg ml$^{-1}$; application area, 100 cm$^2$) via topical application for 4 h. Cy5 fluorescence is shown in red, DAPI in blue, and NBD-C6-HPC fluorescence in green. The images are representative of $n = 3$ independent experiments. **c**, Line-scan analysis of NBD-C6-HPC and Cy5 fluorescence along the yellow arrow in (**b**). **d**, Flow cytometry analysis of the effect of cell density on OP-I$^{Cy5}$ transfer efficiency (Cy5-equivalent dose, 1 μg ml$^{-1}$). The cell densities were 1.25 × 10$^5$ per well or 2.5 × 10$^5$ per well for both cell types (OP-I$^{Cy5}$-pretreated HaCat cells and untreated HaCat$^{GFP}$ cells), with untreated HaCat cells and HaCat$^{GFP}$ cells as controls. Data are mean ± s.d.; $n = 3$ independent experiments. Significance was determined using a two-tailed unpaired Student's t-test.

# Reporting Summary

## Statistics

For all statistical analyses, confirm that the following items are present in the figure legend, table legend, main text, or Methods section.

| n/a | Confirmed | |
|---|---|---|
| ☐ | ☒ | The exact sample size (*n*) for each experimental group/condition, given as a discrete number and unit of measurement |
| ☐ | ☒ | A statement on whether measurements were taken from distinct samples or whether the same sample was measured repeatedly |
| ☐ | ☒ | The statistical test(s) used AND whether they are one- or two-sided<br>*Only common tests should be described solely by name; describe more complex techniques in the Methods section.* |
| ☒ | ☐ | A description of all covariates tested |
| ☒ | ☐ | A description of any assumptions or corrections, such as tests of normality and adjustment for multiple comparisons |
| ☐ | ☒ | A full description of the statistical parameters including central tendency (e.g. means) or other basic estimates (e.g. regression coefficient) AND variation (e.g. standard deviation) or associated estimates of uncertainty (e.g. confidence intervals) |
| ☐ | ☒ | For null hypothesis testing, the test statistic (e.g. *F*, *t*, *r*) with confidence intervals, effect sizes, degrees of freedom and *P* value noted<br>*Give P values as exact values whenever suitable.* |
| ☒ | ☐ | For Bayesian analysis, information on the choice of priors and Markov chain Monte Carlo settings |
| ☒ | ☐ | For hierarchical and complex designs, identification of the appropriate level for tests and full reporting of outcomes |
| ☐ | ☒ | Estimates of effect sizes (e.g. Cohen's *d*, Pearson's *r*), indicating how they were calculated |

*Our web collection on statistics for biologists contains articles on many of the points above.*

## Software and code

Policy information about availability of computer code

| Data collection | GPC traces of samples were obtained by a Shimadzu Prominence Plus LC-20AD liquid chromatography equipped with two columns connected in series (PL aquagel-OH MIXED-H and PL aquagel-OH 30), a refractive index detector, and a UV/VIS detector.<br>RP-HPLC chromatography of samples were acquired by an Agilent 1260 Infinity II system equipped with a ZORBAX SB-C18 column (5 μm, 4.6 x 250 mm), and a 1260 infinity II ariable wavelength detector.<br>Far UV-CD spectra were recorded on a JASCO J-815 spectropolarimeter.<br>Mass spectrometry was obtained on a MALDI-TOF MS (Bruker Daltonics Inc.).<br>Fluorescence microscopy images were acquired by Nikon A1 confocal microscope, total internal reflection fluorescence microscopy (Olympus IX83) and Leica STELLARIS STED confocal microscope with 405-nm, 488-nm, and 640-nm lasers.<br>SEM images were obtained using a Hitichi SU-70 SEM.<br>STEM-HAADFI images and EDS-mapping were obtained using a STEM electron microscope (JEOL JEM-F200).<br>IVIS fluorescence images were acquired by the imaging system (IVIS Lumina XRMS Series III, PerkinElmer).<br>Two-photon fluorescence images were acquired by intravital two-photon microscopy platform (IVM-CMS model, IVIM Technology).<br>GROMACS (2020.6) was used to collect data from molecular dynamics simulations.<br>Flow cytometry data were obtained using a BD FACS Calibur and a CytoFLEX flow cytometer (Beckman Coulter).<br>Blood glucose data were collected by a blood glucose meter (Sinocare).<br>Blood routine data were collected using a Blood Cell Counter.<br>Biochemical indicator data were collected using an Automatic Biochemical Analyzer.<br>SPR experiments were performed on a Biacore X100 instrument (Cytiva).<br>Prep-HPLC was performed using a Waters Prep 150 LC system equipped with a Pursuit 5 C18 column (250 × 21.2 mm) and a Waters 2489 UV/Visible detector.<br>A Nano ZS Zetasizer (Malvern Instruments) was employed to measure zeta potentials. |
|---|---|

Infrared spectrum was collected using a Thermo Scientific™ Nicolet™ iS50 FTIR spectrometer.
Western blot data were obtained by ChemiScope 3600 Mini Imaging System (Clinx Science Instruments Co., Ltd.).
Slices were collected by LEICA CM1950 cryostat (Leica Microsystems, Bannockburn, IL).
The ultrathin sections were prepared using a UC7 ultramicrotome (Leica Microsystems, Bannockburn, IL).
Fluorescence intensities and Optical Density were measured using a SpectraMax M2e microplate spectrophotometer (Molecular Devices).
CHARMM-GUI was used to build the SC lipid membrane in MD simulations.
CGMS15 (Dexcom G4 Platinum Continuous Glucose Monitor System, Dexcom) was used to measure blood glucose levels in minipigs.
Ex vivo transdermal experiments were performed using the Franz diffusion cell (RYJ-12B, Shanghai Huanghai Pharmaceutical Inspection Instrument Co., Ltd.).

| Data analysis |
| --- |

Graphpad Prism (version 10.4.0) was used for data analysis.
Biacore X100 system control software (version 2.0.1.201) was used for SPR data acquisition.
FlowJo (version 10.0) was used for flow cytometry data analysis.
LabSolutions GPC (version 5.111) Software was used for the analysis of GPC data.
NIS-Elements Viewer (version 5.21), Leica Application Suite X (LAS X, version 3.7.5.24914), and Fiji ImageJ (version 2017) was used for fluorescence microscopy image analysis.
Living image®4.5.2 was used for IVIS fluorescence image analysis.
IVIM Studio (version 3.0509) was used for two-photon fluorescence image analysis.
GROMACS (version 2020.6) and VMD (version 1.9.4a51) were used for data analysis from MD simulations.
Omnic (version 9.2.0.41) was used for infrared spectrum analysis.
ChromScope Instrument Edition (version 2.1) was used to process prep-HPLC data.
Zetasizer Software (version 8.01.4906) was used to analyze DLS data.
Clinx ChemiCaptureMini (version 2.1.9.23) was used to analyze data from the chemiluminescence imager.
SoftMax Pro (version 5.4.6) was used to analyze microplate reader data.
OpenLab CDS ChemStation Edition (version C.01.07 SR3) was used to process RP-HPLC data.
DAS2 software (version 2.0)  was used to analyze half-life.
The structures of OP and OP-I were constructed using Avogadro software (version 1.2).

For manuscripts utilizing custom algorithms or software that are central to the research but not yet described in published literature, software must be made available to editors and reviewers. We strongly encourage code deposition in a community repository (e.g. GitHub). See the Nature Portfolio guidelines for submitting code & software for further information.

# Data

Policy information about availability of data

All manuscripts must include a data availability statement. This statement should provide the following information, where applicable:
- Accession codes, unique identifiers, or web links for publicly available datasets
- A description of any restrictions on data availability
- For clinical datasets or third party data, please ensure that the statement adheres to our policy

Data supporting the findings of this study are available in the Source Data file provided with this paper. The input files (.tpr files) for the key simulations in this study are available online (https://zenodo.org/records/17078486; accession code DOI 10.5281/zenodo.17078485).

# Research involving human participants, their data, or biological material

Policy information about studies with human participants or human data. See also policy information about sex, gender (identity/presentation), and sexual orientation and race, ethnicity and racism.

| Reporting on sex and gender | Human research participants are not applicable to this study. |
| --- | --- |
| Reporting on race, ethnicity, or other socially relevant groupings | Human research participants are not applicable to this study. |
| Population characteristics | Human research participants are not applicable to this study. |
| Recruitment | Human research participants are not applicable to this study. |
| Ethics oversight | Human research participants are not applicable to this study. |

Note that full information on the approval of the study protocol must also be provided in the manuscript.

# Field-specific reporting

Please select the one below that is the best fit for your research. If you are not sure, read the appropriate sections before making your selection.

☒ Life sciences    ☐ Behavioural & social sciences    ☐ Ecological, evolutionary & environmental sciences

For a reference copy of the document with all sections, see nature.com/documents/nr-reporting-summary-flat.pdf

# Life sciences study design

All studies must disclose on these points even when the disclosure is negative.

| | |
|---|---|
| Sample size | No sample size calculation was performed. The sample size was chosen based on previous experimental experience or rationals as follow. For in vivo experiments including skin permeability assay, tissue distribution assay, insulin biological half-life assay, lymphatic vessel co-localization assay, and western blot assay, we used at least n = 3 animals per group. Similar sample size was used in previous publications (Zhu, et al, (2024), Nat. Commun.). For in vivo studies in mice including subcutaneous glucose-lowering assay, transdermal glucose-lowering assay, and plasma insulin concentration detection assay, we used at least n = 5 mice per group. Similar sample size was used in previous publications (Han, et al, (2020), Nat. Nanotechnol.; Yu, et al, (2020), Nat. Biomed. Eng.; Yang, et al, (2022), Nat. Commun.). For transdermal glucose-lowering assay in diabetic minipigs, we used n = 3 minipigs. Similar sample size was used in previous publications (Zhang, et al, (2023), Nat. Nanotechnol.). For all in vitro experiments, we performed n = 3 independent experiments. Similar sample size was used in previous publications (Zhou, et al, (2019), Nat. Nanotechnol.; Chen, et al, (2021), Nat. Biomed. Eng.). |
| Data exclusions | There were no data exclusions except those resulting from technical errors making data interpretation impossible. |
| Replication | All experiments were successfully replicated with at least three biological replicates. |
| Randomization | All samples and animals were randomly allocated into experimental groups. |
| Blinding | Blinding was not performed in this study. All experiments used automated instruments and computational analysis, without subjective evaluations prone to investigator bias. |

# Behavioural & social sciences study design

All studies must disclose on these points even when the disclosure is negative.

| | |
|---|---|
| Study description | *Briefly describe the study type including whether data are quantitative, qualitative, or mixed-methods (e.g. qualitative cross-sectional, quantitative experimental, mixed-methods case study).* |
| Research sample | *State the research sample (e.g. Harvard university undergraduates, villagers in rural India) and provide relevant demographic information (e.g. age, sex) and indicate whether the sample is representative. Provide a rationale for the study sample chosen. For studies involving existing datasets, please describe the dataset and source.* |
| Sampling strategy | *Describe the sampling procedure (e.g. random, snowball, stratified, convenience). Describe the statistical methods that were used to predetermine sample size OR if no sample-size calculation was performed, describe how sample sizes were chosen and provide a rationale for why these sample sizes are sufficient. For qualitative data, please indicate whether data saturation was considered, and what criteria were used to decide that no further sampling was needed.* |
| Data collection | *Provide details about the data collection procedure, including the instruments or devices used to record the data (e.g. pen and paper, computer, eye tracker, video or audio equipment) whether anyone was present besides the participant(s) and the researcher, and whether the researcher was blind to experimental condition and/or the study hypothesis during data collection.* |
| Timing | *Indicate the start and stop dates of data collection. If there is a gap between collection periods, state the dates for each sample cohort.* |
| Data exclusions | *If no data were excluded from the analyses, state so OR if data were excluded, provide the exact number of exclusions and the rationale behind them, indicating whether exclusion criteria were pre-established.* |
| Non-participation | *State how many participants dropped out/declined participation and the reason(s) given OR provide response rate OR state that no participants dropped out/declined participation.* |
| Randomization | *If participants were not allocated into experimental groups, state so OR describe how participants were allocated to groups, and if allocation was not random, describe how covariates were controlled.* |

# Ecological, evolutionary & environmental sciences study design

All studies must disclose on these points even when the disclosure is negative.

| | |
|---|---|
| Study description | *Briefly describe the study. For quantitative data include treatment factors and interactions, design structure (e.g. factorial, nested, hierarchical), nature and number of experimental units and replicates.* |
| Research sample | *Describe the research sample (e.g. a group of tagged Passer domesticus, all Stenocereus thurberi within Organ Pipe Cactus National Monument), and provide a rationale for the sample choice. When relevant, describe the organism taxa, source, sex, age range and* |

any manipulations. State what population the sample is meant to represent when applicable. For studies involving existing datasets, describe the data and its source.

| | |
|---|---|
| Sampling strategy | Note the sampling procedure. Describe the statistical methods that were used to predetermine sample size OR if no sample-size calculation was performed, describe how sample sizes were chosen and provide a rationale for why these sample sizes are sufficient. |
| Data collection | Describe the data collection procedure, including who recorded the data and how. |
| Timing and spatial scale | Indicate the start and stop dates of data collection, noting the frequency and periodicity of sampling and providing a rationale for these choices. If there is a gap between collection periods, state the dates for each sample cohort. Specify the spatial scale from which the data are taken |
| Data exclusions | If no data were excluded from the analyses, state so OR if data were excluded, describe the exclusions and the rationale behind them, indicating whether exclusion criteria were pre-established. |
| Reproducibility | Describe the measures taken to verify the reproducibility of experimental findings. For each experiment, note whether any attempts to repeat the experiment failed OR state that all attempts to repeat the experiment were successful. |
| Randomization | Describe how samples/organisms/participants were allocated into groups. If allocation was not random, describe how covariates were controlled. If this is not relevant to your study, explain why. |
| Blinding | Describe the extent of blinding used during data acquisition and analysis. If blinding was not possible, describe why OR explain why blinding was not relevant to your study. |

Did the study involve field work? ☐ Yes ☐ No

## Field work, collection and transport

| | |
|---|---|
| Field conditions | Describe the study conditions for field work, providing relevant parameters (e.g. temperature, rainfall). |
| Location | State the location of the sampling or experiment, providing relevant parameters (e.g. latitude and longitude, elevation, water depth). |
| Access & import/export | Describe the efforts you have made to access habitats and to collect and import/export your samples in a responsible manner and in compliance with local, national and international laws, noting any permits that were obtained (give the name of the issuing authority, the date of issue, and any identifying information). |
| Disturbance | Describe any disturbance caused by the study and how it was minimized. |

# Reporting for specific materials, systems and methods

We require information from authors about some types of materials, experimental systems and methods used in many studies. Here, indicate whether each material, system or method listed is relevant to your study. If you are not sure if a list item applies to your research, read the appropriate section before selecting a response.

### Materials & experimental systems

| n/a | Involved in the study |
|---|---|
| ☐ | ☒ Antibodies |
| ☐ | ☒ Eukaryotic cell lines |
| ☒ | ☐ Palaeontology and archaeology |
| ☐ | ☒ Animals and other organisms |
| ☒ | ☐ Clinical data |
| ☒ | ☐ Dual use research of concern |
| ☒ | ☐ Plants |

### Methods

| n/a | Involved in the study |
|---|---|
| ☒ | ☐ ChIP-seq |
| ☐ | ☒ Flow cytometry |
| ☒ | ☐ MRI-based neuroimaging |

## Antibodies

| | |
|---|---|
| Antibodies used | LYVE1 Conjugated Antibody, Signalway Antibody (C49343-AF488)<br>Rabbit anti-phospho-AKT, Cell Signaling Technology (4056)<br>Rabbit anti-phospho-IR-IGF1R, Cell Signaling Technology (3024)<br>Mouse anti-GAPDH, Proteintech Group, Inc. (60004-1-Ig)<br>Rabbit anti-Insulin Receptor β, Cell Signaling Technology (3025)<br>Rabbit anti-AKT, Proteintech Group, Inc. (10176-2-AP)<br>Anti-histidine antibody, Cytiva Bio-technology Co., Ltd (28995056)<br>Secondary antibodies: |

HRP-labeled Goat Anti-Rabbit IgG (H+L), Beyotime Biotechnology (A0208)
HRP-labeled Goat Anti-Mouse IgG (H+L), Beyotime Biotechnology (A0216)

| Validation | All antibodies used were validated antibody suppliers per quality assurance as detailed on each supplier's website.<br>LYVE1 Conjugated Antibody: https://www.sabbiotech.com/products/C49343.<br>Rabbit anti-phospho-AKT (CST, 4056): https://www.cellsignal.cn/products/primary-antibodies/phospho-akt-thr308-244f9-rabbit-mab/4056.<br>Rabbit anti-phospho-IR-IGF1R (CST, 3024): https://www.cellsignal.cn/products/primary-antibodies/phospho-igf-i-receptor-b-tyr1135-1136-insulin-receptor-b-tyr1150-1151-19h7-rabbit-mab/3024.<br>Mouse anti-GAPDH (60004-1-Ig): https://ptgcn.com/products/GAPDH-Antibody-60004-1-Ig.htm#.<br>Rabbit anti-Insulin Receptor β  (CST, 3025): https://www.cellsignal.cn/products/primary-antibodies/insulin-receptor-b-4b8-rabbit-mab/3025.<br>Rabbit anti-AKT (Proteintech, 10176-2-AP): https://ptgcn.com/products/AKT-Antibody-10176-2-AP.htm.<br>HRP-labeled Goat Anti-Rabbit IgG (H+L): https://www.beyotime.com/product/A0208.htm<br>HRP-labeled Goat Anti-Mouse IgG (H+L): https://www.beyotime.com/product/A0216.htm<br>Anti-histidine antibody: https://www.cytivalifesciences.com.cn/zh/cn/shop/protein-analysis/spr-label-free-analysis/spr-consumables/capture-reagents/his-capture-kit-p-05985.<br>Validated from manufacturer's website. |
|---|---|

# Eukaryotic cell lines

Policy information about cell lines and Sex and Gender in Research

| Cell line source(s) | HaCat cell line was obtained from the Cell Bank of the Chinese Academy of Sciences (Shanghai, China). The cell expressing green fluorescence protein (HaCat-GFP cell) was established by lentivirus transfection of the GFP plasmids into HaCat cell according to the manufacturer's protocol (Shanghai Genechem).<br>Mouse hepatoma cell line (AML-12), mouse embryonic fibroblast cell line (3T3-L1), and mouse skeletal muscle cell line (MSMC) were purchased from Shanghai Zhong Qiao Xin Zhou Biotechnology Co., Ltd. |
|---|---|
| Authentication | Cell lines have not been subjected to additional authentication. |
| Mycoplasma contamination | Cell lines have not been tested for mycoplasma contamination. |
| Commonly misidentified lines<br>(See ICLAC register) | No commonly misidentified cell lines were used. |

# Palaeontology and Archaeology

| Specimen provenance | *Provide provenance information for specimens and describe permits that were obtained for the work (including the name of the issuing authority, the date of issue, and any identifying information). Permits should encompass collection and, where applicable, export.* |
|---|---|
| Specimen deposition | *Indicate where the specimens have been deposited to permit free access by other researchers.* |
| Dating methods | *If new dates are provided, describe how they were obtained (e.g. collection, storage, sample pretreatment and measurement), where they were obtained (i.e. lab name), the calibration program and the protocol for quality assurance OR state that no new dates are provided.* |

☐ Tick this box to confirm that the raw and calibrated dates are available in the paper or in Supplementary Information.

| Ethics oversight | *Identify the organization(s) that approved or provided guidance on the study protocol, OR state that no ethical approval or guidance was required and explain why not.* |
|---|---|

Note that full information on the approval of the study protocol must also be provided in the manuscript.

# Animals and other research organisms

Policy information about studies involving animals; ARRIVE guidelines recommended for reporting animal research, and Sex and Gender in Research

| Laboratory animals | C57BL/6J mice, male, 6-8 weeks old, 25 g; SD rats, female, 8-12 weeks old, 200 g; Guangxi Bama-minipigs, male, 6 months old, 35-40 kg. |
|---|---|
| Wild animals | No wild animals were used in the study. |
| Reporting on sex | Male mice, male minipigs, and female rats were used in this study. |
| Field-collected samples | No field collected samples were used in the study. |

| Ethics oversight | All animal experiments were carried out according to the protocols approved by the Institutional Animal Care and Use Committee of Zhejiang University (Approval No. ZJU20250230). |
|---|---|

Note that full information on the approval of the study protocol must also be provided in the manuscript.

# Clinical data

Policy information about clinical studies
All manuscripts should comply with the ICMJE guidelines for publication of clinical research and a completed CONSORT checklist must be included with all submissions.

| Clinical trial registration | *Provide the trial registration number from ClinicalTrials.gov or an equivalent agency.* |
|---|---|
| Study protocol | *Note where the full trial protocol can be accessed OR if not available, explain why.* |
| Data collection | *Describe the settings and locales of data collection, noting the time periods of recruitment and data collection.* |
| Outcomes | *Describe how you pre-defined primary and secondary outcome measures and how you assessed these measures.* |

# Dual use research of concern

Policy information about dual use research of concern

## Hazards

Could the accidental, deliberate or reckless misuse of agents or technologies generated in the work, or the application of information presented in the manuscript, pose a threat to:

No | Yes
- [ ] | [ ] Public health
- [ ] | [ ] National security
- [ ] | [ ] Crops and/or livestock
- [ ] | [ ] Ecosystems
- [ ] | [ ] Any other significant area

## Experiments of concern

Does the work involve any of these experiments of concern:

No | Yes
- [ ] | [ ] Demonstrate how to render a vaccine ineffective
- [ ] | [ ] Confer resistance to therapeutically useful antibiotics or antiviral agents
- [ ] | [ ] Enhance the virulence of a pathogen or render a nonpathogen virulent
- [ ] | [ ] Increase transmissibility of a pathogen
- [ ] | [ ] Alter the host range of a pathogen
- [ ] | [ ] Enable evasion of diagnostic/detection modalities
- [ ] | [ ] Enable the weaponization of a biological agent or toxin
- [ ] | [ ] Any other potentially harmful combination of experiments and agents

# Plants

| Seed stocks | This study does not include plants. |
|---|---|
| Novel plant genotypes | This study does not include plants. |
| Authentication | This study does not include plants. |

# ChIP-seq

## Data deposition

☐ Confirm that both raw and final processed data have been deposited in a public database such as GEO.

☐ Confirm that you have deposited or provided access to graph files (e.g. BED files) for the called peaks.

| | |
|---|---|
| Data access links<br>*May remain private before publication.* | *For "Initial submission" or "Revised version" documents, provide reviewer access links. For your "Final submission" document, provide a link to the deposited data.* |
| Files in database submission | *Provide a list of all files available in the database submission.* |
| Genome browser session<br>(e.g. UCSC) | *Provide a link to an anonymized genome browser session for "Initial submission" and "Revised version" documents only, to enable peer review. Write "no longer applicable" for "Final submission" documents.* |

## Methodology

| | |
|---|---|
| Replicates | *Describe the experimental replicates, specifying number, type and replicate agreement.* |
| Sequencing depth | *Describe the sequencing depth for each experiment, providing the total number of reads, uniquely mapped reads, length of reads and whether they were paired- or single-end.* |
| Antibodies | *Describe the antibodies used for the ChIP-seq experiments; as applicable, provide supplier name, catalog number, clone name, and lot number.* |
| Peak calling parameters | *Specify the command line program and parameters used for read mapping and peak calling, including the ChIP, control and index files used.* |
| Data quality | *Describe the methods used to ensure data quality in full detail, including how many peaks are at FDR 5% and above 5-fold enrichment.* |
| Software | *Describe the software used to collect and analyze the ChIP-seq data. For custom code that has been deposited into a community repository, provide accession details.* |

# Flow Cytometry

## Plots

Confirm that:

☒ The axis labels state the marker and fluorochrome used (e.g. CD4-FITC).

☒ The axis scales are clearly visible. Include numbers along axes only for bottom left plot of group (a 'group' is an analysis of identical markers).

☒ All plots are contour plots with outliers or pseudocolor plots.

☒ A numerical value for number of cells or percentage (with statistics) is provided.

## Methodology

| | |
|---|---|
| Sample preparation | For direct transfer experiment, the OP-ICy5 treated HaCat cells were extensively rinsed with sterilized PBS, isolated, and mixed with untreated HaCat-GFP cells at the same cell density. The mixed cells were co-cultured further 24 h. The cells were washed twice with PBS and analyzed by flow cytometry. Flow cytometry was performed on a BD FACS Calibur, and analysis was performed using the FlowJo software. |
| Instrument | Flow cytometry data were obtained using a BD FACS Calibur and a CytoFLEX flow cytometer (Beckman Coulter). |
| Software | Data analysis was performed using the FlowJo v10.0. |
| Cell population abundance | Cell sorting was not performed. |
| Gating strategy | First, an FSC-A vs. SSC-A gate was used to exclude debris and select for live cells based on their forward scatter (FSC) and side scatter (SSC) characteristics. Next, a GFP gate (FL1-H) was set to include cells that exhibit fluorescence in the GFP channel. Subsequently, a Cy5 gate (FL4-H ) was established to isolate cells that are positive for Cy5 fluorescence. Finally, an intersection gate was applied between the GFP and Cy5 gates to identify cells that are positive for both GFP and Cy5, representing the dual-stained population. |

☒ Tick this box to confirm that a figure exemplifying the gating strategy is provided in the Supplementary Information.

# Magnetic resonance imaging

## Experimental design

**Design type**
*Indicate task or resting state; event-related or block design.*

**Design specifications**
*Specify the number of blocks, trials or experimental units per session and/or subject, and specify the length of each trial or block (if trials are blocked) and interval between trials.*

**Behavioral performance measures**
*State number and/or type of variables recorded (e.g. correct button press, response time) and what statistics were used to establish that the subjects were performing the task as expected (e.g. mean, range, and/or standard deviation across subjects).*

## Acquisition

**Imaging type(s)**
*Specify: functional, structural, diffusion, perfusion.*

**Field strength**
*Specify in Tesla*

**Sequence & imaging parameters**
*Specify the pulse sequence type (gradient echo, spin echo, etc.), imaging type (EPI, spiral, etc.), field of view, matrix size, slice thickness, orientation and TE/TR/flip angle.*

**Area of acquisition**
*State whether a whole brain scan was used OR define the area of acquisition, describing how the region was determined.*

**Diffusion MRI**  ☐ Used   ☐ Not used

## Preprocessing

**Preprocessing software**
*Provide detail on software version and revision number and on specific parameters (model/functions, brain extraction, segmentation, smoothing kernel size, etc.).*

**Normalization**
*If data were normalized/standardized, describe the approach(es): specify linear or non-linear and define image types used for transformation OR indicate that data were not normalized and explain rationale for lack of normalization.*

**Normalization template**
*Describe the template used for normalization/transformation, specifying subject space or group standardized space (e.g. original Talairach, MNI305, ICBM152) OR indicate that the data were not normalized.*

**Noise and artifact removal**
*Describe your procedure(s) for artifact and structured noise removal, specifying motion parameters, tissue signals and physiological signals (heart rate, respiration).*

**Volume censoring**
*Define your software and/or method and criteria for volume censoring, and state the extent of such censoring.*

## Statistical modeling & inference

**Model type and settings**
*Specify type (mass univariate, multivariate, RSA, predictive, etc.) and describe essential details of the model at the first and second levels (e.g. fixed, random or mixed effects; drift or auto-correlation).*

**Effect(s) tested**
*Define precise effect in terms of the task or stimulus conditions instead of psychological concepts and indicate whether ANOVA or factorial designs were used.*

**Specify type of analysis:**  ☐ Whole brain   ☐ ROI-based   ☐ Both

**Statistic type for inference**
*Specify voxel-wise or cluster-wise and report all relevant parameters for cluster-wise methods.*

(See Eklund et al. 2016)

**Correction**
*Describe the type of correction and how it is obtained for multiple comparisons (e.g. FWE, FDR, permutation or Monte Carlo).*

## Models & analysis

| n/a | Involved in the study |
|---|---|
| ☐ | ☐ Functional and/or effective connectivity |
| ☐ | ☐ Graph analysis |
| ☐ | ☐ Multivariate modeling or predictive analysis |

**Functional and/or effective connectivity**
*Report the measures of dependence used and the model details (e.g. Pearson correlation, partial correlation, mutual information).*

**Graph analysis**
*Report the dependent variable and connectivity measure, specifying weighted graph or binarized graph,*

Graph analysis

*subject- or group-level, and the global and/or node summaries used (e.g. clustering coefficient, efficiency, etc.).*

Multivariate modeling and predictive analysis

*Specify independent variables, features extraction and dimension reduction, model, training and evaluation metrics.*

