## [Peer Review File · Nature]

A skin-permeable polymer for noninvasive transdermal insulin delivery

Corresponding Author: Professor Rongjun Chen

Version 0:

Reviewer comments:

Referee #2

(Remarks to the Author)

In this study, Wei et al. investigated the transdermal permeation of peptides and proteins using the zwitterionic polymer poly[2-(N-oxide-N,N-dimethylamino)ethyl methacrylate] (OP). The findings revealed that both OP and OP conjugated with insulin (OP-I) demonstrated high transdermal permeation efficiency. The plasma concentration achieved by transdermal delivery of OP-I was comparable to that of insulin administered via injection, with superior blood glucose level control efficacy. This research represents one of the few comprehensive analyses of the transdermal delivery mechanisms for hydrophilic substances, providing valuable insights into this field. The study presents a novel approach with systematic experimentation and analysis, supported by appropriate references. However, the following points need to be addressed:

- The introduction lacks sufficient detail. It should include a rationale for choosing OP among various zwitterionic polymers as the delivery material.
- The source of the STEM and EDS data for the OP-AuNP conjugate within the skin is unclear, and data at the sub-hundred nanometer scale are insufficient to validate skin permeation. Multiple data points from different skin regions are necessary.
- In Figure 3, the quantities of insulin administered transdermally and by injection differ. This has implications for cost-efficiency, a crucial factor in therapeutic application, and warrants discussion.
- Figures 3a and 3b show similar plasma concentrations for OP-I and injected insulin (s.c.), yet there is a significant difference in blood glucose levels. This discrepancy needs an explanation.
- Figure 4 indicates stronger binding between OP/OP-I and SC lipids, yet better diffusion is observed. A more convincing explanation for this phenomenon is needed.
- Figure 5 suggests that substances "hopped" between cells, but Movie 4 shows fluorescence mostly remaining stationary. Additionally, while the substance appears to penetrate the skin and enter the bloodstream within 2 hours, the experiments in Figure 5 observe phenomena on a scale of over 12 hours, reducing the persuasiveness of the results. Supplementary experiments are required to address these inconsistencies.

Referee #3

(Remarks to the Author)

All individuals with type 1 diabetes and many with type 2 diabetes require insulin to control their hyperglycemia. For over 90 years insulin had to be given by injection. Multiple daily injections are not acceptable to all patients and a search for a non-injectable form of insulin has been pursued for many years. Inhaled forms of insulin have been available in some countries since 2006, but it has not been widely used, in large part due to the potentially severe side effects.

In this study, the authors describe the attachment of a skin-permeable polymer to insulin for transdermal delivery. They synthesise a soluble zwitterionic polymer, poly[2-(N-oxide-N,N-dimethylamino)ethyl methacrylate] (OP), which they conjugate to insulin (termed OP-I). They demonstrate that OP-I in a cream penetrates the skin of mice and minipigs and investigate the mechanism of penetration. OP-I entered the blood in both animals and reduced blood glucose concentrations. The authors conclude that OP-I could be used in individuals with diabetes.

A transdermal therapy for diabetes would be important for the care of patients with this disease. Nevertheless, the study has several deficiencies and requires considerably more work.

Specific comments:

1. A major deficiency is the very limited analysis of the biological effects of OP-I. The vast majority of the data presented use models and microscopy to focus on the penetration of OP and OP-I through the skin and tissues of the animals. Considerably more characterization of the biological effects of OP-I is needed. Analyses include, but are not limited to, the potential effects of the conjugate on binding to insulin receptors, receptor activation and signalling pathways. In addition, specificity for binding to insulin receptors and pharmacokinetics need to be studied.
2. The number of replicates should be given in the legends for all figures. This is essential to evaluate reproducibility of the data.
3. Patients are treated with several forms of insulin with different durations of action. Which insulin was used in this study?
4. Fig. 2b. How do you explain the highest OP-I at 24 h when peak blood insulin concentrations are at 2 h?
5. Fig. 2. What is the biological relevance of OP-I in the lymphatics? While insulin has effects on all organs in humans, the main target organs for glucose homeostasis and fat metabolism are adipose tissue, skeletal muscle and the liver. What was OP-I distribution in fat and skeletal muscle? Can OP-I enter human fat, liver and skeletal muscle cells? Insulin secreted by the pancreas undergoes first pass metabolism in the liver, resulting in a very short (3-10 min) biological half-life for secreted insulin. What is the half-life of OP-I?
6. Fig 2d. The Merged image does not reflect individual images. There is no mention of rats in Methods. Scale bars are needed in all microscopic images.
7. Fig S4. What is the in vivo relevance of OP-I in the skin at 4 h? how long does it remain in the skin? How long is OP-I biologically active?
8. The authors extensively characterise skin penetration of OP and OP-I. Patients inject insulin into the subcutaneous fat (not intradermally, as stated line 162). How long does OP-I take to reach the subcutaneous fat?
9. Fig. 3. How was the dose of OP-I selected? Patients are very careful about the dose of insulin they inject. Please show the reproducibility of topical application of OP-I on plasma insulin concentration. Variation among different individuals applying OP-I should be shown. Does plasma insulin concentration vary with the site of topical application?
10. Fig. 3. The goal of insulin therapy in patients with diabetes is to get glucose concentrations as close to normal as feasible. Hypoglycemia is the most feared complication of insulin administration. It is concerning that OP-I produced a prolonged reduction in glucose. What was the nadir glucose concentration? The authors should provide the reference interval ("normal range") for the mouse strain used to facilitate data interpretation. In panel c, the glucose concentrations at 90 and 120 min appear very low. How long did this effect persist?
11. Fig. 3. Topical PEG-I and insulin reduced glucose, despite not being absorbed. Explanation?
12. Considerably more experimental detail should be provided for all mouse and minipig experiments. Were animals fed/fasted? If the latter, how long?
13. Insufficient numbers of animals are used in the experiments. At least 7-10 are needed. This is particularly important in light of the large variability in fig S6. What is the reason for the large (~2-fold) variability at t=0 in Fig S6a?
14. The units on the Y-axis in S6b appear incorrect. The Y-axis units in S6c are confusing.
15. Considerably more experimental detail is needed in Methods and legends.
16. Statistical analysis should be performed wherever feasible.

Version 1:

Reviewer comments:

Referee #2

(Remarks to the Author)

The authors have thoroughly addressed the comments raised by this reviewer. I believe the significance of this study is high, as it presents a well-executed investigation that not only enables the transdermal delivery of insulin but also provides a detailed explanation of the transdermal delivery mechanism—an aspect that, to the best of this reviewer's knowledge, has not been effectively explained in previous studies.

Although this is a minor issue, I suggest that the authors consider including a more detailed summary of their findings in the conclusion section, which would enhance the overall structure and clarity of the manuscript.

Referee #3

(Remarks to the Author)

This is a revised and improved manuscript. An effective form of insulin that could be applied topically to the skin for noninvasive transdermal uptake is likely to appeal to individuals with diabetes who require insulin. Moreover, the strategy adopted by the authors could potentially be applied to other parenteral therapeutics.

The authors have performed additional experiments that adequately addressed some of my prior concerns. A few items require attention.

1. The SPR data do not address the question of specificity for binding to insulin receptors. In addition, OP-1 pharmacokinetics should be determined as previously requested.
2. Extended Fig. 2f. The authors should examine the effect on insulin signalling of OP-I applied to the skin to show that it is

effective when used as intended. Better quality Western blots should be shown. Total IR and Akt should be shown and the extent of phosphorylation should be corrected for the total amount of protein in the same sample. The analysis should be repeated in several (≥ 3) independent experiments to validate reproducibility, data should be quantified and the results compared to regular insulin given sq.

3. Supp Fig 7. Statistical analysis should be performed to compare the half-life of OP-I and insulin to support the authors' "marginally longer" claim in the text.
4. Fig. 3. The authors should explain in the legend what the shaded area represents in panels c, d, f, g & i.
5. Fig. 3c, Suppl Fig 15. Statistical analysis should be performed and included to compare the effect of OP-I and sq insulin. The data in 3c do not support the claim that topical insulin and PEG-I had "neglectable [sic] impact on BGL", eg, insulin at 4 h.
6. P. 6, line 212. What is meant by "...caused almost no significant changes in...BGL"? Was statistical analysis performed?
7. Fig. 3i. In the Methods, the authors state 3 minipigs were injected with STZ. However, in 3i they have 5 different assay conditions for each pig. Explanation?
8. Fig. 3i and Suppl Fig 18. No data are shown to support the claim of "over 12 h". It is important to determine the duration of action to avoid hypoglycemia.
9. Suppl Fig 15. The scale of Y-axis at 2 h appears different to those at the other time points.

Referee #4

(Remarks to the Author)

The authors have designed an insulin 1:1 conjugate that seems to enable efficient delivery of insulin across the skin. It is claimed that the polymer is permeating the skin into the blood circulation, due to the polyzwitterion structure leading to change in net charge at neutral pH. Uptake pathway is discussed to be related to improved diffusion across epidermis via cell-to-cell membrane diffusion and subsequently drained via the lymphatic vessels to the systemic circulation. The first revision led to an improved manuscript.

The work is original and with an interesting compound, promising results. May indeed be of high significance if useful for other compounds, and thus more generally applicable.

A variety of methods (in silico, in vitro, in vivo) have been employed to measure and understand different aspects of the use of the polymer and its properties. Methods descriptions are at some not detailed enough. Questions about the design of different experiments. Some descriptions are not sufficient. Not always clear what the number of replicates refers to. Some details and controls are missing to truly understand how this compound delivers (in this case) insulin, and to allow for more specific comments and questions below.

- Control in the form of free Cy5 (or as close to this as possible) should be included – to ensure that the tracking both in vivo and in vitro were not (partly) due to the label detaching from the polymer. The authors should comment on the risk of quantifying PK only on the label.
- What was the stability of the labelled conjugate over time and under in vivo-like conditions?
- The samples from PK studies and permeation studies were analyzed for presence of insulin by ELISA.
- From the methods description it seems as if the PK and PD was done in the same animals, but why are PK studies the reported as n=8 and PD as n=5? (e.g. figure 3) – was it not the same animals that are used for these assessments?
- In the methods section for the PK/PD, the dosing of the s.c. should be described, including the actual dose given.
- Please explain why the dose of insulin s.c. used is so high - 5 U/kg? Typically, rats are dosed 1, max 2 U/kg and (healthy) mice around 0.5–1 U/kg (although varies a lot in literature). Were other doses tested? These might give similar profiles to the one observed for 5 U/kg.
- The skin was prepared using depilatory cream – was it controlled that this cream/procedure did not induce any changes in the skin permeability barrier?
- What did the controls look like in terms of e.g. the OP + label and/or insulin co-administered in terms of blood glucose (and absorption)?
- Why are rats/pigs used in some experiments for imaging (e.g. Figure 2) and not mice?

Why is the in vivo proof of concept done in one animal species and the compatibility assessment and MoA studies in other species?

- The methods description of the minipigs studies only describe how the dosing was done, not how samples were taken and analyzed.
- It is stated that compatibility studies are done to investigate morphological alterations and e.g. immune responses, in mice exposed to the OP-I but it is not clearly described how, nor are data shown.
- Abstract, line 31: Rephrase. "As efficiently as s.c. injected insulin" – an overstatement. While impressive BGC lowering effects, the BA/Fab should be included in statements like this. The topical doses are (for good reasons) x times higher than s.c., which should be accounted for.
- P4, line 127-129: the authors claim that the extended circulation time/half-life of the conjugate is due to non-stickiness to plasma proteins, a was also discussed in ref. 23. However, it is a well-known strategy to extend circulating time due to "stickiness" to plasma proteins. The authors should reflect on this.
- P5, line 157-158: ...adipose tissue labelled with BODIPY.... Please clarify if it is Ab-labelling or ? Also: Why was the PEG-I and insulin FITC-labelled and the OP-I Cy5 labelled (extended data, fig 3)? This can be important for the interpretation of images as FITC fades much faster than Cy5. The authors should address this in their interpretation of images. It seems that OP-I FITC was also used for some studies (figure 2), why switch between labels?
- P6, line 173: typo. Correct plasm to plasma
- P20, line 742: change transdermally to topically (or cutaneously)
- WB: It is argued that the downstream pAKT is activated, and while the pIR is not activated by PBS control injections,

this seems to be the case for pAKT (Extended Data Fig. 2f). Please explain.

- MD: If the polymer is positively charged at < 7.4 , one would expect that it would bind/assemble insulin with a pI of 5.3 and in that way form a depot in the skin. Any reflections on this? Were the simulations done with more molecules in the box to investigate this?
- Figure 4: What do $n=3$ mean for these results. The images are representative of 3? Are the MD done three times and the shown is a representative?
- Figure 5d: How do the buffer control and membrane-stained cells look? From the membrane stain, the cells seem to exist not in a confluent monolayer but rather as individual cells, and the membrane of many of them seems quite thick (is it maximum projection intensities that is displayed?).
- Figure 5f: how is the relative transfer efficacy calculated from the flow cytometry data? Which gating was used? Some explanation is given in the methods description but it is not clear how the authors get three times higher transfer after 12 h.
- It is concluded that the polymer-Insulin conjugate does not affect the morphology of the skin cells, nor cause any inflammation or cell death. But does this polymer fluidize the skin or in another way change the barrier to absorption of other compounds? It would be highly beneficial to demonstrate that the enhanced permeation was specific for the conjugated insulin.
- A question of whether co-administration instead of conjugation would have any effect on skin permeation in vitro/ex vivo and absorption in vivo? The authors should discuss this, and data show the benefit of conjugating.
- It is claimed that the approach is likely viable also for other macromolecular drugs, however, it should be considered that the conjugates are essentially new drugs, which might limit the general applicability.

Version 2:

Reviewer comments:

Referee #2

(Remarks to the Author)

This reviewer believes that the authors have adequately addressed all of the comments and concerns raised in this reviewer's previous report. Provided that the issues highlighted by the other reviewers are also satisfactorily resolved, this reviewer considers the manuscript suitable for publication.

Referee #3

(Remarks to the Author)

My comments have been adequately addressed.

Referee #4

(Remarks to the Author)

The authors have addressed my earlier comments; to quite some extent this needed conduction of new/additional experiments to answer the questions. Most of these are now included in the manuscript and SI. Overall, the findings and earlier claims are substantiated, and the manuscript is improved.

However, I have requests to some of the comments:

Re. response to comment 4): the argument not to stress the animals and by that affect the PD readout is good, however if analysis is done using the same blood samples, it should not stress the animals. I think it should be mentioned that it was indeed done in different animals, since the PK and PD profiles in fig 3 do not correspond - the PD seem much more extended than what is reflected in the PK profile.

Re. response to comment 6): The authors have conducted studies with both 1 U/kg and 5 U/kg insulin dosed s.c. These data should be included in the SI. They are important to show that there is not a huge difference in the AOC of the two doses, and for sure not a five-fold increase with the higher dose compared to the lower dose. There could be many reasons for this, for example that the insulin (especially in the higher concentration formulation) partly aggregate and thus does not exist in free form to be absorbed from the site of injection to the blood stream. Although the authors do not underline bioavailability data of their conjugate, the dosed conc. of the reference (here s.c.) will affect the bioavailability hugely. Indeed, the authors show - in response to comment 18 - that the by MD simulations, the insulin molecules aggregate. It is of course known that insulin assembles into multimers, that this is affected by surrounding factors and dose etc. and potentially larger aggregates.

Re. response to comment 13): the word nonstickness is unspecific and somewhat non-scientific. Can this be specified?

Re. response to comment 20): So the images are maximum projection images, it seems. This should be noted.

Re. response to comment 24): Sounds promising, but is this an approach much different from the control that I asked for in comment 23?

RESPONSE TO REFEREES' COMMENTS

Nature Submission ID: 2024-05-10937

Title: A skin-permeable polymer for noninvasive transdermal insulin delivery

Authors: Qiuyu Wei, Zhi He, Zifan Li, Zhuxian Zhou, Ying Piao, Jianxiang Huang, Yu Geng, Runnan Zhang, Yaqi Fu, Jiayi Ye, Yue Yuan, Haoru Zhu, Jiaheng Zeng, Yan Zhang, Quan Zhou, Mingyu Xu, Shiqun Shao, Jianbin Tang, Jiajia Xiang, Rongjun Chen, Ruhong Zhou & Youqing Shen

We would like to express our sincere gratitude to the referees for their helpful and constructive suggestions, which we believe have greatly improved the revised version. We have taken careful note of their comments and suggestions, carried out further experiments, and revised our manuscript accordingly to include the critical new information and data.

A point-by-point response to all referees' comments is given in blue typescript alongside their comments below. We have highlighted the changes in our revised manuscript in blue typescript.

Referee #2:

In this study, Wei et al. investigated the transdermal permeation of peptides and proteins using the zwitterionic polymer poly[2-(N-oxide-N,N-dimethylamino)ethyl methacrylate] (OP). The findings revealed that both OP and OP conjugated with insulin (OP-I) demonstrated high transdermal permeation efficiency. The plasma concentration achieved by transdermal delivery of OP-I was comparable to that of insulin administered via injection, with superior blood glucose level control efficacy. This research represents one of the few comprehensive analyses of the transdermal delivery mechanisms for hydrophilic substances, providing valuable insights into this field. The study presents a novel approach with systematic experimentation and analysis, supported by appropriate references. However, the following points need to be addressed:

Thank you for your positive comments on our work.

Comment 1) The introduction lacks sufficient detail. It should include a rationale for choosing OP among various zwitterionic polymers as the delivery material.

Authors' response: Thank you for your comment. Following your suggestion, we have added the rationale for selecting OP among zwitterionic polymers in the revised manuscript (Page 2; Lines 51-67), as follows:

“One may imagine that a skin-permeable material is required to concentrate on the skin surface and then efficiently diffuse through the hydrophobic inter-corneocyte lipid matrix of the SC into the hydrophilic VE¹⁶. Cationic peptides, which can electrostatically bind to the negatively charged alkyl carboxylic acids in the sebum and SC, have been tested for transdermal delivery, and some have been reported to be “skin-permeable”^{17,18}. However, such skin permeation is actually not *via* diffusion through the inter-corneocyte because the strong binding immobilizes them in the SC without diffusion, but through the appendageal paths¹⁹, including hair follicles and sweat glands, and thus inefficient in humans because the appendageal areas occupy less than 0.1% of the human skin area^{20,21}.

We thus propose that a polymer capable of transitioning from a polycation that binds to the skin SC surface to a polyzwitterion in the deeper SC layers for free diffusion would be skin-permeable. Inspired by the skin's characteristic acidic (pH~5)-to-neutral pH gradient from the sebum layer and

SC surface to the deeper layers of the SC and VE²², we further propose a highly water-soluble polyzwitterion, poly[2-(*N*-oxide-*N,N*-dimethylamino)ethyl methacrylate] (OP)²³, to be skin-permeable. OP was protonated and positively charged at pH 5 or lower and deprotonated to a polyzwitterion at the neutral pH (Supplementary Fig. 1). This alignment of OP's pH-dependent charge-transition with the skin pH-gradient can render it skin-surface binding and efficient diffusion through the inter-corneocyte lipid matrix and thus high skin-permeability (Scheme 1).”

Supplementary Fig. 1 | The pH-dependent zeta-potentials of OP (0.1 mg/mL) and OP-I (0.04 mg/mL) in HEPES buffer at different pH. Data are mean \pm SD of biological replicates (n = 3).

Scheme 1 | Schematic of the skin penetration mechanism of OP and its conjugate with insulin (OP-I).

Comment 2) The source of the STEM and EDS data for the OP-AuNP conjugate within the skin is unclear, and data at the sub-hundred nanometer scale are insufficient to validate skin permeation. Multiple data points from different skin regions are necessary.

Authors' response: Thank you for your comment. We have supplemented a large view of the SC region to clear this concern, as shown in Supplementary Fig. 4b.

Supplementary Fig. 4b | The HAADF-STEM image of the SC of the mouse skin after 4 h of topical application with OP-AuNPs (OP-eq. dose: 0.2 mL of 0.5 mg/mL; application area: 1.13 cm²).

Comment 3) In Figure 3, the quantities of insulin administered transdermally and by injection differ. This has implications for cost-efficiency, a crucial factor in therapeutic application, and warrants discussion.

Authors' response: Thank you for pointing this out. We agree that cost-efficiency is a crucial consideration for clinical translation.

In this study, we intend to demonstrate the first efficient skin (particularly stratum corneum)-permeable polymer OP for effective transdermal delivery, using insulin as a model drug to show that OP could percutaneously deliver sufficient amount to exert similar pharmacodynamics as the subcutaneous one.

The initial dose of OP-I (116 U/kg) was selected to ensure sufficient insulin delivery into the bloodstream to achieve a hypoglycemic effect comparable to subcutaneous injection (Fig. 3c). So, at this proof-of-concept stage, the dose was indeed not optimized for cost-efficiency.

Following your suggestion, we conducted additional experiments using reduced doses of OP-I (58 U/kg and 29 U/kg, corresponding to 10 and 5 times of the subcutaneous dose, respectively). The results showed that normoglycemia could be achieved within 4 h for 58 U/kg and within 6 h for 29 U/kg, compared to 2 h at 116 U/kg (Fig. 3d). These findings indicate that lower doses of OP-I are feasible and effective, albeit a slightly slower onset of action.

It is worth noting that noninvasive delivery methods, such as transdermal or oral administration, often require higher doses than subcutaneous injection due to the multiple physiological barriers involved. For example, the clinically approved oral semaglutide requires doses 30-50 times higher than its subcutaneous counterpart to achieve comparable efficacy (Lancet, 2023, 402, 693-704). Despite the higher doses, the significant clinical advantages of noninvasive methods, such as enhanced patient convenience and compliance, often outweigh this limitation. Additionally, compared to insulin microneedles, which necessitate doses exceeding 100 times higher than those required for subcutaneous injections (Nat. Biomed. Eng., 2020, 4, 499; Sci. Adv., 2022, 8, eadd3197; Proc. Natl. Acad. Sci. U.S.A., 2020, 117, 29512), the dosage of OP-I was comparatively moderate.

We will further improve the cost-efficiency of OP-I in the future work by optimizing the polymer-to-insulin ratio, developing long-acting patches, and incorporating skin permeability enhancers.

We have incorporated these new findings and discussions into the revised manuscript (Page 6, Lines 190-192).

Fig. 3c | Blood glucose levels (BGLs) of STZ-induced diabetic mice after different treatments as indicated in (a); n = 8.

Fig. 3d | BGLs in the diabetic mice after topical application of lower doses of OP-I (insulin-eq. dose: 58 or 29 U/kg; 0.05 mL or 0.1 mL of 0.5 mg/mL solution applied on 1.13 cm² dorsal skin); n = 8.

Comment 4) Figures 3a and 3b show similar plasma concentrations for OP-I and injected insulin (s.c.), yet there is a significant difference in blood glucose levels. This discrepancy needs an explanation.

Authors' response: Thank you for your comment. Following your suggestion, we conducted additional experiments to show that the observed discrepancy is attributed to the differences in their pharmacokinetics and biodistribution.

While OP-I exhibited comparable plasma concentrations to s.c. injected insulin during the initial 2 h post-administration, its plasma levels remained consistently higher than s.c. injected insulin

beyond this point by 1.6 to 6 folds (Fig. 3a).

Furthermore, topical OP-I was effectively absorbed into the circulation and subsequently accumulated in glucose-regulating tissues, including the liver (Supplementary Fig. 9), adipose tissues (Supplementary Fig. 10), and skeletal muscles (Supplementary Fig. 11). These tissues are critical for insulin-mediated glucose uptake, glycogenesis, and suppression of gluconeogenesis (Nat. Rev. Mol. Cell Biol., 2021, 22, 751). In contrast, *s.c.* injected insulin was rapidly cleared from the bloodstream, with minimal accumulations in adipose and muscle tissues.

We have included these explanations in the revised manuscript (Page 6, Lines 174-181) and added the additional results in the revised Supplementary Information.

Fig. 3a | Plasma insulin concentrations in STZ-induced diabetic mice after topical application of PBS, insulin, PEG-I, or OP-I (insulin-eq. dose: 116 U/kg; 0.2 mL of 0.5 mg/mL solution applied on 1.13 cm² dorsal skin); n = 5. Mice were subcutaneously (*s.c.*) injected with native insulin (5 U/kg) as a positive control.

Supplementary Fig. 9 | Fluorescence images of the major organs, including heart (H), liver (L), spleen (S), lungs (Lu), and kidneys (K), of the mice after topical administration of OP-I^{Cy5}, PEG-I^{Cy5}, or insulin^{Cy5} solutions (Cy5-eq. dose: 0.2 mL of 10 µg/mL; application area: 1.13 cm²) on the dorsal skin, or after subcutaneous (*s.c.*) injection of insulin^{Cy5} (Cy5-eq. dose: 50 µL of 25 µg/mL); n = 3.

Supplementary Fig. 10 | a,b, Fluorescence images of mouse adipose tissues including brown adipose (BAT), subcutaneous white adipose (iWAT), and visceral white adipose (eWAT) after topical administration of OP-ICy5 (Cy5-eq, dose: 0.2 mL of 10 $\mu\text{g}/\text{mL}$; application area: 1.13 cm^2) on the dorsal skin (a) or *s.c.* injection of insulin^{Cy5} (Cy5-eq, dose: 50 μL of 25 $\mu\text{g}/\text{mL}$) (b); $n = 3$.

Supplementary Fig. 11 | a,b, Fluorescence images of the mouse forelimb muscle (FLM) and hindlimb muscle (HLM) after topical administration of OP-I^{Cy5} (Cy5-eq, dose: 0.2 mL of 10 µg/mL; application area: 1.13 cm²) on the dorsal skin (a) or *s.c.* injection of insulin^{Cy5} (Cy5-eq, dose: 50 µL of 25 µg/mL) (b); n = 3.

Comment 5) Figure 4 indicates stronger binding between OP/OP-I and SC lipids, yet better diffusion is observed. A more convincing explanation for this phenomenon is needed.

Authors' response: Thank you for pointing out this seemingly contradictory observation. The apparent paradox of stronger binding between OP/OP-I and SC lipids, yet better diffusion, is well explained by the interplay between binding affinity and lateral motion governed by frictional forces on the lipid surface.

The lateral motion of OP/OP-I on the SC lipid surface is primarily influenced by the friction coefficient rather than binding affinity (vertical separation) alone, as has been seen in other systems such as protein absorption on graphene surface (the protein displays high binding affinity to pristine graphene but can move on its surface freely; Mathesh et al, ACS Catalysis, 6, 4760-4768, 2016). Using the Stokes-Einstein relation, we derived the friction coefficients (γ) for OP, OP-I, and insulin on SC lipids to be 2.07×10^{-10} Ns/m, 4.31×10^{-10} Ns/m, and 8.49×10^{-10} Ns/m, respectively. This indicates that OP experiences the lowest friction, followed by OP-I and insulin, correlating directly with their relative ease of motion on the SC lipid surface. Accordingly, potential of mean force (PMF) analyses along the SC lipid surface (in the X direction) estimated the energy barrier for lateral diffusion to be 0.9 ± 0.1 kcal/mol for OP, 1.7 ± 0.2 kcal/mol for OP-I, and 2.2 ± 0.3 kcal/mol for insulin. These energy barriers align with the friction coefficients, demonstrating reduced resistance for OP and OP-I during diffusion. The distinct binding modes of OP and OP-I with SC lipids, compared to insulin, contribute to their lower friction and energy barriers (Extended Data Fig. 6). Consequently, OP and OP-I experience less "local trapping" or restriction in motion, enabling

faster diffusion despite their strong binding affinity.

We have revised the manuscript to incorporate this explanation and clarify the relationship between binding, friction, and diffusion dynamics (Page 9, Lines 274-280).

Extended Data Fig. 6 | Modeling of the interaction of OP-I with SC lipids. a, PMF analyses along the surface of the model SC lipids in different paths to estimate the energy barriers for diffusion. The energy barriers were 2.2 ± 0.3 kcal/mol for insulin, 0.9 ± 0.1 kcal/mol for OP, and 1.7 ± 0.2 kcal/mol for OP-I. Umbrella sampling distance: 2 nm; window resolution: 0.1 nm; sampling time: 35 ns per window; restraint force constant: 1000 kJ/mol/nm^2 . **b**, Representative local binding modes of insulin and OP on the rough model SC lipid surface at pH 5.5. **c**, Traces of insulin, OP, and OP-I on

the model SC lipids at pH 5.5 in three 100-ns independent MD trajectories for each. See also Extended Data Movie 3. Data are from $n = 3$.

Comment 6) i) Figure 5 suggests that substances "hopped" between cells, but Movie 4 shows fluorescence mostly remaining stationary. **ii)** Additionally, while the substance appears to penetrate the skin and enter the bloodstream within 2 hours, the experiments in Figure 5 observe phenomena on a scale of over 12 hours, reducing the persuasiveness of the results. Supplementary experiments are required to address these inconsistencies.

Authors' response: Thank you for your suggestions. We conducted additional experiments to address Comment 6i and 6ii.

To elucidate the movement of OP-I on the cell membrane surface as raised in Comment 6i, we utilized two complementary microscopy techniques. In Extended Data Movie 5, we employed total internal reflection fluorescence microscopy (TIRFM) to observe the real-time dynamics of OP-I^{Cy3} on the cell membrane. Focusing on a thin layer adjacent to the membrane surface (~100 nm), TIRFM revealed that OP-I^{Cy3} particles primarily exhibited vertical movements toward or away from the field of view, as well as lateral displacements along the membrane plane. The disappearance of some fluorescent particles suggests that these entities underwent longitudinal displacement away from the TIRFM detection plane. These observations highlight the dynamic nature of OP-I^{Cy3} particles on the cell membrane level.

In parallel, we used laser confocal microscopy in Extended Data Movie 6 to observe the movement of fluorescent particles over a broader spatial range. This technique provided a more comprehensive view of the overall behavior of OP-I^{Cy5} particles. Together, the detailed local insights from TIRFM and the broader spatial context from laser confocal microscopy corroborate our conclusion that OP-I can exhibit both lateral and longitudinal movements relative to the cell membrane.

As for Comment 6ii, the experiment in Fig. 5 involved the 12 h observation; the purpose was to show that OP-I always stayed on the cell membrane rather than entering the cytoplasm. This is important for the direct transfer of OP-I from the cell membranes. Following your suggestion, we conducted additional experiments at 3, 6, and 12 h intervals, confirming the OP-I transfer among cells increased over time (Fig. 5f). We also conducted further experiments to consolidate the contact-dependent cell-to-cell transfer of OP-I (Fig. 5g and Supplementary Fig. 22), further supporting the "hopping" mechanism.

Additionally, *in vitro* cell experiments demonstrate that the efficiency of intercellular contact transfer increases with prolonged treatment time. However, once OP-I enters the bloodstream *in vivo*, it undergoes dynamic physiological processes such as distribution to peripheral tissues, hepatic metabolism, and renal clearance. These dynamic processes give rise to a pharmacokinetic characteristic: plasma insulin levels rapidly increase, peaking at approximately 2 h post-administration, followed by a gradual decline due to elimination and metabolism. Hence, there exists a temporal discrepancy between the *in vivo* peak plasma insulin levels and the *in vitro* intercellular transfer of OP-I.

These additional experiments and analyses provide further evidence for the intercellular transfer of OP-I. We have incorporated these results into the revised manuscript (Page 11, Lines 320-333) and

Supplementary Information.

Extended Data Movie 5 (separate file) TIRFM imaging of OP-I^{Cy3} movement on HaCat cell membranes.

Extended Data Movie 6 (separate file) Direct transfer of OP-I^{Cy5} (shown in red) between HaCat cells observed by CLSM. The cells were imaged in time-lapse-acquisition mode right after adding OP-I^{Cy5}. The green color represented cell membranes stained with NBD-C6-HPC.

Fig. 5f | Flow cytometry analysis of transfer efficiency at different time points. Cy5-eq. dose: 1 $\mu\text{g}/\text{mL}$. Data are presented as mean \pm SD; $n = 3$. *** $P < 0.0001$

Fig. 5g | Cell-contact-dependent transfer of OP-I^{Cy5} between the HaCat cells on two face-to-face placed coverslips. HaCat cells on coverslip 1 were cultured with OP-I^{Cy5} (Cy5-eq. dose: 1 $\mu\text{g}/\text{mL}$) for 12 h, washed with PBS and imaged by CLSM (Coverslip 1). Coverslip 1 was then pressed face-to-face with Coverslip 2 with untreated HaCat cells. The two coverslips together were incubated in a 24-well plate containing fresh medium for 0.5 or 1 h at 37 $^{\circ}\text{C}$. The coverslips were washed and imaged: CLSM images (left) and quantitative analysis of Cy5 fluorescence by using imageJ software (right). All HaCat cells were imaged using CLSM under fixed parameters. Data are presented as mean \pm SD; $n = 3$. ** $P < 0.001$, *** $P < 0.0001$.

Supplementary Fig. 22 | Intercellular transfer of OP-I^{Cy5} among non-contact cells visualized by CLSM. HaCat cells ($\sim 10^5$) on Coverslip i were cultured in a medium containing OP-I^{Cy5} (Cy5-eq. dose: 1 $\mu\text{g}/\text{mL}$) for 4 h and then rinsed and imaged (Coverslip i-1). The coverslip i-1 was transferred into fresh culture medium with Coverslip ii pre-seeded with HaCat cells, while both coverslips were not in contact. After 12 h of incubation, both Coverslip i (now noted as Coverslip i-2) and Coverslip ii were rinsed and imaged; $n = 3$. Cell membranes were stained with NBD-C6-HPC (green).

Referee #3:

All individuals with type 1 diabetes and many with type 2 diabetes require insulin to control their hyperglycemia. For over 90 years insulin had to be given by injection. Multiple daily injections are not acceptable to all patients and a search for a non-injectable form of insulin has been pursued for many years. Inhaled forms of insulin have been available in some countries since 2006, but it has not been widely used, in large part due to the potentially severe side effects.

In this study, the authors describe the attachment of a skin-permeable polymer to insulin for transdermal delivery. They synthesise a soluble zwitterionic polymer, poly[2-(N-oxide-N,N-dimethylamino)ethyl methacrylate] (OP), which they conjugate to insulin (termed OP-I). They demonstrate that OP-I in a cream penetrates the skin of mice and minipigs and investigate the mechanism of penetration. OP-I entered the blood in both animals and reduced blood glucose concentrations. The authors conclude that OP-I could be used in individuals with diabetes.

A transdermal therapy for diabetes would be important for the care of patients with this disease. Nevertheless, the study has several deficiencies and requires considerably more work.

Thank you for your positive comments on our work.

Comment 1) A major deficiency is the very limited analysis of the biological effects of OP-I. The vast majority of the data presented use models and microscopy to focus on the penetration of OP and OP-I through the skin and tissues of the animals. Considerably more characterization of the biological effects of OP-I is needed. Analyses include, but are not limited to, the potential effects of the conjugate on binding to insulin receptors, receptor activation and signalling pathways. In addition, specificity for binding to insulin receptors and pharmacokinetics need to be studied.

Authors' response: Thank you for your comment. OP-I exhibited the same circular dichroism (CD) spectrum and hypoglycemic effects as native insulin (Extended Data Fig. 1d, e), indicating that OP-I has similar biological effects. Following your suggestions, we conducted additional experimental and computational simulation work to further compare the biological effects of OP-I with insulin in terms of insulin receptor (IR) binding specificity and dynamics, IR activation, and downstream signaling pathways.

Binding specificity to IR: We conducted Surface Plasmon Resonance (SPR) assays to investigate the binding specificity and affinity of OP-I for the insulin receptor extracellular domains (ECD-IR). The results, presented in Extended Data Fig. 2a-c and Supplementary Table 1, indicate that OP-I exhibited an IR-binding comparable to native insulin, with no significant differences in key parameters such as the dissociation constant (K_d), maximum response (R_{max}), Chi-square (χ^2) (Chi), and resonance angle offset (Offset). These findings demonstrate that OP-I can effectively bind to the IR with similar affinity and specificity as native insulin.

Receptor activation and signaling pathways: We analyzed an important protein in the the IR's downstream signaling pathways in the mouse muscle. STZ-induced diabetic mice were intravenously administered with equal doses of OP-I and native insulin to analyze whether OP conjugation affects insulin's ability to activate the IR's downstream signaling pathways. Western blot analysis of muscle tissues showed that OP-I effectively activated the IR signaling pathway and its downstream protein AKT comparable to those induced by native insulin (Extended Data Fig. 2f).

Molecular dynamics (MD) simulations: All-atom MD simulations were performed to further investigate the interaction of OP-I with the IR. The representative binding modes of insulin and OP-I on the IR illustrated that OP-I stably adsorbed on all binding sites of the receptor (Extended Data Fig. 2d and Extended Data Movie 1) similar to insulin. PMF analyses estimated the binding affinities of OP-I with the two major IR binding site 1 and site 2 to be -14.0 kcal/mol and -22.9 kcal/mol, respectively, comparable to those of insulin (-16.7 kcal/mol and -22.0 kcal/mol) (Extended Data Fig. 2e).

We have incorporated these results into the revised manuscript (Page 4, Lines 109-124).

Extended Data Fig. 1 | d, CD spectra in H₂O. **e**, BGLs of diabetic mice after *s.c.* injection of OP-I, PEG-I, or native insulin; insulin-eq. dose: 5 U/kg. Data are presented as mean ± SD; n = 5.

Extended Data Fig. 2 | Insulin receptor (IR) binding and activation of insulin and OP-I. a, Schematic of the SPR assay: ECD-IR was immobilized onto the SPR surface, followed by incubation with insulin or OP-I. Created in BioRender.com. **b**, SPR traces showing the binding of insulin and OP-I to ECD-IR; RU, resonance units. **c**, The dissociation constants (K_d) calculated from the binding curves of insulin and OP-I in (b); no significant difference. **d**, Representative binding modes from all-atom MD simulations of insulin and OP-I on the ECD-IR. **e**, PMF analysis of the binding affinities with two major ECD-IR binding sites. Umbrella sampling distance: 5 nm; window resolution: 0.1 nm; sampling time: 20 ns per window; restraint force constant: 1000 kJ/mol/nm². **f**, Western blot analysis of phosphorylated IR (pIR) and phosphorylated AKT (pAKT) levels of the mouse skeletal muscle.

Tissues were harvested from the mice at 10 min after a single intravenous injection of PBS, insulin, or OP-I (insulin-eq. dose: 1.5 U/kg).

Extended Data Movie 1 (separate file) Molecular dynamics trajectories comparing the binding of insulin and OP-I on the IR.

Supplementary Table 1 The dissociation constant (K_d), maximum response (R_{max}), Chi-square (χ^2) (Chi), and resonance angle offset (Offset) of insulin or OP-I binding to ECD-IR, calculated from SPR binding curves in Extended Data Fig. 2b; $n = 3$. RU, resonance unit.

	K_d (nM)	R_{max} (RU)	Offset (RU)	Chi ² (RU ²)
insulin	31.9 ± 9.8	6.2 ± 0.8	2.4 ± 0.2	0.08 ± 0.09
OP-I	24.4 ± 4.5	5.7 ± 0.5	2.1 ± 1.2	0.02 ± 0.01

Comment 2) The number of replicates should be given in the legends for all figures. This is essential to evaluate reproducibility of the data.

Authors' response: Thank you for your comment. Following your suggestion, we have reviewed all figures and their legends and included the n values of all experiments in the revised manuscript.

Comment 3) Patients are treated with several forms of insulin with different durations of action. Which insulin was used in this study?

Authors' response: Thank you for your question. In this study, we used recombinant human insulin (Solarbio, I8830), which is a short-acting native insulin commonly employed in experimental and clinical settings. This information has been included in the revised manuscript (Page 14, Lines 503-504).

Comment 4) Fig. 2b. How do you explain the highest OP-I at 24 h when peak blood insulin concentrations are at 2 h?

Authors' response: Thank you for pointing this out. It is worth mentioning that the 24 h was for *in vitro* experiment and the 2 h was for *in vivo* experiments.

In the *in vitro* EpiKutis® model used in Fig. 2b, the system was static and designed to measure the cumulative amount of OP-I penetrating through the skin into the receiver compartment over time. Unlike *in vivo* systems, this model lacks physiological clearance mechanisms, such as hepatic metabolism or renal excretion. As a result, the OP-I concentration in the receiver compartment progressively accumulated over time until equilibrium was established between the donor and receiver compartments. This explains the continuous increase in OP-I levels observed at 24 h.

In the *in vivo* experiments discussed in Fig. 3a and Supplementary Fig. 9, topically applied OP-I penetrated the skin and entered the systemic circulation. Once in the bloodstream, OP-I distributed in peripheral tissues, hepatic metabolism, renal clearance, and other dynamic physiological processes.

These dynamic processes led to a pharmacokinetic profile characterized by a rapid rise in blood insulin concentration, peaking at ~ 2 h post-administration, followed by a gradual decline due to clearance and metabolism.

We have incorporated these explanations in the revised manuscript (Page 6, Lines 177-179) accordingly to bring the clarity.

Fig. 2b | Time-dependent permeation curves of OP-I, PEG-I, and insulin across the EpiKutis® model (insulin-eq. dose: 0.2 mL of 0.5 mg/mL; application area: 0.081 cm^2).

Fig. 3a | Plasma insulin concentrations in STZ-induced diabetic mice after topical application of PBS, insulin, PEG-I, or OP-I (insulin-eq. dose: 116 U/kg; 0.2 mL of 0.5 mg/mL solution applied on 1.13 cm^2 dorsal skin); $n = 5$. Mice were subcutaneously (*s.c.*) injected with native insulin (5 U/kg) as a positive control.

Supplementary Fig. 9 | Fluorescence images of the major organs, including heart (H), liver (L), spleen (S), lungs (Lu), and kidneys (K), of the mice after topical administration of OP-I^{Cy5}, PEG-I^{Cy5}, or insulin^{Cy5} solutions (Cy5-eq. dose: 0.2 mL of 10 $\mu\text{g}/\text{mL}$; application area: 1.13 cm^2) on the dorsal skin, or after subcutaneous (*s.c.*) injection of insulin^{Cy5} (Cy5-eq. dose: 50 μL of 25 $\mu\text{g}/\text{mL}$); $n = 3$.

Comment 5) Fig. 2. What is the biological relevance of OP-I in the lymphatics? While insulin has effects on all organs in humans, the main target organs for glucose homeostasis and fat metabolism are adipose tissue, skeletal muscle and the liver. What was OP-I distribution in fat and skeletal muscle? Can OP-I enter human fat, liver and skeletal muscle cells? Insulin secreted by the pancreas undergoes first pass metabolism in the liver, resulting in a very short (3-10 min) biological half-life for secreted insulin. What is the half-life of OP-I?

Authors' response: Thank you for your comment. Following your suggestions, we conducted additional experiments and provide detailed responses to address each point you raised:

Biological relevance of OP-I in the lymphatics: Subcutaneously administered macromolecular therapeutics, including proteins, are known to enter lymphatic vessels and then systemic circulation (Pharm. Res., 2001, 18, 1620). In this study, we observed that topically applied OP-I penetrated the skin and localized in subcutaneous lymphatic vessels. This suggests that lymphatic transport serves as the primary pathway for OP-I to enter systemic circulation, aligned with the established

mechanisms for macromolecule absorption.

Biodistribution of OP-I: To investigate OP-I's distribution in glucose-regulating tissues, we performed *ex vivo* imaging of liver, adipose, and muscle tissues from the treated mice using an IVIS system. The obtained results demonstrated significant accumulation of OP-I in the liver (Supplementary Fig. 9), adipose tissues (Supplementary Fig. 10a), and skeletal muscles (Supplementary Fig. 11a), primary sites for insulin-mediated glucose uptake, storage, and metabolism. This targeted biodistribution of topical OP-I accounted for its observed robust hypoglycemic effects, as these tissues play key roles in maintaining glucose homeostasis.

Cellular uptake of OP-I: We performed *in vitro* cellular uptake studies to evaluate OP-I's ability to enter fat, skeletal muscle, and liver cells. The results confirmed efficient uptake of OP-I by liver cells (Supplementary Fig. 12), adipocytes (Supplementary Fig. 13), and skeletal muscle cells (Supplementary Fig. 14), consistent with the observed accumulation in these tissues. These findings support OP-I's ability to exert its biological effects on key target tissues involved in glucose regulation.

Biological half-life of OP-I: The plasma insulin profile after intravenous injection of native insulin and OP-I showed that OP-I exhibited slower blood clearance than native insulin, with an extended half-life of 10-20 min (Supplementary Fig. 7). This prolonged circulation time was attributed to the conjugated zwitterionic OP, which reduced interactions with plasma proteins and reticuloendothelial system uptake, thereby prolonging blood circulation (Nat. Biomed. Eng., 2021, 5, 1019).

We have incorporated these new results into the revised manuscript (Page 4, Lines 125-129; Page 6, Lines 174-181).

Supplementary Fig. 9 | Fluorescence images of the major organs, including heart (H), liver (L), spleen (S), lungs (Lu), and kidneys (K), of the mice after topical administration of OP-I^{Cy5}, PEG-I^{Cy5},

or insulin^{Cy5} solutions (Cy5-eq. dose: 0.2 mL of 10 µg/mL, application area: 1.13 cm²) on the dorsal skin, or after subcutaneous (*s.c.*) injection of insulin^{Cy5} (Cy5-eq. dose: 50 µL of 25 µg/mL); n = 3.

Supplementary Fig. 10a | Fluorescence images of mouse adipose tissues including brown adipose (BAT), subcutaneous white adipose (iWAT), and visceral white adipose (eWAT) after topical administration of OP-I^{Cy5} (0.2 mL, Cy5-eq. dose: 0.2 mL of 10 µg/mL; application area: 1.13 cm²) on the dorsal skin; n = 3.

Supplementary Fig. 11a | Fluorescence images of the mouse forelimb muscle (FLM) and hindlimb muscle (HLM) muscle after topical administration of OP-I^{Cy5} (Cy5-eq, dose: 0.2 mL of 10 µg/mL, application area: 1.13 cm²) on the dorsal skin; n = 3.

Supplementary Fig. 12 | Cellular uptake of OP-I^{Cy5} or insulin^{Cy5} (red) by AML-12 cells observed

using CLSM; n = 3. Cy5-eq. dose: 1 $\mu\text{g}/\text{mL}$; 4 h incubation.

Supplementary Fig. 13 | Cellular uptake of OP-I^{Cy5} and insulin^{Cy5} (red) by 3T3-L1 differentiated adipocytes and their colocalization with adipocyte lipid droplets (green) observed using CLSM; n = 3. Cy5-eq. dose: 1 $\mu\text{g}/\text{mL}$; 4 h incubation.

Supplementary Fig. 14 | Cellular uptake of OP-I^{Cy5} and insulin^{Cy5} (red) by mouse skeletal muscle cells observed using CLSM; n = 3. Cy5-eq. dose: 1 $\mu\text{g}/\text{mL}$; 4 h incubation.

Supplementary Fig. 7 | Normalized blood clearance profiles, expressed as the percentage of

remaining fluorescence intensity relative to the first sampling point as a function of time. OP-I^{Cy5} or insulin^{Cy5} was intravenously injected *via* the tail vein at a Cy5-eq dose of 0.1 mg/kg. Data are mean \pm SD of biological replicates (n = 3).

Comment 6) Fig 2d. The Merged image does not reflect individual images. There is no mention of rats in Methods. Scale bars are needed in all microscopic images.

Authors' response: Thank you for pointing these out. We have included an image of each individual channel in the revised manuscript. These additions provide a detailed representation of the co-localization data presented in Fig. 1a and Fig. 2c.

In our study, Sprague-Dawley (SD) rats were utilized for the co-localization analysis of subcutaneous lymphatic vessels. We have added a detailed description of the methodology in the Methods section of the revised manuscript, including information on the species, strain, age, weight, and experimental procedures (Page 19, Lines 704-708).

Additionally, we have carefully reviewed all microscopic images and added scale bars in the revised manuscript where necessary.

Comment 7) Fig S4. What is the *in vivo* relevance of OP-I in the skin at 4 h? how long does it remain in the skin? How long is OP-I biologically active?

Authors' response: Thank you for your comment. We conducted additional experiments and provide detailed responses to address each point you raised:

In vivo relevance of OP-I in the skin at 4 h: We evaluated the skin permeation of OP-I following topical application for 2 and 4 h. The results demonstrated that the amount of OP-I penetrating into the skin increased with application time (Extended Data Fig. 3b). Based on these findings, we selected 4 h as the standard duration for transdermal treatment in our study. This duration ensures effective delivery of OP-I while maintaining feasibility for therapeutic applications.

Retention time of OP-I in the skin: To assess how long OP-I remains in the skin after application, we tracked the retention of OP-I^{Cy5} over time. After a 4 h topical application, the formulation was removed, and fluorescence was monitored. The results showed that OP-I^{Cy5} in the skin decreased rapidly and was nearly completely cleared within 8 h (Supplementary Fig. 17).

Biological activity stability of OP-I: The biological activity of OP-I was assessed by evaluating its hypoglycemic effects in STZ-induced diabetic mice. Following a 4 h transdermal application, OP-I exhibited a sustained hypoglycemic effect lasting more than 12 h (Fig. 3c), indicating that OP-I remains biologically active over an extended period after application.

We have incorporated these findings into the revised manuscript (Page 5, Line 159; Page 6, Lines 193-194).

Extended Data Fig. 3b | CLSM images of the slices of the C57BL/6J mouse dorsal skin after topical application with OP-I^{Cy5} (Cy5-eq. dose: 0.2 mL of 10 μg/mL applied on 1.13 cm²) for 2 or 4 h. The slices were stained with DAPI (blue) to label the nuclei.

Supplementary Fig. 17 | Fluorescence imaging the skin retention of OP-I^{Cy5}. OP-I^{Cy5} in a diffusion cell (Cy5-eq. dose: 0.2 mL of 10 μg/mL; application area: 1.13 cm²) was topically applied on the dorsal skin of C57BL/6J mice for 4 h and then removed; one group was sacrificed, another group was sacrificed after 4 h or 8 h (n = 3) for analysis. **a**, IVIS *ex vivo* imaging the skins. **b**, CLMS imaging of the sections of the skins.

Fig. 3c | Blood glucose levels (BGLs) of STZ-induced diabetic mice after different treatments as indicated in (a); n = 8.

Comment 8) The authors extensively characterise skin penetration of OP and OP-I. Patients inject insulin into the subcutaneous fat (not intradermally, as stated line 162). How long does OP-I take to reach the subcutaneous fat?

Authors' response: Thank you for your comment. We apologize for the small mistake and have corrected it in the revised manuscript.

To address your question, we conducted an additional experiment to evaluate the skin penetration kinetics of OP-I. OP-I^{Cy5} was topically applied on the dorsal skin, and the skin was isolated, sectioned, and analyzed using CLSM. The results reveal that fluorescence signals were already significant in the subcutaneous adipose tissues after 10 min of topical application and accumulated more after 30 min (Extended Data Fig. 3a), demonstrating the efficient transdermal delivery of OP-I. We have incorporated these findings into the revised manuscript (Page 5, Lines 156-159).

Extended Data Fig. 3a | Fluorescence distribution in the subcutaneous fat of the C57BL/6J mouse dorsal skin after topical application with OP-I^{Cy5} for timed intervals (Cy5-eq. dose: 0.2 mL of 10 µg/mL applied on 1.13 cm²). BODIPY (green) was used to label subcutaneous adipose tissue and DAPI (blue) was used to label the nuclei.

Comment 9) Fig. 3. How was the dose of OP-I selected? Patients are very careful about the dose of insulin they inject. Please show the reproducibility of topical application of OP-I on plasma insulin concentration. Variation among different individuals applying OP-I should be shown. Does plasma insulin concentration vary with the site of topical application?

Authors' response: Thank you for your comment. Following your suggestions, we conducted additional experiments and provide detailed responses address each of your questions:

Selection of the OP-I dosage: The initial dose of OP-I used in this study (116 U/kg, ~20 times the subcutaneous dose) was selected as a proof of concept to demonstrate OP-I's transdermal ability (Fig. 3c). As per your request, we have evaluated the dose-dependent effects of OP-I on its hypoglycemic efficacy. The results demonstrated that lower doses of OP-I were both feasible and effective, with 58 U/kg achieving normoglycemia within 4 h post-administration and 29 U/kg within

6 h (Fig. 3d). These findings underscore the potential of OP-I for personalized treatment, as insulin dosing in clinical practice is highly individualized to achieve optimal glycemic control.

It is worth pointing out that noninvasive delivery methods, such as transdermal or oral administration, always require higher doses than subcutaneous injection due to multiple physiological barriers. For instance, insulin microneedles require doses over 100 times of subcutaneous doses (Nat. Biomed. Eng., 2020, 4, 499; Sci. Adv., 2022, 8, eadd3197; Proc. Natl. Acad. Sci. U.S.A., 2020, 117, 29512), and the clinically approved oral semaglutide requires 30 to 50 times of subcutaneous doses (Lancet, 2023, 402, 693-704). The topical OP-I dosage just needed 5-23 times of subcutaneous doses. Despite the higher dosage, OP-I transdermal delivery offers significant advantages in terms of patient convenience and compliance, which outweigh the higher dose drawback. Furthermore, we will explore strategies such as adjusting the OP-to-insulin ratio in the future work to reduce the required transdermal dose while maintaining therapeutic efficacy.

Reproducibility of topical application of OP-I on plasma insulin concentration: To assess the reproducibility of transdermal OP-I administration, the same dose was applied to the same mice over three consecutive days. As shown in Fig. 3b, the consistent plasma insulin profiles observed across treatments demonstrated high reproducibility of transdermal insulin delivery in individual mice following repeated applications of OP-I.

Variation among different individuals and topical application sites: Following transdermal administration of OP-I in different mice (n = 5), the inter-individual variability was small in plasma insulin concentrations (Fig. R1). To evaluate the effect of the application sites, OP-I was applied to the abdomen and back skin of mice. There was no significant difference in the transdermal delivery efficiency (Fig. 3e) or hypoglycaemic activity (Fig. 3f). These findings underscore OP-I's potential for autonomous and flexible administration.

We have incorporated these findings into the revised manuscript (Page 6, Lines 167-173; Page 6, Lines 182-192) with additional discussions and figures.

Fig. 3c | Blood glucose levels (BGLs) of STZ-induced diabetic mice after different treatments as indicated in (a); n = 8.

Fig. 3d | BGLs in the diabetic mice after topical application of lower doses of OP-I (insulin-eq. dose: 58 or 29 U/kg; 0.05 mL or 0.1 mL of 0.5 mg/mL solution applied on 1.13 cm² dorsal skin); n = 8.

Fig. 3b | Plasma insulin concentrations in STZ-induced diabetic mice following three consecutive days of topical application of OP-I as in (a); n = 5.

Fig. R1 | Individual plasma insulin concentrations of STZ-induced diabetic mice after topical

application of OP-I (insulin-eq. dose: 116 U/kg, 0.2 mL of 0.5 mg/mL); n = 5.

Fig. 3e | Plasma insulin concentrations (n = 5) of the diabetic mice after topical application of OP-I on the dorsal or abdominal skin at the same dose as in (a).

Fig. 3f | BGLs (n = 8) of the diabetic mice after topical application of OP-I on the dorsal or abdominal skin at the same dose as in (a).

Comment 10) Fig. 3. The goal of insulin therapy in patients with diabetes is to get glucose concentrations as close to normal as feasible. Hypoglycemia is the most feared complication of insulin administration. It is concerning that OP-I produced a prolonged reduction in glucose. What was the nadir glucose concentration? The authors should provide the reference interval ("normal range") for the mouse strain used to facilitate data interpretation. In panel c, the glucose concentrations at 90 and 120 min appear very low.

Authors' response: Thank you for raising this important concern.

Avoiding hypoglycemia is indeed a critical consideration during insulin therapy. For C57BL/6J mice, blood glucose levels below 50 mg/dL are considered hypoglycemic (Sci. Adv., 2022, 8, eadd3197). So, we have delineated the "normal" range (50-200 mg/dL) in the revised figures (Fig. 3c,

3d, 3f, 3g and 3i) using a gray background. The lowest glucose concentration observed following transdermal administration of OP-I was 69.7 mg/dL at 90 min post-initiation of the IPGTT experiment (Fig. 3g). This value was within the normoglycemic range, confirming the safety of transdermal OP-I administration under the experimental conditions.

Moreover, we observed dose-dependent glycemic regulation with OP-I, allowing for individualizing dose adjustments to achieve effective glycemic control while minimizing the risk of hypoglycemia (Fig. 3d). These findings underscore the flexibility of OP-I in enabling personalized treatment strategies.

These updates have been incorporated into the revised manuscript (Page 6, Lines 190-192).

Fig. 3g | Intraperitoneal glucose tolerance tests (IPGTTs) in diabetic mice; n = 5. Mice received treatments as indicated in (a). One hour later, they were intraperitoneally (*i.p.*) injected with glucose at 1.5 g/kg, and then BGLs were measured; healthy mice were used as a control.

Fig. 3d | BGLs in the diabetic mice after topical application of lower doses of OP-I (insulin-eq. dose: 58 or 29 U/kg; 0.05 mL or 0.1 mL of 0.5 mg/mL solution applied on 1.13 cm² dorsal skin); n = 8.

Comment 11) Fig. 3. Topical PEG-I and insulin reduced glucose, despite not being absorbed. Explanation?

Authors' response: Thank you for your question. Upon careful analysis of the data for the

transdermal PEG-I and insulin groups, we observed a slight reduction in blood glucose levels. However, this reduction was not statistically significant compared to the control group and fell within the normal physiological fluctuations observed in fasting mice.

Comment 12) Considerably more experimental detail should be provided for all mouse and minipig experiments. Were animals fed/fasted? If the latter, how long?

Authors' response: Thank you for your comment. We have thoroughly revised the Methods section to include additional details about the experimental conditions for both mice and minipigs. Specifically, both animal models were subjected to a fasting period of 12 h during the experimental procedures.

Comment 13) Insufficient numbers of animals are used in the experiments. At least 7-10 are needed. This is particularly important in light of the large variability in fig S6. What is the reason for the large (~2-fold) variability at t=0 in Fig S6a?

Authors' response: Thank you for your comment. Following your suggestion, we have repeated the glucose-lowering experiment using 8 mice per group, meeting the recommended sample size to ensure robust and reliable conclusions (Fig. 3c, 3d, 3f and Supplementary Fig. 15).

The observed variability at $t = 0$ was attributed to the different baseline blood glucose levels among individual diabetic mice, which was due to differences in degrees of beta-cell damages induced by STZ, metabolic states, or individual stress responses. As per your request, we refined our experimental protocol by optimizing and standardizing the STZ dose used to induce diabetes and implemented strict inclusion criteria, selecting only diabetic mice with fasting blood glucose levels ranging from 300 to 600 mg/dL. These measures successfully reduced variability in baseline glucose levels, as confirmed in the repeated experiments.

We have incorporated these updates into the revised manuscript (Page 7, Lines 222-226) and supplementary figures.

Fig. 3c | Blood glucose levels (BGLs) of STZ-induced diabetic mice after different treatments as indicated in (a); $n = 8$.

Fig. 3d | BGLs in the diabetic mice after topical application of lower doses of OP-I (insulin-eq. dose: 58 or 29 U/kg; 0.05 mL or 0.1 mL of 0.5 mg/mL solution applied on 1.13 cm² dorsal skin); n = 8.

Fig. 3f | BGLs (n = 8) of the diabetic mice after topical application of OP-I on the dorsal or abdominal skin at the same dose as in (a).

Supplementary Fig. 15 | Blood glucose levels (BGLs) of mice in each group at the time points shown in Fig. 3c. Data are mean \pm SD of biological replicates (n = 8).

Comment 14) The units on the Y-axis in S6b appear incorrect. The Y-axis units in S6c are confusing.

Authors' response: Thank you for pointing these out. We have carefully reviewed and corrected the Y-axis labels. The Y-axis unit in Supplementary Fig. 16 has been revised to "Relative to initial blood glucose level (%)", and the Y-axis unit in Fig. 3h has been corrected to "AUC (mg/dL·min)".

Supplementary Fig. 16 | Relative to initial BGLs of healthy mice after topical administration of OP-I, PEG-I, or native insulin (insulin-eq. dose: 116 U/kg; 0.2 mL of 0.5 mg/mL; application area: 1.13 cm²), or *s.c* injection of insulin (5 U/kg). Data are mean \pm SD of biological replicates (n = 5).

Fig. 3h | Area under the curve (AUC) from 0-120 min of the IPGTT experiment in (g); n = 5.

Comment 15) Considerably more experimental detail is needed in Methods and legends.

Authors' response: Thank you for your comment. Following your suggestion, we have revised the manuscript to include comprehensive experimental details in both the Methods section and figure legends.

Comment 16) Statistical analysis should be performed wherever feasible.

Authors' response: Thank you for your suggestion. We have carefully reviewed all experimental datasets and performed appropriate statistical analyses wherever feasible. These analyses have been detailed in the figure legends.

RESPONSES TO REFEREES' COMMENTS

Nature Submission ID: 2024-05-10937A-Z

Title: A skin-permeable polymer for noninvasive transdermal insulin delivery

Authors: Qiuyu Wei, Zhi He, Zifan Li, Zhuxian Zhou, Ying Piao, Jianxiang Huang, Yu Geng, Runnan Zhang, Yaqi Fu, Jiayi Ye, Yue Yuan, Haoru Zhu, Jiaheng Zeng, Yan Zhang, Quan Zhou, Mingyu Xu, Shiqun Shao, Jianbin Tang, Jiajia Xiang, Rongjun Chen, Ruhong Zhou & Youqing Shen

We would like to express our sincere gratitude to the referees for their helpful and constructive suggestions, which have greatly improved the revised version. We have taken careful note of their comments and suggestions, carried out further experiments, and revised our manuscript accordingly.

A point-by-point response to all referees' comments is given in blue color alongside the referees' comments below. The revisions in the revised manuscript are highlighted in blue color.

Referee #2:

The authors have thoroughly addressed the comments raised by this reviewer. I believe the significance of this study is high, as it presents a well-executed investigation that not only enables the transdermal delivery of insulin but also provides a detailed explanation of the transdermal delivery mechanism—an aspect that, to the best of this reviewer's knowledge, has not been effectively explained in previous studies.

Although this is a minor issue, I suggest that the authors consider including a more detailed summary of their findings in the conclusion section, which would enhance the overall structure and clarity of the manuscript.

Authors' response: Thank you very much for your positive evaluation and constructive suggestion. We have revised the conclusion section (Page 12; Lines 373-377): “In summary, this study presents the first non-invasive transdermal insulin delivery system that achieves *in vivo* hypoglycemic efficacy comparable to subcutaneous injections for diabetes treatment, resulting from the OP's efficient skin permeation. The OP conjugation is versatile for transdermal delivery of biomacromolecules such as peptides, proteins, and nucleic acids, with broad therapeutic applications, warranting further investigation in future studies.”

As the Summary is just after the Discussion, we did not repeat the mechanism content in the Summary.

Referee #3:

This is a revised and improved manuscript. An effective form of insulin that could be applied topically to the skin for noninvasive transdermal uptake is likely to appeal to individuals with diabetes who require insulin. Moreover, the strategy adopted by the authors could potentially be applied to other parenteral therapeutics.

The authors have performed additional experiments that adequately addressed some of my prior concerns. A few items require attention.

Thank you for your positive comments on our work.

Comment 1) The SPR data do not address the question of specificity for binding to insulin receptors. In addition, OP-I pharmacokinetics should be determined as previously requested.

Authors' response: We sincerely appreciate your insightful comments. In response to your concerns, we conducted the required experiments (**Extended Data Fig. 2a**).

i) Binding specificity: We used SPR and measured the interaction between OP-I and IGF1R, a membrane receptor closely related to the insulin receptor (IR). The results in **Extended Data Fig. 2d** demonstrate that OP-I exhibits negligible binding affinity to ECD-IGF1R, indicating that OP-I maintains the insulin specificity to its receptor.

ii) Pharmacokinetics of OP-I (*in vitro*): we calculated the kinetic dissociation rate constants and half-life ($t_{1/2}$) for both OP-I and insulin. As illustrated in **Extended Data Fig. 2b-c**, OP-I had pharmacokinetic properties highly comparable to those of insulin.

iii) Pharmacokinetics of OP-I (*in vivo*): We measured the *in vivo* pharmacokinetics of OP-I as previously requested. The data from the blood clearance curves show that the blood half-life of OP-I (approximately 15-20 minutes) was slightly longer than that of native insulin (approximately 5-10 minutes) (**Supplementary Fig. 8a**).

Extended Data Fig. 2 | SPR characterization of insulin receptor (IR) binding and activation of insulin and OP-I. **a**, Schematic illustration of the experimental setup, where anti-His antibody was immobilized on the CM5 chip sensor surface, followed by the addition of Histag-ECD-IR/IGF1R. Insulin/OP-I was introduced to study their interactions. Created in BioRender.com. **b**, SPR traces showing the binding of insulin or OP-I to ECD-IR; RU, resonance unit. **c**, Dissociation constant (K_d), association rate constant (K_{on}), dissociation rate constant (K_{off}), and half-life ($t_{1/2}$) of insulin or OP-I binding to ECD-IR, calculated from SPR binding curves in Extended Data Fig. 2b. **d**, SPR traces showing minimal binding of insulin and OP-I to ECD-IGF1R; RU, resonance unit.

Supplementary Fig. 8a, Normalized blood clearance profiles, expressed as the percentage of remaining fluorescence intensity relative to the first sampling point as a function of time. OP-I^{Cy5} or insulin^{Cy5} was intravenously injected *via* the tail vein at a Cy5-eq dose of 0.1 mg/kg. Data are mean \pm SD (n = 3).

We have incorporated these new findings and discussions into the revised manuscript (Page 4, Lines 111-119 and 127-129).

Comment 2) Extended Fig. 2f. The authors should examine the effect on insulin signalling of OP-I applied to the skin to show that it is effective when used as intended. Better quality Western blots should be shown. Total IR and Akt should be shown and the extent of phosphorylation should be corrected for the total amount of protein in the same sample. The analysis should be repeated in several (≥ 3) independent experiments to validate reproducibility, data should be quantified and the results compared to regular insulin given sq.

Authors' response: We thank your constructive suggestion. We have conducted experiments as suggested. Specifically, we investigated the activation of insulin receptor (IR) in muscle tissues of streptozotocin (STZ)-induced diabetic mice following transdermal administration of OP-I, using subcutaneous insulin injection as a control.

Western blot analysis of the muscle tissues revealed that topical administration of OP-I effectively activated the IR signaling pathway and its downstream protein AKT. To ensure data reliability and reproducibility, we conducted three independent experiments. Additionally, we quantified the data and normalized the extent of phosphorylation to the total amount of IR and AKT within each sample (**Supplementary Fig. 17**). These results provide robust evidence that OP-I efficiently triggered insulin signaling.

We have incorporated these results into the revised manuscript (Page 6, Lines 181-183).

Supplementary Fig. 17 | Western blot analysis of phosphorylated IR (pIR) and phosphorylated AKT (pAKT) levels of the mouse skeletal muscle. Tissues were harvested from the STZ-induced diabetic mice 4 h after topical application of OP-I (insulin-eq. dose: 116 U/kg; 0.2 mL of 0.5 mg/mL solution applied on 1.13 cm² dorsal skin) or 1 h after subcutaneously (s.c.) injected with native insulin (5 U/kg) as a positive control. (n = 3)

Source Data for **Supplementary Fig. 17**

Comment 3) Supp Fig 7. Statistical analysis should be performed to compare the half-life of OP-I and insulin to support the authors' "marginally longer" claim in the text.

Authors' response: Thank you for your valuable suggestion.

We employed the DAS2 software, a widely recognized tool for pharmacokinetic analysis, to precisely calculate the half-life values of OP-I and insulin. The detailed statistical analysis and results are included in **Supplementary Fig. 8a**. The half-life of s.c. injected native insulin is approximately 9.4 min, while OP-I has a longer half-life (~21.5 min), which is consistent with our statement

“marginally longer” . We have included these statistical results in the revised manuscript, as suggested (Page 4, Lines 127-129).

Supplementary Fig. 8a, Normalized blood clearance profiles, expressed as the percentage of remaining fluorescence intensity relative to that of the first sampling point as a function of time. OP-I^{Cy5} or insulin^{Cy5} was intravenously injected *via* the tail vein at a Cy5-eq dose of 0.1 mg/kg. Data are mean ± SD (n = 3).

Comment 4) Fig. 3. The authors should explain in the legend what the shaded area represents in panels c, d, f, g & i.

Authors' response: Thank you for pointing out this oversight. We have added the following description (Page 7, Lines 239-240): “In panels c, d, f, g, and i, the shaded areas outline the normal blood glucose range (50 – 200 mg/dL).”

Comment 5) Fig. 3c, Suppl Fig 15. Statistical analysis should be performed and included to compare the effect of OP-I and sq insulin. The data in 3c do not support the claim that topical insulin and PEG-I had “neglectable [sic] impact on BGL”, eg, insulin at 4 h.

Authors' response: We sincerely appreciate your valuable instruction. We performed a comprehensive series of statistical analyses in **Supplementary Fig. 18**:

i) Statistical comparison of hypoglycemic effects between OP-I and subcutaneous insulin:

The results showed that there was no significant difference in hypoglycemic performance between OP-I and s.c. insulin from 0 - 2 hours. However, a highly significant difference (****P < 0.0001) emerged during the 4-12 h period. This indicates that topically administered OP-I maintained its hypoglycemic effect, while s.c. insulin was metabolized and cleared, losing its hypoglycemic capacity.

ii) Verification of negligible impacts of topical insulin and PEG-I on blood glucose levels (BGLs): The statistical analysis revealed that, except at 6 hours, there were no significant differences in BGLs between the topical insulin/PEG-I groups and the PBS group. At the 6-hour time point, a marginal significance (*P < 0.05) was observed, which was attributed to inter-animal variability and fasting conditions. To provide additional context, we compared these groups with the OP-I group and

found highly significant differences (**** $P < 0.0001$), strongly supporting our conclusion that PEG-I and insulin did not cause significant changes in BGLs.

Supplementary Fig. 18 | Blood glucose levels (BGLs) of mice in each group at the time points shown in Fig. 3c. Data are mean \pm SD of biological replicates ($n = 8$; **** $P < 0.0001$, * $P < 0.05$, ns, not significant, one-way ANOVA).

Comment 6) P. 6, line 212. What is meant by “...caused almost no significant changes in...BGL” ? Was statistical analysis performed?

Authors' response: Thank you very much for your insightful comment. In response to your concern, we conducted statistical analysis on the BGL data of the PEG-I group, insulin group, and PBS control group. Specifically, we employed one-way ANOVA to compare the BGL values across all experimental time points in **Supplementary Fig. 21**.

The statistical analysis clearly demonstrated that there were no significant differences in BGLs between the PEG-I group, insulin group, and PBS group at any of the time points examined.

Supplementary Fig. 21 | BGLs of minipigs in each group at the timed points in Fig. 3i. Data are mean \pm SD of biological replicates ($n = 3$; ns, not significant, one-way ANOVA).

Comment 7) Fig. 3i. In the Methods, the authors state 3 minipigs were injected with STZ. However, in 3i they have 5 different assay conditions for each pig. Explanation?

Authors' response: Thank you for highlighting this important issue. We apologize for our unclear presentation. Each assay condition shown in Fig. 3i was evaluated on three separate minipigs, resulting in a total of 15 experimental data points (5 assay conditions \times 3 minipigs). We have revised the relevant text in the Methods section (Page 20, Line 805).

Comment 8) Fig. 3i and Suppl Fig 18. No data are shown to support the claim of “over 12 h”. It is important to determine the duration of action to avoid hypoglycemia.

Authors' response: Thank you for carefully reviewing our figures and pointing out this issue. The data indeed demonstrated the effect “for 12 h”. We have corrected the text in the figure legends and the main manuscript. (Page 6, Lines 197 and 217)

Comment 9) Suppl Fig 15. The scale of Y-axis at 2 h appears different to those at the other time points.

Authors' response: Thank you for your careful observation. We apologize for this mistake. In the revised manuscript, we have corrected it **Supplementary Fig. 18** (newly numbered).

Supplementary Fig. 18 | Blood glucose levels (BGLs) of mice in each group at the time points shown in Fig. 3c. Data are mean \pm SD of biological replicates ($n = 8$; **** $P < 0.0001$, * $P < 0.05$, ns, not significant, one-way ANOVA).

Referee #4:

The authors have designed an insulin 1:1 conjugate that seems to enable efficient delivery of insulin across the skin. It is claimed that the polymer is permeating the skin into the blood circulation, due to the polyelectrolyte structure leading to change in net charge at neutral pH. Uptake pathway is discussed to be related to improved diffusion across epidermis via cell-to-cell membrane diffusion and subsequently drained via the lymphatic vessels to the systemic circulation. The first revision led to an improved manuscript.

The work is original and with an interesting compound, promising results. May indeed be of high significance if useful for other compounds, and thus more generally applicable.

A variety of methods (in silico, in vitro, in vivo) have been employed to measure and understand different aspects of the use of the polymer and its properties. Methods descriptions are at some not detailed enough. Questions about the design of different experiments. Some descriptions are not sufficient. Not always clear what the number of replicates refers to.

Some details and controls are missing to truly understand how this compound delivers (in this case) insulin, and to allow for

More specific comments and questions below.

Thank you for your positive comments on our work.

Comment 1) Control in the form of free Cy5 (or as close to this as possible) should be included – to ensure that the tracking both in vivo and in vitro were not (partly) due to the label detaching from the polymer. The authors should comment on the risk of quantifying PK only on the label.

Authors' response: We sincerely appreciate your instruction. We have conducted experiments as required:

Stability of the labeled conjugate: We performed stability studies under in vivo-mimicking conditions (incubation in 10% FBS at 37 °C). ~99.3% of Cy5 remained covalently bound to OP-I for up to 12 hours, with free Cy5 levels below the detectable threshold (**Supplementary Fig. 7**).

Transdermal permeability of free Cy5: We also evaluated the skin-penetration capability of free Cy5 alone. Results in **Supplementary Fig. 24** showed that free Cy5 or its mixture with OP had no detectable transdermal permeability, confirming that unconjugated Cy5 could not enter the circulatory system or contribute to the PK quantification. Thus, the PK data observed for OP-I^{Cy5} reflected the true conjugate behavior rather than free label migration.

We have incorporated these updates into the revised manuscript and supplementary figures. (Page 4, Lines 109-110; Page 9, Lines 277-280)

Supplementary Fig. 7 | HPLC analysis of OP-I^{Cy5} stability in DMEM medium containing 10% FBS, with peak area percentage detected at 640 nm excitation and 660 nm emission over time. Data are mean ± SD of biological replicates (n = 3).

Supplementary Fig. 24 | The CLSM images of the mice dorsal skin slices after topical application for 4 h with OP^{Cy5}, a mixture of OP and Cy5, or free Cy5 (Cy5-eq. dose: 0.2 mL of 10 μg/mL; application area: 1.13 cm²).

Comment 2) What was the stability of the labelled conjugate over time and under in vivo-like conditions?

Authors' response: We sincerely appreciate your instruction. To address this, we dissolved OP-I^{Cy5} in cell culture medium containing 10% FBS and incubated it at 37 °C to mimic in vivo conditions. As analyzed by HPLC (**Supplementary Fig. 7**), the fluorescence intensity of OP-I^{Cy5} remained highly stable at least for 12 hours, with 99.3% of the Cy5 conjugation intact. These findings strongly indicate that OP-I^{Cy5} was stable.

These updates have been incorporated into the revised manuscript (Page 4, Lines 109-110).

Supplementary Fig. 7 | HPLC analysis of the OP-I^{Cy5} stability in DMEM medium with 10% FBS. Peak area detected at 640 nm excitation/660 nm emission was normalized to the initial value. Data are mean \pm SD (n = 3).

Comment 3) The samples from PK studies and permeation studies were analyzed for presence of insulin by ELISA.

Authors' response: We greatly appreciate your suggestion. We performed additional experiments to use ELISA for analyzing samples from both pharmacokinetics (PK) and permeation studies.

i) PK studies: ELISA results in **Supplementary Fig. 8b** demonstrated that OP-I slightly prolonged the half-life of insulin compared to control groups, similar to the profiles using fluorescence measurement.

ii) Permeation studies: ELISA kit was used to quantify insulin levels within the skin, revealing a significant increase in insulin accumulation over time following transdermal administration (**Supplementary Fig. 10**).

We have incorporated these data into the revised manuscript (Page 4, Lines 127-129; Page 5, Lines 163-164) with additional discussions and figures.

Supplementary Fig. 8b, Blood clearance profiles of insulin and OP-I analysed by ELISA. Insulin-eq. dose (1 U/kg) of OP-I or insulin was intravenously injected via the tail vein. The insulin concentration relative to that of the first sampling point is shown as a function of time. Data are mean \pm SD (n = 5).

Supplementary Fig. 10 Insulin concentrations in the mouse dorsal skin at 2 or 4 hours post-topical application of OP-I, detected by ELISA. (insulin-eq. dose: 116 U/kg; 0.2 mL of 0.5 mg/mL solution applied on 1.13 cm² dorsal skin); n = 3.

Comment 4) From the methods description it seems as if the PK and PD was done in the same animals, but why are PK studies the reported as n=8 and PD as n=5? (e.g. figure 3) – was it not the same animals that are used for these assessments?

Authors' response: Thank you for your insightful question.

The PK and PD assessments were conducted in separate groups of animals rather than in the same cohort. This is because blood samplings can cause stress to the animals, which influences their metabolic state and, consequently, the PD. For instance, frequent blood samplings induce stress responses that alter the mouse blood glucose level. So we used separate groups of mice to minimize this effect.

Comment 5) In the methods section for the PK/PD, the dosing of the s.c. should be described, including the actual dose given.

Authors' response: We sincerely appreciate your instruction. We have now added the necessary information in the revised Methods section (Page 20, Lines 781-782).

Comment 6) Please explain why the dose of insulin s.c. used is so high - 5 U/kg? Typically, rats are dosed 1, max 2 U/kg and (healthy) mice around 0.5–1 U/kg (although varies a lot in literature). Were other doses tested? These might give similar profiles to the one observed for 5 U/kg.

Authors' response: Thank you for your insightful question. This higher dose was selected based on literature and other considerations. For instance, several recent studies used a 5 U/kg dose in their experiments (Nat. Commun., 2022, 13, 6649; Nat. Nano., 2020, 15, 605-61; Nat. Nano., 2024, 19, 1880-1891).

To answer your question, we conducted experiments in both healthy and diabetic mice using doses of 1 U/kg and 5 U/kg. The results in **Fig.R1** showed that the 5 U/kg dose achieved a faster glucose-lowering effect and a longer duration of action. Therefore, we thought that the 5 U/kg dose was more suitable as a positive control group.

Fig. R1 BGLs of healthy mice and diabetic mice after s.c. injection of different doses of native insulin. Data are presented as mean \pm SD; n = 5.

Comment 7) The skin was prepared using depilatory cream – was it controlled that this cream/procedure did not induce any changes in the skin permeability barrier?

Authors' response: We deeply appreciate your instruction.

We conducted the following experiments and excluded the cream effect: Histological analysis (H&E staining) and TUNEL staining were performed on the skin samples from minipigs treated with the depilatory cream, compared to untreated healthy control skin (**Extended Data Fig. 4 d,e**). The results showed no significant differences in skin architecture (e.g., stratum corneum, epidermis, or dermis layers) between the treated and control groups, confirming that the cream did not compromise skin structure.

Furthermore, in the minipig hypoglycemia experiments (**Supplementary Fig. 21**), topical application of insulin or PEG-conjugated insulin (PEG-I) dispersed in the cream did not significantly reduce blood glucose levels. This also confirmed that the depilatory cream did not change the skin barrier.

Collectively, these results demonstrate that the depilatory cream and procedure used did not compromise the skin's permeability barrier.

Extended Data Fig. 4 d,e, Representative images of H&E staining (d) and immunohistological staining with TUNEL assay (green) and Hoechst (blue) (e) of the treated sites of the minipig abdominal skin after 4 h of topical treatment with OP-I, PEG-I, native insulin, or PBS (insulin-eq. dose: 40 mL of 1 mg/mL, 29 U/kg, application area: 400 cm²). The untreated skin was designated as the Control group. Data are from n = 3.

Supplementary Fig. 21 | BGLs of minipigs in each group at the timed points in Fig. 3i. Data are mean ± SD of biological replicates (n = 3; ns, not significant, one-way ANOVA).

Comment 8) What did the controls look like in terms of e.g. the OP + label and/or insulin co-administered in terms of blood glucose (and absorption)?

Authors' response: We appreciate your suggestion to validate whether OP can promote the penetration of the label and insulin by simple co-administration. We did the following experiments:

Skin permeation studies: Physical mixtures of OP with Cy5 or insulin could not enhance transdermal penetration of the cargo molecules, as assessed by *in vivo* assays (**Supplementary Fig. 24** and **25**).

Transdermal hypoglycemic assays: Topical application of a mixture of OP and insulin had no effect on blood glucose levels in diabetic mice (**Supplementary Fig. 26**).

So, mixing OP with Cy5 or insulin could not grant them skin permeation. We have incorporated these results into the revised manuscript (Page 9, Lines 277-281).

Supplementary Fig. 24 | The CLSM images of the mouse dorsal skin slices after topical application for 4 h with OP^{Cy5}, a mixture of OP and Cy5, or free Cy5 (Cy5-eq. dose: 0.2 mL of 10 μg/mL; application area: 1.13 cm²).

Supplementary Fig. 25 | The CLSM images of the mouse dorsal skin slices after topical application for 4 h with OP-I^{Cy5}, a mixture of OP and insulin^{Cy5}, or insulin^{Cy5} (Cy5-eq. dose: 0.2 mL of 10 μg/mL; application area: 1.13 cm²).

Supplementary Fig. 26 | BGLs in the diabetic mice after topical application of PBS, OP-I, or the mixture of OP and insulin (insulin-eq. dose: 116 U/kg; 0.2 mL of 0.5 mg/mL solution applied on 1.13 cm² dorsal skin); n = 5.

Comment 9) Why are rats/pigs used in some experiments for imaging (e.g. Figure 2) and not mice? Why is the in vivo proof of concept done in one animal species and the compatibility assessment and MoA studies in other species?

Authors' response: Thank you for your insightful questions regarding the choice of animal models in our study. We appreciate the opportunity to clarify our rationale.

Why rats/pigs are used for imaging (e.g., Figure 2) and not mice? In Fig 2d, we tried to precisely image the distribution of OP-I's localization in lymphatic vessels. The rat lymphatic vessels are larger than those in the mouse skin, which allows for better imaging. In Fig 2e, we intended to demonstrate that OP could also permeate pig skin, which is closer to human skin.

Why in vivo proof of concept is done in one animal species and compatibility assessment and MoA studies in others? Our strategy in choosing one species (e.g., mice) for in vivo proof of concept and others (e.g., rats and pigs) for compatibility assessment and mechanism of action (MoA) studies is based on the specific strengths and limitations of each animal model. Mice are widely used for initial proof-of-concept studies due to their well-established genetic background, ease of handling, and cost-effectiveness. However, as mentioned above, their thin skin and fine structure make detailed imaging and mechanistic studies difficult.

Rats and pigs offer several advantages for compatibility assessment and MoA studies. These larger animals enable more detailed and precise imaging, for example, the distribution and effects of OP-I in tissues that are more representative of human physiology.

Furthermore, the consistent results across different species validated the skin permeability of OP and OP-I.

Comment 10) The method description of the minipigs studies only described how the dosing was done, not how samples were taken and analyzed.

Authors' response: We apologize for not presenting this information clearly.

The blood glucose levels of the minipigs were monitored using the Dexcom G4 Platinum Continuous Glucose Monitor System (CGMS), whose sensors were implanted subcutaneously to measure interstitial fluid glucose.

The skin samples of minipigs after the treatment: At the end of the experiment, the minipigs were euthanized, and the treated skin sites were carefully excised. Subsequently, the samples were processed for sectioning, followed by Hematoxylin-Eosin (H&E) staining and TUNEL staining. Additionally, skin samples from the same part of healthy minipigs without treatment were collected as controls.

We have added these detailed descriptions to the revised manuscript (Page 21, Lines 808-811).

Comment 11) It is stated that compatibility studies are done to investigate morphological alterations and e.g. immune responses, in mice exposed to the OP-I but it is not clearly described how, nor are data shown.

Authors' response: Thank you for highlighting the need to clarify our compatibility studies. We conducted morphological and immunological assessments of OP-I in mice, as detailed below with data presented in **Extended Data Fig. 4a–c**:

SEM imaging of stratum corneum: Compared to the PBS control group, the skin treated with OP-I showed intact, continuous lamellar structures in the stratum corneum, with no evidence of

widened cell (cell gaps) or structural damage. The results indicate that OP-I did not disrupt the skin barrier morphology.

H&E staining for inflammation: Histological analysis of skin sections showed the preserved epidermal architecture (including the stratum corneum) in the OP-I-treated mice, with no signs of inflammation (e.g., neutrophil/lymphocyte infiltration, tissue disruption). The lack of inflammatory cell recruitment confirmed OP-I did not induce local immune responses.

TUNEL staining for apoptosis: TUNEL staining showed no significant increase in apoptotic cells in OP-I-treated skin compared to controls, demonstrating that OP-I did not trigger cytotoxicity or cell death.

Collectively, these results confirm that OP-I did not change the skin's structural integrity and did not induce inflammation or apoptosis, supporting its biocompatibility.

Extended Data Fig. 4 | a-c, Representative images of SEM (a), H&E staining (b), and immunohistological staining with TUNEL assay (green) and Hoechst (blue) (c) of the mouse dorsal skin after 4 h of topical treatment with OP-I, PEG-I, native insulin, or PBS (insulin-eq. dose: 116 U/kg; 0.2 mL of 0.5 mg/mL; application area: 1.13 cm²).

Comment 12) Abstract, line 31: Rephrase. “As efficiently as s.c. injected insulin” – an overstatement. While impressive BGC lowering effects, the BA/Fab should be included in statements like this. The topical doses are (for good reasons) x times higher than s.c., which should be accounted for.

Authors' response: Thank you for your insightful instruction. We have revised the sentence in the abstract accordingly (Page 1, Lines 29-33):

“As a result, OP-conjugated insulin efficiently permeates through the skin into the blood circulation; transdermal administration at a dose of 116 U/kg to type 1 diabetic mice quickly lowers their blood glucose levels (BGL) to the normal range, and a transdermal dose of 29 U/kg normalizes the BGLs of diabetic minipigs”

Comment 13) P4, line 127-129: the authors claim that the extended circulation time/half-life of the conjugate is due to non-stickiness to plasma proteins, as was also discussed in ref. 23. However, it is a well-known strategy to extend circulating time due to “stickiness” to plasma proteins. The authors should reflect on this.

Authors' response: We sincerely appreciate your insightful comment.

Indeed, drugs engineered to bind and "hitchhike" albumin is an approach to gain long circulation (Angew Chem. Int. Ed. Engl., 2018, 57, 8994).

So we added “The nonstickness of OP made it unable to hitchhike plasma proteins²⁷.” in Page 4, Lines 131-132.

Comment 14) P5, line 157-158: ...adipose tissue labelled with BODIPY.... Please clarify if it is Ab-labelling or ? Also: Why was the PEG-I and insulin FITC-labelled and the OP-I Cy5 labelled (extended data, fig 3)? This can be important for the interpretation of images as FITC fades much faster than Cy5. The authors should address this in their interpretation of images. It seems that OP-I FITC was also used for some studies (figure 2), why switch between labels?

Authors' response: We sincerely appreciate your comments:

i) Clarification of BODIPY labeling of adipose tissue: BODIPY labeling adipose tissue does not need an antibody. BODIPY is a small molecular fluorescent dye commonly used for lipid staining, including in adipose tissue (Nat. Protoc., 2010, 5, 912-20). We have updated the manuscript to include this reference and a brief explanation of the utility of BODIPY in our study.

ii) Reason for switching between FITC and Cy5 labeling for PEG - I, insulin, and OP - I: We chose different fluorescent labels (FITC and Cy5) for PEG - I, insulin, and OP-I according to our various experiments' requirements. In many experiments, we needed to label different cellular components or structures, such as stratum corneum lipids, cell membranes, or lymphatic vessels, in addition to OP or insulin labeling. Therefore, we had to use Cy5- or FITC-labeled OP or OP-I to avoid fluorescence overlap/interference with the skin-labeling dyes.

iii) FITC fades faster than Cy5. As the samples were not for continuous CLSM imaging, this faster fading should make little difference for instant imaging.

Comment 15) P6, line 173: typo. Correct plasm to plasma

Authors' response: Thank you for pointing out this typo. We have corrected it accordingly in the revised manuscript.

Comment 16) P20, line 742: change transdermally to topically (or cutaneously)

Authors' response: Thank you for pointing out this typo. We have corrected it accordingly in the revised manuscript.

Comment 17) WB: It is argued that the downstream pAKT is activated, and while the pIR is not activated by PBS control injections, this seems to be the case for pAKT (Extended Data Fig. 2f). Please explain.

Authors' response: We greatly appreciate your critical comment. To address your concern, we did the WB experiments again as follows:

We performed three independent WB experiments (with quantitative densitometry) and systematically analyzed IR and Akt activation in the muscle tissues of STZ-induced diabetic mice after transdermal administration of OP-I, using subcutaneous insulin injection as a positive control. OP-I treatment significantly activated both the IR signaling pathway (phosphorylated IR, pIR) and its downstream target Akt (phosphorylated AKT, pAKT) in the muscle tissues. PBS control injections showed no detectable activation of pIR or pAKT. Data were normalized to total IR and Akt protein levels within each sample (**Supplementary Fig. 17**; detailed Methods are in Page 18, Lines 692-711).

We have incorporated these new results into the revised manuscript (Page 6, Lines 181-183).

Supplementary Fig. 17 | Western blot analysis of phosphorylated IR (pIR) and phosphorylated AKT (pAKT) levels of the mouse skeletal muscle. Tissues were harvested from the STZ-induced diabetic mice 4 h after topical application of OP-I (insulin-eq. dose: 116 U/kg; 0.2 mL of 0.5 mg/mL solution applied on 1.13 cm² dorsal skin) or 1 h after subcutaneously (s.c.) injected with native insulin (5 U/kg) as a positive control. (n = 3).

Source Data for **Supplementary Fig. 17**

Comment 18) MD: If the polymer is positively charged at < 7.4 , one would expect that it would bind/assemble insulin with a pI of 5.3 and in that way form a depot in the skin. Any reflections on this? Were the simulations done with more molecules in the box to investigate this?

Authors' response: We appreciate your comment. To investigate the interaction, we conducted molecular dynamics (MD) simulations involving multiple insulin, OP and OP-I molecules under pH 6.0 conditions. During the 400-ns simulations (**Supplementary Fig. 27** and **Extended Data Movie 5**), we observed that OP, owing to its hydrophilic nature, does not form stable binding with insulin, even though they carry opposite charges at pH 6.0. Additionally, the MD results suggest that insulin molecules may aggregate via hydrophobic interactions at high concentrations.

This is reasonable because, at pH 6.0, OP is only slightly positively charged. Its low positive charge density is unable to make OP bind insulin firmly.

We have incorporated these findings into the revised manuscript (Page 9, Lines 299-302).

Supplementary Fig. 27 | Snapshots of 400-ns MD results with multiple molecules showing that OP did not bind insulin even though they carry opposite charges at pH 6.0.

Extended Data Movie 5 (separate file) Molecular dynamics trajectories involving multiple insulin, OP, and OP-I molecules at pH 6.0.

Comment 19) Figure 4: What do $n=3$ mean for these results. The images are representative of 3? Are the MD done three times and the shown is a representative?

Authors' response: We have done MD three times. The snapshot shown was representative, and the data for statistical calculations were collected from all MD trajectories.

Comment 20) Figure 5d: How do the buffer control and membrane-stained cells look? From the membrane stain, the cells seem to exist not in a confluent monolayer but rather as individual cells, and the membrane of many of them seems quite thick (is it maximum projection intensities that is displayed?).

Authors' response: We sincerely appreciate your careful examination on Fig. 5d. To answer your question, we conducted additional control experiments by staining the cell membranes of untreated HaCat cells (without any experimental treatment). As shown in **Fig. R2**, the staining of normal cells exhibits similar variability in membrane “thickness”, indicating that this phenomenon is a normal occurrence and not an artifact caused by experimental treatments or imaging parameters. This variability likely is due to 2D projection of 3D cells (**Fig.R3**).

Fig. R2 CLSM imaging of HaCat cell membranes stained with NBD-C6-HPC.

Fig. R3 2D projections of the same thick membrane may be different.

Comment 21) Figure 5f: how is the relative transfer efficacy calculated from the flow cytometry data? Which gating was used? Some explanation is given in the methods description but it is not clear how the authors get three times higher transfer after 12 h.

Authors' response: For direct transfer experiment, the OP-I^{Cy5} treated HaCat cells were extensively rinsed with sterilized PBS, isolated, and mixed with untreated HaCat-GFP cells at the same cell density. The mixed cells were co-cultured. The cells were washed twice with PBS and analyzed by flow cytometry. Flow cytometry was performed on a BD FACS Calibur and analysis

was performed using the FlowJo software. First, an FSC-A vs. SSC-A gate was used to exclude debris and select for live cells based on their forward scatter (FSC) and side scatter (SSC) characteristics. Next, a GFP gate (FL1-H) was set to include cells that exhibit fluorescence in the GFP channel. Subsequently, a Cy5 gate (FL4-H) was established to isolate cells that are positive for Cy5 fluorescence. Finally, an intersection gate was applied between the GFP and Cy5 gates to identify cells that are positive for both GFP and Cy5, representing the dual-stained population.

$$\text{Transfer efficiency (\%)} = \frac{\text{Number of HaCat}^{\text{GFP+Cy5}} \text{ cells}}{\text{The total counted number of HaCat}^{\text{GFP}} \text{ cells}} \times 100$$

The description of the method is in Page 23, Lines 911-918.

Comment 22) It is concluded that the polymer-Insulin conjugate does not affect the morphology of the skin cells, nor cause any inflammation or cell death. But does this polymer fluidize the skin or in another way change the barrier to absorption of other compounds? It would be highly beneficial to demonstrate that the enhanced permeation was specific for the conjugated insulin.

Authors' response: We greatly appreciate your valuable suggestion. To address this, we conducted FTIR spectroscopy to analyze the SC lipid order of mouse dorsal skin after treatment with OP, OP-I, or oleic acid (positive control) for 24 hours using Franz diffusion cells.

Methylene vibration peaks at $\sim 2850 \text{ cm}^{-1}$ (symmetric) and $\sim 2900 \text{ cm}^{-1}$ (asymmetric) reflect the packing order of lipid hydrocarbon chains in the stratum corneum. A blue shift (higher wavenumber) indicates reduced lipid order and increased fluidity, correlating with enhanced skin permeability (Biochim. Biophys. Acta Biomembr., 2008, 1778, 1344). As shown in **Supplementary Fig. 23**, PBS-treated skin had methylene peaks at 2849.63 cm^{-1} (symmetric) and 2918.18 cm^{-1} (asymmetric), representing normal lipid order. OP or OP-I treated skins red-shifted $0.1 \text{ cm}^{-1} / 0.58 \text{ cm}^{-1}$ or $0.2 \text{ cm}^{-1} / 0.47 \text{ cm}^{-1}$, respectively. This indicates no notable change in lipid packing order or fluidity. Oleic acid (positive control) caused blue-shifted by $2.14 \text{ cm}^{-1} / 3 \text{ cm}^{-1}$, confirming substantial lipid fluidization, consistent with the result in literature (Mol. Pharm., 2023, 20, 6237.). These results demonstrate that OP and OP-I enhance insulin permeation without altering stratum corneum lipid order or inducing fluidization, unlike oleic acid. The permeability enhancement is, therefore, specific to the conjugated insulin-OP complex rather than nonspecific barrier disruption.

We have incorporated these findings into the revised manuscript (Page 9, Lines 275-277).

Supplementary Fig. 23 | The FTIR spectra of the mouse dorsal skin stratum corneum 24 h after topical application of PBS, OP (0.23 mg/cm²), OP-I (OP-eq. dose: 0.23 mg/cm²) or oleic acid (22.7 mg/cm²) using a Franz diffusion cell system.

Comment 23) A question of whether co-administration instead of conjugation would have any effect on skin permeation *in vitro/ex vivo* and absorption *in vivo*? The authors should discuss this, and data show the benefit of conjugating.

Authors' response: We sincerely appreciate your insightful question regarding the impact of co-administration versus conjugation. To address this, we conducted direct comparisons between conjugate formulations and physical mixtures in both permeability and hypoglycemic efficacy experiments:

Skin permeation studies (*in vivo*): As shown in **Supplementary Fig. 24** and **Fig. 25**, physical mixtures of OP with Cy5 or insulin^{Cy5} failed to enhance skin penetration of the cargo molecules, whereas the OP-conjugated counterparts (OP^{Cy5} or OP-insulin^{Cy5}) had significantly improved transdermal permeability. This indicates that covalent conjugation is essential for OP to exert its skin-penetrating effects.

Transdermal hypoglycemic assays (*in vivo*): Topical application of a physical mixture of OP and insulin produced no significant hypoglycemic effect (**Supplementary Fig. 26**).

We have incorporated these findings into the revised manuscript (Page 9, Lines 277-281).

Supplementary Fig. 24 | The CLSM images of the mouse dorsal skin slices after topical application for 4 h with OP^{Cy5}, a mixture of OP and Cy5, or free Cy5 (Cy5-eq. dose: 0.2 mL of 10 μg/mL; application area: 1.13 cm²).

Supplementary Fig. 25 | The CLSM images of the mouse dorsal skin slices after topical application for 4 h with OP-I^{Cy5}, a mixture of OP and insulin^{Cy5}, or insulin^{Cy5} (Cy5-eq. dose: 0.2 mL of 10 μg/mL; application area: 1.13 cm²).

Supplementary Fig. 26 | BGLs in the diabetic mice after topical application of PBS, OP-I, or the mixture of OP and insulin (insulin-eq. dose: 116 U/kg; 0.2 mL of 0.5 mg/mL solution applied on 1.13 cm² dorsal skin); n = 5.

Comment 24) It is claimed that the approach is likely viable also for other macromolecular drugs, however, it should be considered that the conjugates are essentially new drugs, which might limit the general applicability.

Authors' response: Thank you for your insightful comment.

Indeed, conjugates are considered new drugs and thus need additional efforts for clinical translation. However, as long as the conjugation offers significant benefits, the conjugates are still highly translational, as seen in the PEG-conjugates (Nat. Rev. Drug Discov., 2019, 18, 273).

Furthermore, we have found a way to utilize OP for the transdermal delivery of peptides and insulin without conjugation. We will report it soon.

Thank you again for your valuable feedback. We hope this response addresses your concerns.

RESPONSES TO REFEREES' COMMENTS

Nature Submission ID: 2024-05-10937B

Title: A skin-permeable polymer for noninvasive transdermal insulin delivery

Authors: Qiuyu Wei, Zhi He, Zifan Li, Zhuxian Zhou, Ying Piao, Jianxiang Huang, Yu Geng, Runnan Zhang, Yaqi Fu, Jiayi Ye, Yue Yuan, Haoru Zhu, Jiaheng Zeng, Yan Zhang, Quan Zhou, Mingyu Xu, Shiqun Shao, Jianbin Tang, Jiajia Xiang, Rongjun Chen, Ruhong Zhou & Youqing Shen

We would like to express our sincere gratitude to the referees for their helpful and constructive suggestions, which we believe have greatly improved the revised version. We have taken careful note of their comments and revised our manuscript accordingly.

A point-by-point response to each of the referees' comments is provided below. The revisions in the revised manuscript are highlighted in blue.

Reviewer #2:

This reviewer believes that the authors have adequately addressed all of the comments and concerns raised in this reviewer's previous report. Provided that the issues highlighted by the other reviewers are also satisfactorily resolved, this reviewer considers the manuscript suitable for publication.

➤ Thank you very much for your positive feedback.

Reviewer #3:

My comments have been adequately addressed.

➤ Thank you very much for your positive assessment of our responses.

Reviewer #4:

The authors have addressed my earlier comments; to quite some extent this needed conduction of new/additional experiments to answer the questions. Most of these are now included in the manuscript and SI. Overall, the findings and earlier claims are substantiated, and the manuscript is improved. However, I have requests to some of the comments:

➤ Thank you for acknowledging our responses.

Comment 1) Re. response to comment 4): the argument not to stress the animals and by that affect the PD readout is good, however if analysis is done using the same blood samples, it should not stress the animals. I think it should be mentioned that it was indeed done in different animals, since the PK and PD profiles in fig 3 do not correspond - the PD seem much more extended than what is reflected in the PK profile.

➤ Thank you for your detailed comments. To avoid ambiguity, the revised Figure 3 caption now clearly states that the PK and PD data are derived from separate animal groups (Lines 413-414).

Comment 2) Re. response to comment 6): The authors have conducted studies with both 1 U/kg and 5 U/kg insulin dosed s.c. These data should be included in the SI. They are important to show that there is not a huge difference in the AOC of the two doses, and for sure not a five-fold increase with

the higher dose compared to the lower dose. There could be many reasons for this, for example that the insulin (especially in the higher concentration formulation) partly aggregate and thus does not exist in free form to be absorbed from the site of injection to the blood stream. Although the authors do not underline bioavailability data of their conjugate, the dosed conc. of the reference (here s.c.) will affect the bioavailability hugely. Indeed, the authors show - in response to comment 18 - that the by MD simulations, the insulin molecules aggregate. It is of course known that insulin assembles into multimers, that this is affected by surrounding factors and dose etc. and potentially larger aggregates.

- Thank you for your suggestion. As recommended, we have included the data in Supplementary Fig. 7 and additional discussion in the revised manuscript (Lines 96-98). We acknowledge that the lack of a dose-proportional AOC increase may be due to factors such as partial insulin aggregation, especially in higher-concentration formulations, which reduces the amount of free insulin available for absorption. This suggests that our future research and formulation development should focus on preventing this issue through special processes or the use of excipients.

Comment 3) Re. response to comment 13): the word nonstickiness is unspecific and somewhat non-scientific. Can this be specified?

- Thank you for pointing out this issue. We have replaced the term "nonstickiness" with the more precise and scientifically appropriate term "protein resistance" (Lines 120-121).

Comment 4) Re. response to comment 20): So the images are maximum projection images, it seems. This should be noted.

- Thank you for your valuable suggestion. The images in Figure 5d are maximum projection images. As suggested, we have noted this in the Methods part (Line 859).

Comment 5) Re. response to comment 24): Sounds promising, but is this an approach much different from the control that I asked for in comment 23?

- Thank you for your nice question. The non-conjugation strategy for transdermal delivery of peptides and insulin using OP (to be reported soon) differs fundamentally from the physical mixture tested as the control in Comment 23. Specifically, because OP and insulin do not interact inherently, in the non-conjugation approach, we have to introduce a third component that binds OP and proteins to form three-component nanoparticles for skin penetration, rather than simply mixing OP with proteins or peptides.